# Molecular requirements for PLK1 activation by T-loop phosphorylation

Arianna Esposito-Verza [1,2]✉, Duccio Conti [1], Paulo D Rodrigues Pedroso [1,3], Lina Oberste-Lehn [1], Carolin Koerner[1], Sabine Wohlgemuth[1], Artem Mansurkhodzhaev[1], Ingrid R Vetter [1], Marion E Pesenti [1] & Andrea Musacchio [1,2]✉

## Abstract

**Activation of PLK1, a master mitotic kinase, requires phosphorylation of its activation segment on Thr210, within a basic consensus sequence for Aurora kinases. Aurora B-dependent phosphorylation of Thr210 has been reported, but other evidence identified a strict requirement for the Aurora A partner Bora for Thr210 phosphorylation. Here, we investigate the elusive mechanistic basis for this requirement. We show that Aurora A:Bora phosphorylates Thr210 of PLK1 in vitro. On the contrary, T210 was not phosphorylated by isolated Aurora A, additional Aurora A:activator complexes, or Aurora B:INCENP, even when used at high kinase/substrate ratios. A transient interaction of Bora and PLK1, identified by structural modeling and probed mutationally, is uniquely required for Thr210 phosphorylation. Dependency on Bora for Thr210 phosphorylation is eliminated after mutating Lys208, in the Aurora consensus, into arginine. This conservative mutation turns PLK1 into a substrate of nearly all tested active Aurora kinases, including Aurora B. Collectively, these results shine a new light on the specificity of the PLK1 activation mechanism.**

**Keywords** kinetochore; Kinase; Cell Cycle; Polo-like Kinase; Aurora
**Subject Categories** Cell Cycle; Post-translational Modifications & Proteolysis; Structural Biology

See also: JA Miles et al, A Pillan et al & Monica Gobran & Peter Lenart et al

## Introduction

The spatial and temporal orchestration of a cell's entry into mitosis reflects the ordered activation of several master mitotic serine/threonine (Ser/Thr) kinases, including cyclin-dependent kinase 1 (CDK1), Aurora A and B, and Polo-like kinase 1 (PLK1) (Nigg, 2001; Saurin, 2018). Besides reciprocally controlling their own activation and the activity of several downstream targets, these kinases are also embedded in a regulated network of protein phosphatases, including different forms of the protein phosphatases 1 and 2A (PP1 and PP2A, respectively) (Brautigan and Shenolikar, 2018; Saurin, 2018).

To reach full activity, many protein kinases require post-translational phosphorylation of their activation loop (or T-loop) either by a distinct trans-activating kinase or by auto-phosphorylation (Bayliss et al, 2012; Huse and Kuriyan, 2002; Johnson et al, 1996). The phosphate group, typically attached to a threonine or serine residue, interacts with positively charged amino acids to induce conformational changes in the kinase that stabilize its active conformation. This activation mechanism has a prominent role in the temporal and spatial control of kinase activity. A full comprehension of its regulation is therefore a crucial aspect of dissecting how the activity of protein kinases is controlled in their specific biological context (Bayliss et al, 2012; Huse and Kuriyan, 2002; Johnson et al, 1996).

PLK1 activity is required for the success of several distinct events occurring predominantly between the G2 phase and the subsequent M and G1 phases, and that include, among several others, nuclear envelope breakdown, centrosome maturation and spindle assembly, chromosome bi-orientation, cytokinesis, as well as the deposition of the specialized centromere histone CENP-A in early G1 (Pintard and Archambault, 2018; Zasadzinska and Foltz, 2017). In G2, PLK1 becomes progressively activated through phosphorylation of Thr210 (in human PLK1) on the activation loop by trans-activation (Archambault and Carmena, 2012; Bayliss et al, 2012; Jang et al, 2002; Kelm et al, 2002; Lasek et al, 2016; Lee and Erikson, 1997; Qian et al, 1999; Silva Cascales et al, 2021). PLK1 then promotes mitotic entry mainly by activating Cell Division Cycle 25 (CDC25), which dephosphorylates and activates the master regulator of mitosis, Cyclin-dependent kinase 1 (CDK1) (Abrieu et al, 1998; Gheghiani et al, 2017; Gobran et al, 2025; Vigneron et al, 2018).

PLK1 phosphorylates target motifs matching the consensus sequence [D/N/E]-X-[**S/T**]-[I/L/M/V/F/W/Y] (Alexander et al, 2011; Grosstessner-Hain et al, 2011; Santamaria et al, 2011). This substrate preference likely explains why PLK1 is unable to auto-phosphorylate on Thr210, a residue embedded in a sequence motif

[1]Department of Mechanistic Cell Biology, Max Planck Institute of Molecular Physiology, Otto-Hahn-Straße 11, Dortmund 44227, Germany. [2]Centre for Medical Biotechnology, Faculty of Biology, University Duisburg-Essen, Essen, Germany. [3]Present address: i3S - Instituto de Investigação e Inovação em Saúde, Rua Alfredo Allen, 208, Porto 4200-135, Portugal. ✉E-mail: arianna.espositoverza@mpi-dortmund.mpg.de; andrea.musacchio@mpi-dortmund.mpg.de

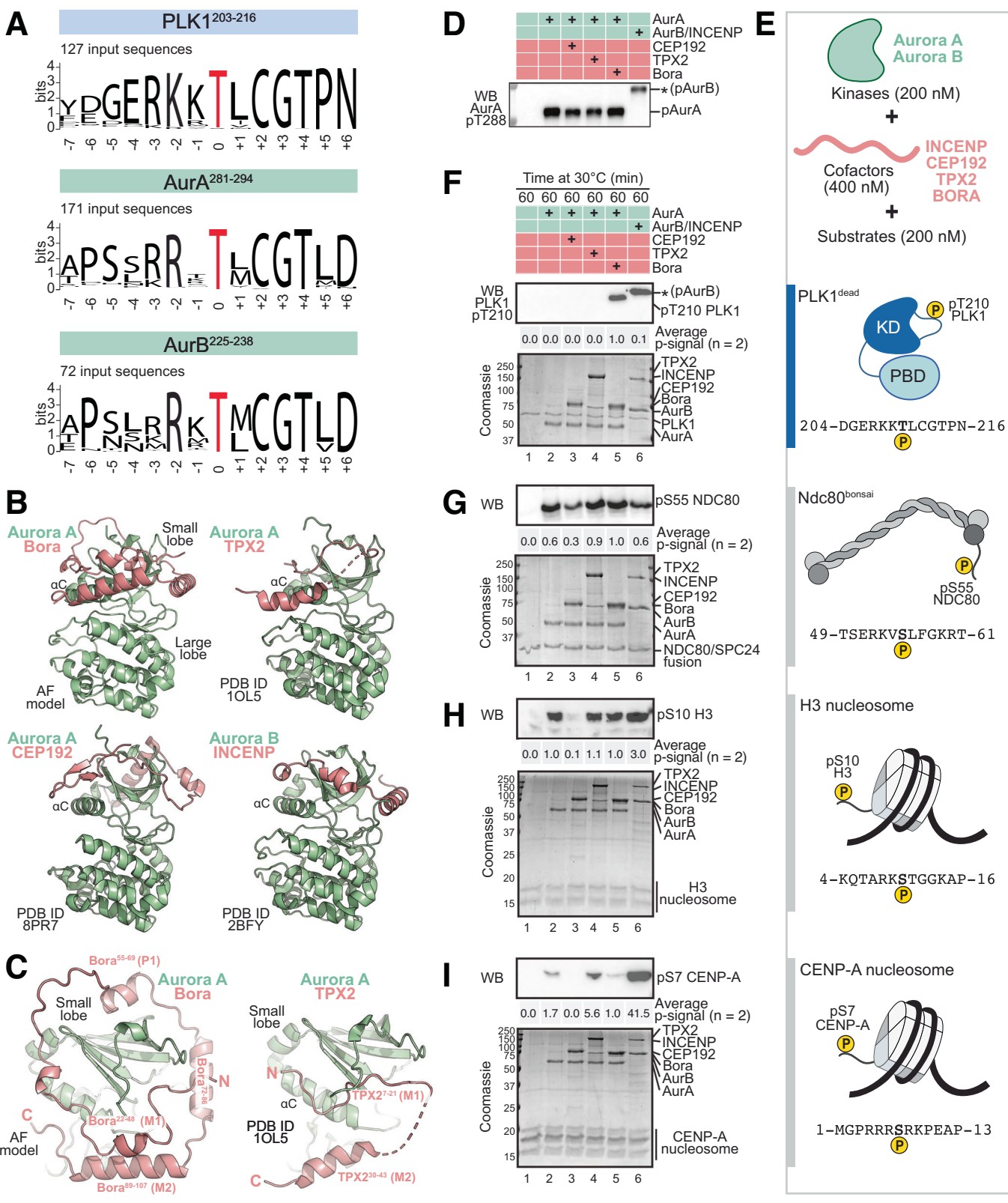

◄ **Figure 1. PLK1 is a poor substrate for Aurora kinases.**

(A) Sequence Logo for human PLK1, human Aurora A, and human Aurora B T-loop sequences. Sequences were obtained from TreeFam database, and Logos were generated with WebLogo (UC Berkeley). (B) AlphaFold3 (AF3) model of Aurora A in complex with Bora is shown in comparison to the experimental models of Aurora A bound to TPX2 (PDB code: 1OL5), CEP192 (PDB code: 8PR7), and to Aurora B in complex with INCENP (PDB code: 2BFY). Aurora A and Aurora B models are displayed as green cartoons, and protein cofactors are displayed as coral red cartoons. (C) Top view of the N-lobe of Aurora A bound to Bora (AF2 model) and TPX2 (PDB code: 1OL5). Coloring scheme as in (B). (D) Western blot showing activation loop phosphorylation of all Aurora kinase complexes. The anti-Aurora A pThr288 antibody cross-reacts with Aurora B pThr232 (indicated by an asterisk). (E) Schematic representation of proteins used for the biochemical assay, including Aurora A and B, the listed protein cofactors, and the substrates PLK1, NDC80$^{bonsai}$, H3 nucleosome, and CENP-A nucleosome. (F–I) Phosphorylation of PLK1 Thr210 (F), Ndc80 Ser55 (G), H3 Ser10 (H), and CENP-A Ser7 (I) was monitored by Western blotting with phospho-specific antibodies and quantified with Aurora A:Bora activity as reference (equal to 1.0). The arithmetic mean of two independent experiments is reported. A Coomassie Brilliant Blue-stained SDS–PAGE gel of the same samples was combined to verify equal protein loading between conditions. Source data are available online for this figure.

that is typically targeted by basophilic kinases of the AGC group, which includes Protein Kinase A, and closely related members such as the Aurora kinase family (Manning et al, 2002). The activation segments of Aurora A and B kinases share high sequence similarity to that of PLK1 (Fig. 1A), and these basophilic kinases can auto-activate through phosphorylation on T288 and T232, respectively (Littlepage et al, 2002; Walter et al, 2000; Yasui et al, 2004). Both Aurora A and Aurora B have been proposed to phosphorylate PLK1 on Thr210 to activate it (Archambault and Carmena, 2012). Aurora B has been implicated in PLK1 activation in the model organism *Drosophila melanogaster* (Carmena et al, 2012; Kachaner et al, 2014). Evidence of an Aurora B involvement in PLK1 activation in other organisms, however, remains limited. Conversely, there is ample evidence of a ubiquitous role of Aurora A in PLK1 activation in late G2 phase as well as in maintaining PLK1 activation during mitosis (Bruinsma et al, 2014; Gheghiani et al, 2017; Gobran et al, 2025; Macurek et al, 2008; Seki et al, 2008b; Tavernier et al, 2021; Vigneron et al, 2018).

Aurora A promotes PLK1 activation in complex with the protein cofactor Bora (Bruinsma et al, 2015; Hutterer et al, 2006; Macurek et al, 2008; Seki et al, 2008b; Tavernier et al, 2021). Bona fide activators of Aurora A, such as TPX2 and CEP192, contain unfolded segments that bind the kinase's N-terminal lobe to stabilize the active, inward conformation of the activation loop (Bayliss et al, 2003; Holder et al, 2024; Park et al, 2023; Zorba et al, 2014). A similar mechanism leads to activation of Aurora B by the IN-box segment of INCENP (Bishop and Schumacher, 2002; Honda et al, 2003; Segura-Pena et al, 2023; Sessa et al, 2005). Bora's intrinsic disorder and recognizable sequence similarities with TPX2 led to the proposal that it may activate Aurora A through a structural mechanism reminiscent of that of other Aurora A activators (Tavernier et al, 2021).

Besides interacting with Aurora A's kinase domain, Bora also interacts with PLK1 (Cheng et al, 2003; Elia et al, 2003a; Elia et al, 2003b). Phosphorylation of Ser252 of human Bora (hBora) by CDK1 promotes this interaction with the polo-box domain of PLK1 (PBD) (Chan et al, 2008; Gheghiani et al, 2017; Parrilla et al, 2016; Seki et al, 2008b; Silva Cascales et al, 2021; Tavernier et al, 2015; Tavernier et al, 2021; Thomas et al, 2016; Vigneron et al, 2018). The PBD is believed to stabilize an autoinhibited conformation of PLK1. Its engagement through a phosphopeptide has been proposed to relieve auto-inhibition and also to promote phosphorylation of the PLK1 activation loop (Seki et al, 2008b; Xu et al, 2013). Binding of the PLK1 PBD to pSer252 of Bora, however, is not required for Thr210 phosphorylation, which proceeds normally on the isolated kinase domain of PLK1 in the absence of the PBD (Tavernier et al,

2021). Moreover, Bora$_{1-224}$, a minimal Bora fragment lacking the PLK1 docking site but still capable of interacting with Aurora A, remains proficient in PLK1 phosphorylation (Tavernier et al, 2021; Thomas et al, 2016). PLK1 binding to Ser252 is instead involved in partial degradation of Bora at the onset of mitosis through β-TrCP, a subunit of the SCF ubiquitin ligase (Chan et al, 2008; Seki et al, 2008a). More recent studies have implicated phosphorylation of Ser252 in PLK1 dimerization and its controlled activation (Raab et al, 2022).

Another way in which CDK activity regulates Bora towards PLK1 is through phosphorylation of Ser112 of Bora. After phosphorylation, this residue can act in trans as a mimic of Aurora A Thr288 phosphorylation, activating the dephosphorylated kinase and even bypassing the deleterious effects on Aurora A activity resulting from mutating Thr288 into valine (Tavernier et al, 2021). Mutation of Ser112 to a non-phosphorylatable residue also prevents timely mitotic entry (Tavernier et al, 2021), indicating that its phosphorylation enables exquisite control of the Aurora A:Bora complex by CDK activity around mitotic entry.

If the only contribution of Bora to PLK1 T210 phosphorylation were through the activation of Aurora A, other Aurora:cofactor complexes, such as Aurora B:INCENP or Aurora A:CEP192, may also be expected to activate PLK1, unless prevented by strict lack of co-localization. Indeed, besides the already mentioned evidence of PLK1 activation by the Aurora B:INCENP complex at centromeres and the central spindle during cytokinesis, evidence of PLK1 phosphorylation by Aurora A:CEP192 or Aurora A:Protein furry homolog (FRY) at centrosomes has also been reported (Carmena et al, 2012; Gallaud et al, 2022; Ikeda et al, 2012; Joukov and De Nicolo, 2018; Joukov et al, 2010; Joukov et al, 2014; Kachaner et al, 2014; Meng et al, 2015; Shao et al, 2015). Puzzlingly, however, compensation by other kinases of the decrease in pThr210 levels observed upon Bora depletion has not been observed (Bruinsma et al, 2014; Macurek et al, 2008), supporting the specificity of the Bora-dependent mechanism.

To shed further light on this issue, it is therefore important to investigate the precise mechanism through which Bora activates Aurora A and targets it to PLK1, and to explore whether comparable mechanisms are employed by other cofactors of Aurora kinases. Here, we dissected the biochemical underpinnings of the Bora:Aurora A catalytic complex and of its activity towards PLK1 in vitro, and validated our findings in human cells when applicable. Our results indicate that Aurora A:Bora relies on a transient interaction to the PLK1 kinase domain for Thr210 phosphorylation. This interaction overcomes an intrinsic inefficiency of Aurora kinases in phosphorylating the non-canonical

Thr210 motif, that is effectively bypassed by the mutation of a single crucial gatekeeper residue.

## Results

### A structural prediction of the Aurora A:Bora complex

Previous structural work identified distinct but related binding modes for Aurora kinase activators (herewith referred to as Aurora cofactors). These 40- to 100-residue fragments are devoid of a stable own conformation, but adopt one upon binding to the small lobe of the cognate Aurora kinase, embracing it (as in the case for CEP192 and INCENP), or simply lining it (as is the case for TPX2; see gallery in Fig. 1B). We used AlphaFold (AF) (Jumper et al, 2021) to obtain a structural model of the complex of Aurora A with a previously identified (Tavernier et al, 2021) minimal activation segment of Bora (residues 18–120). AF predicted with high confidence that Bora$^{18-120}$ wraps around the small lobe of Aurora A, extending even beyond a full circumference (Figs. 1C and E-V1A,B). In the complex of Aurora A with TPX2, two fragments of TPX2 (previously identified as M1 and M2, and corresponding approximately to segments TPX$_{7-21}$ and TPX$_{30-43}$ (Tavernier et al, 2021) form a hairpin near the αC helix of Aurora A (Fig. 1C). The contact with Bora predicted by AF is significantly more extensive, but segments encompassing Bora$_{22-48}$ and Bora$_{89-107}$ seem to correspond to the TPX2 M1 and M2 segments, as predicted (Tavernier et al, 2021), with significant differences (e.g., the M1 regions run in opposite directions in the Bora and TPX2 models, Fig. 1C). In both cases, there is a predominance of contacts with the kinase through aromatic residues. In M1, the side chains of Phe25, Leu39, and Phe45 of Bora occupy positions equivalent to those of Tyr8, Tyr10, and Phe19 of TPX2, respectively. In M2, Phe103 and Phe104 are equivalent to Trp34 and Phe35 of TPX2, respectively.

### Kinases and cofactors contribute to substrate preference in vitro

We assembled several Aurora kinase:cofactor complexes and assessed their ability to phosphorylate PLK1 and other substrates in vitro in comparison to Aurora A:Bora. Specifically, we expressed and purified recombinant human Aurora A, and combined it with fragments of Bora (residues 1–224), TPX2 (full length), and CEP192 (residues 400–600) encompassing the predicted or experimentally validated binding sites for Aurora A (Fig. EV1C). We also co-expressed Aurora B kinase with INCENP (residues 351-C) (a detailed procedure is described in "Methods"). Successful binding of the cofactors to Aurora A was verified by analytical size-exclusion chromatography (SEC) (Fig. EV1D). With or without cofactor, western blots identified T288 in the activation segment of Aurora A to be phosphorylated (pT288; Fig. 1D, top), while no signal was observed after dephosphorylation of Aurora A with λ phosphatase, or when T288 was mutated to valine (T288V; Fig. EV1E,F, lanes 1–3).

Next, we used these Aurora:cofactor complexes in kinase assays in vitro with various substrates (Fig. 1E). Substrates included, besides Thr210 in the PLK1 activation loop, Ser55 in the N-terminal tail of the kinetochore protein NDC80, Ser10 in the N-terminal tail of histone H3, and Ser7 in the N-terminal tail of the

centromere-specific histone H3 variant CENP-A (Cheeseman et al, 2006; DeLuca et al, 2006; Hsu et al, 2000; Zeitlin et al, 2001) (Fig. 1E–H). Phosphorylation of these substrates was monitored by using phospho-specific antibodies.

The Aurora A:Bora complex phosphorylated PLK1's Thr210 very efficiently in the conditions of our assay. Conversely, Aurora A alone, or its complexes with TPX2 or CEP192, were unable to phosphorylate Thr210. Phosphorylation of PLK1 by Aurora B in complex with its activating cofactor INCENP (Bishop and Schumacher, 2002; Segura-Pena et al, 2023; Sessa et al, 2005) was also below the detection limit (Fig. 1F). Note that the anti-pThr210 antibody cross-reacts with the auto-phosphorylated activation segment of Aurora B (as indicated by an asterisk in Fig. 1F) but not with the one of Aurora A, which was nonetheless phosphorylated in all cases (Fig. 1D). Thus, only Aurora A:Bora can efficiently phosphorylate PLK1 in this assay.

Unlike Thr210 of PLK1, Ser55 of Ndc80 was efficiently phosphorylated by all the Aurora:cofactor complexes in the same time frame, including by Aurora A:Bora, while Aurora A:CEP192 was the least efficient kinase on this target (Fig. 1G). Histone H3 was also phosphorylated by all kinases except Aurora A:CEP192 (Fig. 1H). CENP-A, on the other hand, was preferentially phosphorylated by Aurora B:INCENP and was poorly phosphorylated by Aurora A and its variants (Fig. 1I). Taken as a whole, these experiments indicate that the Aurora A: CEP192 is the least active of the tested Aurora complexes, at least towards the limited number of substrates in our set, in line with a recent report (Holder et al, 2024). These results also indicate that the activity of the Aurora A:Bora complex, at least in vitro, is not limited to PLK1, as the complex can target additional substrates. Conversely, by identifying Aurora A:Bora as a highly selective activator of PLK1, our experiments suggest that further work in vitro may unveil the mechanistic basis of this phenomenon.

### Bora promotes transient binding to the PLK1 kinase domain

To investigate how Bora directs Aurora A to the PLK1 activation loop, we hypothesized that Bora may directly bind PLK1, as suggested by previous observations that both the PBD and kinase domain of PLK1 bind Bora in cells (Seki et al, 2008b). While the dispensability of the PBD for PLK1 activation has been demonstrated (Tavernier et al, 2021; Thomas et al, 2016), the requirement of Bora binding to the PLK1 kinase domain remains unclear. Attempts to reconstitute a stable complex using Bora, Aurora A, and PLK1 in pull-down and nuclear magnetic resonance binding assays did not yield a stable complex (Tavernier et al, 2021), a finding we independently confirmed (AEV and AM, unpublished results). Nevertheless, we reasoned that in the rapid phospho-transfer reaction, even a transient interaction may contribute to target selection. We therefore queried AlphaFold (AF) Multimer (preprint: Evans et al, 2022) to identify possible interactions of the Aurora A:Bora complex with PLK1. In a highly reproducible prediction of the ternary complex, AF identified a binding interface built up by a segment encompassing residues 55–69 of Bora and a region encompassing the N-terminal extension, the αB-αC helices, and the β4 strand of PLK1, all in the small lobe (Figs. 2A,B and EV2A–C). Residues 55–69 of Bora are positioned between the two TPX2-like M1 and M2 motifs of Bora, and their

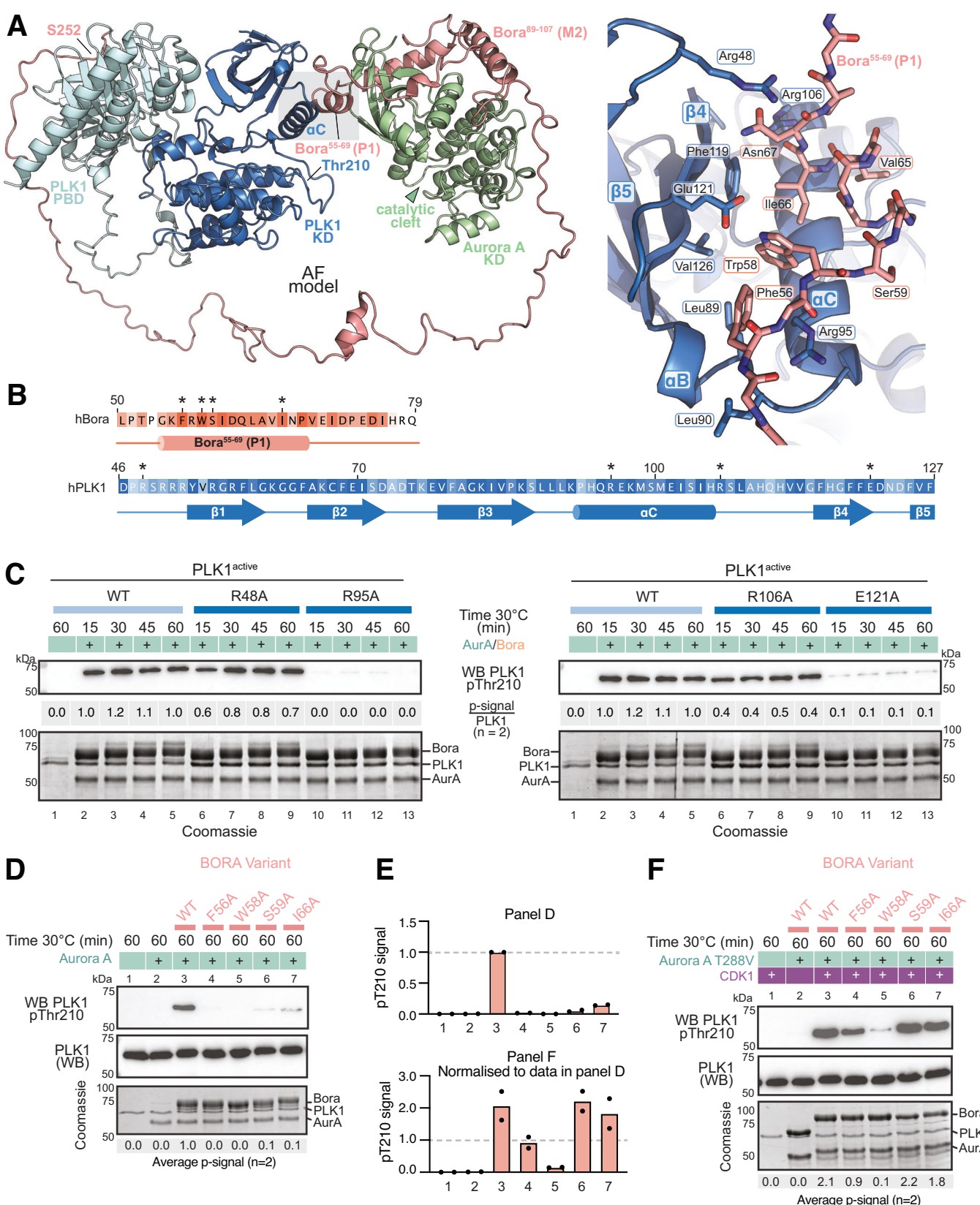

◄ **Figure 2. PLK1-Bora interface drives T-loop phosphorylation catalysis.**

(**A**) Cartoon representation of the Aurora A:Bora$_{18-280}$: PLK1 AlphaFold2 model. PLK1 Kinase Domain (KD) = dark blue; PLK1 Polo-Box Domain (PBD) = light blue; Aurora A kinase domain (KD) = green; Bora = coral red. The close-up on the right shows the residues mainly involved in the interaction between the PLK1 kinase domain and Bora. (**B**) Sequence conservation of the Bora "P1" element and of PLK1 region of interaction with Bora, across species. Residues selected for point mutagenesis are marked with asterisks. White coloring represents the minimum level of sequence conservation; dark orange (Bora) and dark blue (PLK1) represent the highest. The complete version of the alignment of the sequences from all species included in our analysis is shown in Fig. EV2D,E. (**C**) T-loop phosphorylation of PLK1 variants in vitro was checked by Western blotting after 15, 30, 45, 60 min at 30 °C, in the presence of the Aurora A:Bora complex ($n = 2$). (**D**) T-loop phosphorylation of PLK1 by Aurora A in the presence of WT or the indicated Bora variants was checked by Western blotting after 60 min at 30 °C ($n = 2$). (**E**) Quantification of data in (**D**, **F**). The arithmetic mean of two independent experiments is reported. (**F**) The same Bora variants as in (**D**) were phosphorylated with CDK1-phosphorylated Bora for 60 min at 30 °C and then added to the reaction mixture. Phosphorylation of PLK1 was monitored by western blotting ($n = 2$). Source data are available online for this figure.

sequence is conserved across species (Figs. 1C, 2B and EV2D). We refer to this region as the PLK1-binding motif of Bora (abbreviated as P1 motif). The conformation of this segment is largely identical in the predicted binary (Aurora A:Bora) and ternary (Aurora A:Bora:PLK1) complexes, stabilized by extensive interactions with the Aurora A small lobe. In the prediction of the ternary complex, the Aurora A catalytic cleft is positioned in front of the PLK1 activation loop, poised for phosphotransfer. The ternary complex as depicted in Fig. 2A reveals an additional interaction whereby the PLK1 PBD docks on Ser252 of Bora. However, the interaction of the Bora P1 motif with PLK1 was predicted even with a fragment of Bora lacking the PBD docking site, as well as in the absence of Aurora A. Thus, collectively, the predictions suggest that a favorable reciprocal configuration of kinase domains in an Aurora A:Bora:PLK1 ternary complex, orchestrated by the P1 motif of Bora, may direct Aurora A to the PLK1 activation loop.

To validate this model, we generated single amino acid substitutions to alanine of highly conserved exposed residues of PLK1 predicted by AF to interact with the P1 motif, and including Arg48, Arg95, Arg106, and Glu121 (Figs. 2A,B and EV2E). None of the mutations appeared to destabilize PLK1, as demonstrated by essentially identical catalytic activity on a C-terminal truncation of SA2, a bona fide substrate of PLK1 (Sumara et al, 2002) (Fig. EV2F). We then tested the effects of the mutations on PLK1 Thr210 phosphorylation. In a time course assay, replacement of Arg48 and Arg106 with alanine displayed a very moderate reduction in the levels of Thr210 phosphorylation, only visible at the earliest timepoint of 15'. On the contrary, mutations of Arg95 and Glu121 led to a highly potent inhibition of Thr210 phosphorylation (Fig. 2C). Next, we introduced alanine mutations in the Bora P1 motif, and specifically at residues Phe56, Trp58, Ser59, and Ile66, predicted to be exposed and in contact with PLK1 in our AF model. Ser59 is part of a predicted Aurora kinase motif, and has been recently shown to be phosphorylated by PKA and to be important for Bora-mediated activation of PLK1 (Zhu et al, 2025). All the amino acid substitutions tested produced a deleterious effect on PLK1 Thr210 phosphorylation in single-timepoint assays, with mutations of Phe56 and Trp58 appearing most penetrant, with almost no residual Thr210 phosphorylation (Fig. 2D, quantified in Fig. 2E).

CDK phosphorylation on Bora's Ser112 reinforces binding to Aurora A, enhancing its activity towards PLK1 Thr210 (Tavernier et al, 2021). Furthermore, Bora phosphorylation activates Aurora A towards PLK1 Thr210 in the absence of Aurora A activation loop phosphorylation at Thr288, even when this residue is mutated to valine (T288V) to prevent phosphorylation (Tavernier et al, 2021). Building on these previous results, we also observed that

unphosphorylated Bora, while capable of activating phosphorylated Aurora A (Fig. EV1E, lane 10), was unable to activate dephosphorylated Aurora A or Aurora A T288V (Fig. EV1E, lanes 11–12). Conversely, CDK1-phosphorylated Bora activated all three Aurora A variants towards PLK1 T210 (Fig. EV1F, lanes 10–12). Furthermore, dephosphorylated Aurora A re-accumulated phosphorylation on T288 in the presence of CDK1-phosphorylated Bora, but failed to do so in the presence of unphosphorylated Bora (Fig. EV1E,F, lanes 5, 8, and 11). An implication of these experiments in vitro is that CDK1-phosphorylated Bora enhances auto-phosphorylation of Aurora A on T288. Whether these results also apply to the cellular environment is a question for future investigations.

Moving forward, we asked if the deleterious effects on PLK1 phosphorylation from the mutations in the Bora P1 motif could be bypassed by pre-phosphorylation of Bora with CDK1, a condition expected to mimic cellular events near the G2-M transition. In the absence of CDK activity, Bora failed to activate Aurora A$^{T288V}$ to promote PLK1 Thr210 phosphorylation. Addition of CDK1 resulted in robust PLK1 phosphorylation (Fig. 2F, quantified in Fig. 2E; see also Fig. EV1F, lane 12). Addition of CDK1 also resulted in an apparent rescue of the negative effects of alanine mutations of Ser59 and Ile66, but the mutations of Phe56 and Trp58 remained very penetrant even under these conditions (Fig. 2E,F). The same mutations did not affect the ability of the Aurora A:Bora complex to phosphorylate NDC80 on Ser55, while H3 Ser10 phosphorylation appeared very slightly affected in the same reaction conditions (Fig. EV3A). Even when combined, the mutations did not affect the interaction of Bora with Aurora A, as shown by analytical SEC binding assay (Fig. EV3B). Collectively, these results validate the hypothesis, supported by a high-confidence AF model, that Bora and PLK1 physically interact to allow Aurora A to phosphorylate the PLK1 activation loop.

## Preventing binding to Bora affects PLK1 activation in cells

Bora reaches its expression peak during G2 and is then degraded (Chan et al, 2008; Seki et al, 2008a), but a small residual pool of Bora is retained in mitosis (Bruinsma et al, 2014; Feine et al, 2014). Correspondingly, mitotic cells depleted of Bora or acutely treated with an Aurora A inhibitor were shown to possess decreased levels of phospho-PLK1 (Bruinsma et al, 2014). To evaluate the cellular consequences of expressing Bora mutants that are defective in PLK1 activation, we generated stable doxycycline-inducible HeLa cell lines to express mNeonGreen (mNG)-tagged versions of the most penetrant PLK1 and Bora mutated variants, including PLK1$^{R95A}$, PLK1$^{E121A}$, Bora$^{F56A}$, and Bora$^{W58A}$. After inducing the

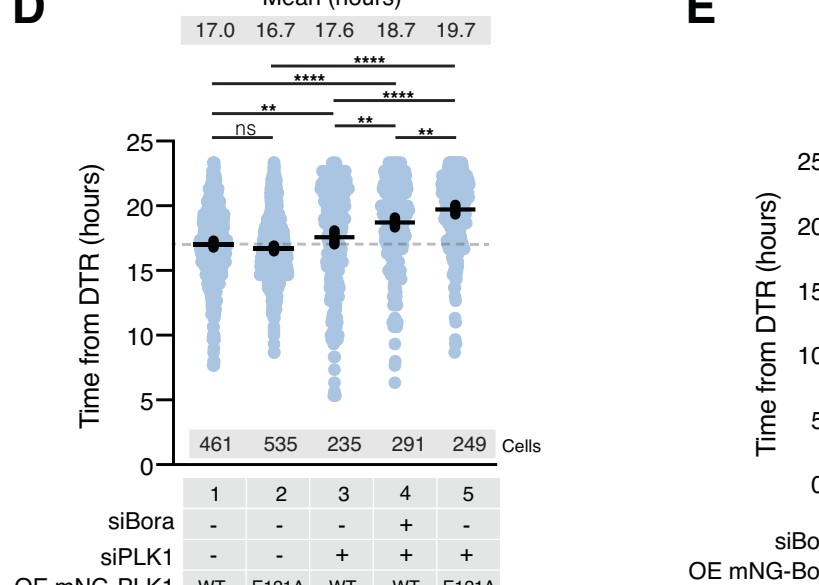

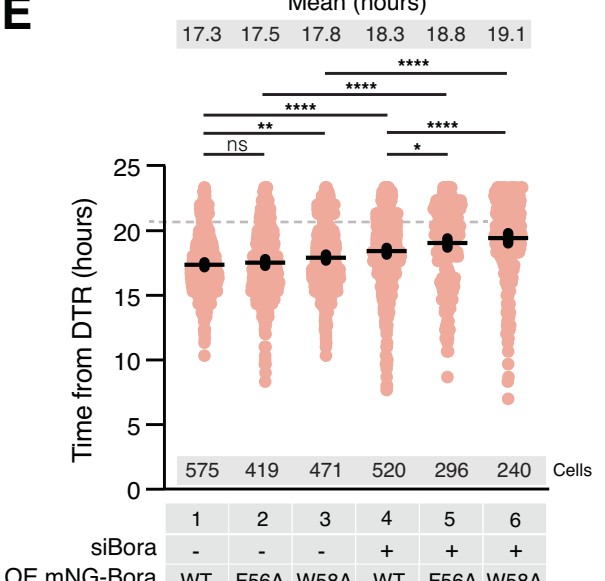

**Figure 3.   PLK1-Bora interface mutants decrease PLK1 phosphorylation in human cells.**

(A) mNeonGreen-PLK1^WT, PLK1^R95A, or PLK1^E121A were purified via pull-down from mitotic HeLa cells inducibly expressing the PLK1 variants. The bar plot represents the arithmetic mean of three replicates, and black dots are the normalized values (over PLK1 signal) of signal intensity for each replicate. The error bar is the standard deviation. OE overexpressed. (B) Pull-down from mitotic HeLa cell lysates expressing mCherry-PLK1^WT, depleted of Bora, and transfected to express mNeonGreen-Bora^WT, Bora^F56A, or Bora^W58A. The bar plot represents the arithmetic mean of four replicates, and black dots are the normalized values (over PLK1 signal) of signal intensity for each replicate. The error bar is the standard deviation. OE overexpressed, siBora siRNA-mediated depletion of endogenous Bora. (C) Scheme of the experiment presented in (D). (D, E) HeLa cell lines expressing mNeonGreen-PLK1 (wild-type or mutants), mNeonGreen-Bora (wild-type or mutants) were synchronized via double-thymidine block and released to follow their entry into mitosis. Dot plots show the distributions of mitotic entry time of individual cells since double-thymidine release (DTR). Each dot represents an individual cell; the horizontal line represents the mean, and the vertical line the 95% confidence interval (CI) of the mean. Statistical comparison was performed in Prism 10. All cell biology experiments were repeated at least three times to attain statistical significance. We used the Kruskal–Wallis test coupled with Dunn's multiple comparisons for experiments comparing 3 or more conditions. To convert $P$ values into an asterisk-based significance system, we used the default GraphPad Prism convention: not significant (ns) $P > 0.05$; *$P \leq 0.05$ but $>0.01$; **$P \leq 0.01$ but $> 0.001$; ***$P \leq 0.001$ but $>0.0001$; and ****$P \leq 0.0001$. OE overexpressed, siBora siRNA-mediated depletion of endogenous Bora. The experiments were interrupted after 24 h of filming due to the significant effects on cell viability caused by phototoxicity after this time point. The exact $P$ values for pairs of conditions in (D) were: 1–2 = 0.1612; 1–3 = 0.0060; 1–4 < 0.0001; 2–5 < 0.0001; 3–4 = 0.0097; 3–5 < 0.0001; 4–5 = 0.0031; and for (E) 1–2 > 0.9999; 1–3 = 0.0077; 1–4 < 0.0001; 2–5 < 0.0001; 3–6 < 0.0001; 4–5 = 0.0390; 4–6 < 0.0001. Source data are available online for this figure.

expression of the relevant transgenes (Fig. EV3D), we performed pull-downs from mitotically arrested cells to enrich PLK1 and monitor pT210 phosphorylation by Western blotting (pre-purification of PLK1 via pull-down was necessary as pT210 detection on the whole cell lysate was suboptimal and did not allow reproducible quantifications). Both PLK1^R95A and PLK1^E121A caused a reduced level of T210 phosphorylation, with E121A having a stronger effect than R95A (Fig. 3A). To assess the function of Bora mutants, endogenous Bora was depleted and replaced by re-expression of a transgenic construct. Bora depletion resulted in a marked reduction of PLK1 phosphorylation in mitotic cells. While seemingly in contrast with an early report (Seki et al, 2008b), this observation is instead consistent with more recent work (Bruinsma et al, 2014). Expression of Bora^WT rescued the reduced phosphorylation of T210 of PLK1, while expression of Bora^F56A or Bora^W58A did not promote any phosphorylation of T210 (Figs. 3B and EV3E).

The effects of Bora depletion have been studied mainly in the context of early mitotic functions, where a delay in mitotic entry as well as spindle alterations, were observed. At least the spindle alterations, however, were less dramatic than those observed upon acute small-molecule inhibition of PLK1 activity (Abrieu et al, 1998; Bruinsma et al, 2014; Chan et al, 2008; Gheghiani et al, 2017; Macurek et al, 2008; Seki et al, 2008a; Seki et al, 2008b; van Vugt et al, 2004). Based on these previous results, we decided to assess the effects of inhibiting the Bora:PLK1 interaction on the timing of mitotic entry. Toward this goal, we depleted endogenous PLK1, or co-depleted endogenous PLK1 and Bora, and re-expressed mNG-PLK1^E121A or mNG-PLK1^WT. To monitor mitotic entry, we synchronized cells at the G1/S transition with a double-thymidine block, released them from the arrest, and followed their progression into G2 and M phase using time-lapse fluorescence microscopy (Fig. 3C).

Overexpression of mNG-PLK1^WT rescued, at least partly, the delay caused by depletion of endogenous PLK1 (Fig. 3D). By co-depleting Bora, overexpressed mNG-PLK1^WT was not properly activated, as testified by the further, albeit modest, increase in the mitotic entry delay. Depletion of endogenous PLK1 combined with overexpression of mNG-PLK1^E121A produced a mitotic entry delay similar or even more severe than the one observed after Bora and PLK1 co-depletion (Figs. 3D and EV3F). Expression of the Bora mutants led to a similar trend: cells expressing Bora^WT were more

proficient than Bora^F56A or Bora^W58A in rescuing the mitotic delay caused by Bora depletion (Figs. 3E and EV3G). As the live-cell measurements in Fig. 3D,E were arrested after 24 h due to increasing phototoxicity, we surmise that the effects of the PLK1 and Bora mutants may be underestimated.

## A gatekeeper residue for Bora selectivity in the PLK1 T-loop

Substrate selectivity of Aurora kinase family members has been extensively characterized. The minimal consensus sequence consists of positively charged residues upstream of the phospho-acceptor site and a hydrophobic residue immediately downstream of it (R/K/N-R/K-X-**S/T**-Φ, where X stands for any amino acid and Φ for hydrophobic amino acids except a proline; the phosphorylated residue is shown in bold) (Alexander et al, 2011; Ferrari et al, 2005; Kettenbach et al, 2011; Koch et al, 2011). This consensus sequence, also present in the activation loops of Aurora A and B, is the basis of their auto-activation mechanism (Fig. 1A). Despite its similarity to the activation loop of Aurora kinases, the PLK1 activation loop carries lysine rather than arginine at the -2 residue (position 208 in the human enzyme; Fig. 1A). This substitution, albeit conservative, is a largely evolutionary conserved feature of the PLK1 activation loop (Fig. EV4A). Presence of a -2 lysine does not universally prevent efficient Aurora phosphorylation of substrates, but arginine is preferred over lysine at this position (Alexander et al, 2011; Dephoure et al, 2008; Ferrari et al, 2005; Hegemann et al, 2011; Kettenbach et al, 2011; Koch et al, 2011; Nousiainen et al, 2006). We therefore hypothesized that this substitution may limit the ability of Aurora kinases to phosphorylate the PLK1 activation loop, and that the limitation might be overcome by the direct interaction of the Aurora A:Bora complex with PLK1.

To test this hypothesis, we replaced Lys208 with arginine (PLK1^K208R) and asked if this amino acid substitution facilitated phosphorylation of PLK1 Thr210 by additional Aurora kinase complexes. In an in vitro kinase assay using SA2 as a substrate, PLK1^K208R appeared as active as PLK1^WT, indicating that the mutation does not cause overt structural or functional perturbations (Fig. EV4B). Again, we compared the ability of Aurora kinases to phosphorylate PLK1 in the presence or absence of the K208R mutation. In agreement with our hypothesis, the PLK1^K208R mutant

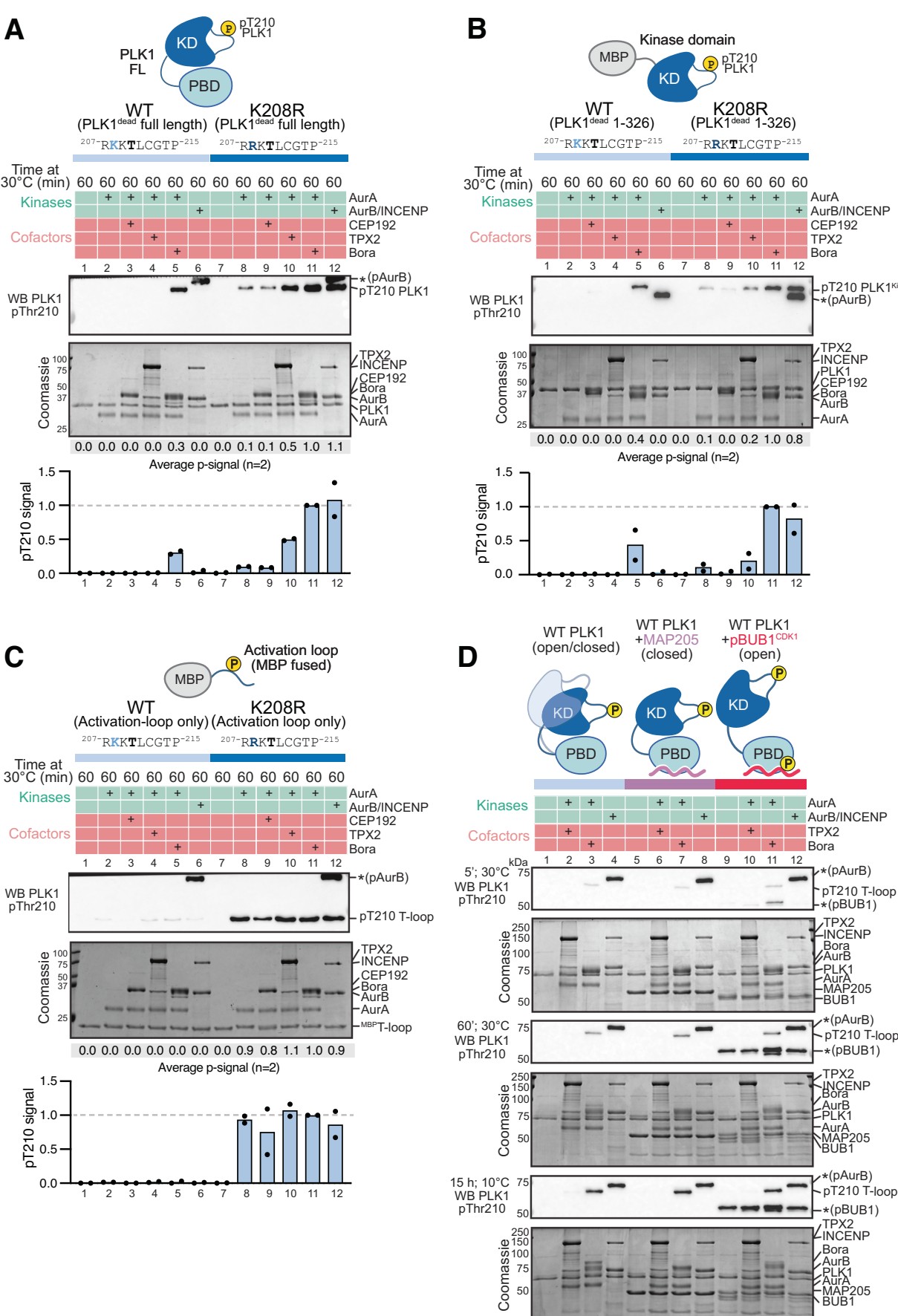

**Figure 4.  The primary sequence of PLK1 T-loop renders phosphorylation contingent on Bora.**

(A) PLK1^WT or PLK1^K208R were assayed in parallel for activation with the same Aurora complexes. A western blot is shown to assess relative phosphorylation of the substrates, quantified with Aurora A:Bora activity on PLK1^K208R as reference (equal to 1.0). The arithmetic mean of two independent experiments is shown, and each black dot represents the signal intensity for a single experiment. A Coomassie Brilliant Blue-stained gel of the same samples was combined to verify equal protein loading between conditions. (B) An MBP fused PLK1 kinase domain WT or K208R (1–326) was assayed under the same conditions as for PLK1 in (A). (C) An MBP fused peptide containing PLK1 T-loop sequence WT or K208R was assayed under the same conditions as for PLK1 in (A). (D) PLK1 WT was assayed for activation in the presence or absence of MAP205 and BUB1 PBD docking motifs. Samples were collected during the phosphorylation reaction at 5 and 60 min. Another set of samples was incubated for 15 h at 10 °C, to saturate the reaction. Source data are available online for this figure.

allowed every Aurora species we tested to phosphorylate PLK1 Thr210, with the Aurora A:TPX2 and Aurora B:INCENP complexes being apparently as active on Thr210 as Aurora A:Bora, in sharp contrast with their complete inability to phosphorylate wild-type PLK1 within the reaction time (Fig. 4A; a schematic description of this experiment is presented in Fig. EV4C). Mutations of two additional residues neighboring Lys208 – i.e., Glu206 to leucine (the corresponding residue in the Aurora B activation loop) and Arg207 to lysine, did not recapitulate this effect. PLK1^E206L was phosphorylated essentially indistinguishably from PLK1^WT, showing the same selectivity for the Aurora A:Bora complex (Fig. EV4E, lanes 1–6). PLK1^R207K, on the other hand, was also phosphorylated selectively, but to lower levels, in line with the general importance of the -3 arginine residue (Fig. EV4E, lanes 7–12). So, neither mutation made PLK1 accessible to other Aurora kinases, a property that, among the mutants we tested, was only conferred by the Lys208 to arginine mutation. Using orthogonal detection methods to visualize T210 phosphorylation of PLK1^WT and PLK1^K208R, or their isolated activation segments, we also concluded that the signal increase observed with PLK1^K208R does not reflect increased reactivity of the antibody in the presence of the K208R mutation (Fig. EV5A,B).

To further probe the specificity of this effect, we increased the concentration of PLK1 substrate ~12-fold (to a concentration of 2.5 µM) and tested Thr210 phosphorylation with increasing concentration of Aurora A:Bora or Aurora B:INCENP, up to an equimolar concentration with the substrate. Wild-type PLK1 continued to remain entirely impervious to phosphorylation by the Aurora B:INCENP complex, even under these extremely facilitating conditions, while it was readily phosphorylated by Aurora A:Bora, as anticipated. In contrast, PLK1^K208R was phosphorylated at all concentrations of Aurora B:INCENP and with apparently indistinguishable rates in comparison to Aurora A:Bora, likely reflecting saturation over the entire range of conditions (Fig. EV4F). Thus, Lys208 of PLK1 is a crucial gatekeeper deterring Aurora kinase:cofactor variants other than the Aurora A:Bora complex from phosphorylating PLK1 Thr210.

## The PLK1 activation loop is an intrinsically poor Aurora substrate

The "closed", autoinhibited conformation of PLK1 is stabilized by the interaction of the PBD and kinase domains. It has been proposed that in this closed conformation, the activation segment of PLK1 may be inaccessible to phosphorylation, and that the crucial step for the Aurora A:Bora complex to be granted access to PLK1 Thr210 is the release of the closed conformation upon engagement of the PBD on phosphorylated Bora (Seki et al, 2008b).

Our new observations now suggest that the amino acid sequence of the activation segment is also a crucial determinant of Aurora kinase selectivity. To investigate whether the sequence of the activation segment plays a role independently of its accessibility, we truncated PLK1 at the beginning of the interdomain linker (aa 326), a few residues downstream from the C-terminal end of the kinase domain. Confirming our results that the selectivity for Aurora A:Bora is attributable to features largely localized to the kinase domain, the phosphorylation patterns of PLK1$_{1\text{-}326}$^WT and of PLK1$_{1\text{-}326}$^K208R remained unchanged, with Aurora B gaining access to the mutant but not to the wild-type (Fig. 4B).

These results suggest that the PLK1 activation loop is intrinsically a better substrate for the Aurora A:Bora complex. However, it is also possible that the mutation of Lys208 to arginine affects the overall exposure of the PLK1 activation loop, improving its accessibility of the PLK1 activation loop to Aurora kinases other than Aurora A:Bora. We therefore asked if selective phosphorylation of Thr210 by the various Aurora species was also retained when the activation loop was fully exposed. For this, we fused a 15-residue peptide encompassing Thr210 and the seven residues downstream and upstream to His-MBP, and subjected it to in vitro phosphorylation. The T-loop peptide with wild-type sequence was poorly phosphorylated by any Aurora species, including the Aurora A:Bora complex (Fig. 4C). On the contrary, an equivalent construct carrying the K208R mutation was efficiently phosphorylated by all Aurora complexes, recapitulating results obtained with the full-length PLK1^K208R mutant (Fig. 4C). These observations show that the sequence of the PLK1 activation loop is an intrinsically poor Aurora substrate, and that its incorporation in the context of the kinase domain, and possibly also of other PLK1 features, builds the selectivity for Aurora A:Bora.

## The PLK1 activation loop in closed-PLK1 is accessible to Aurora A:Bora

In principle, a requirement for a transition from a closed to an open conformation as a condition for PLK1 Thr210 phosphorylation is not mutually exclusive with the observed selectivity of Thr210 phosphorylation on Aurora A:Bora. For instance, the Aurora A:Bora complex may perform both functions, first "opening" PLK1 and then accessing the activation loop. A counterintuitive aspect of this model, however, is that it would imply that Aurora B:INCENP, which turned into an excellent PLK1 Thr210 kinase after mutating Lys208 to arginine, can also "open" PLK1 as a precondition for its phosphorylation.

To further investigate this issue, we focused on the coupling between conformational changes of PLK1 and its activation. For this, we measured Aurora A:Bora accessibility to the activation loop

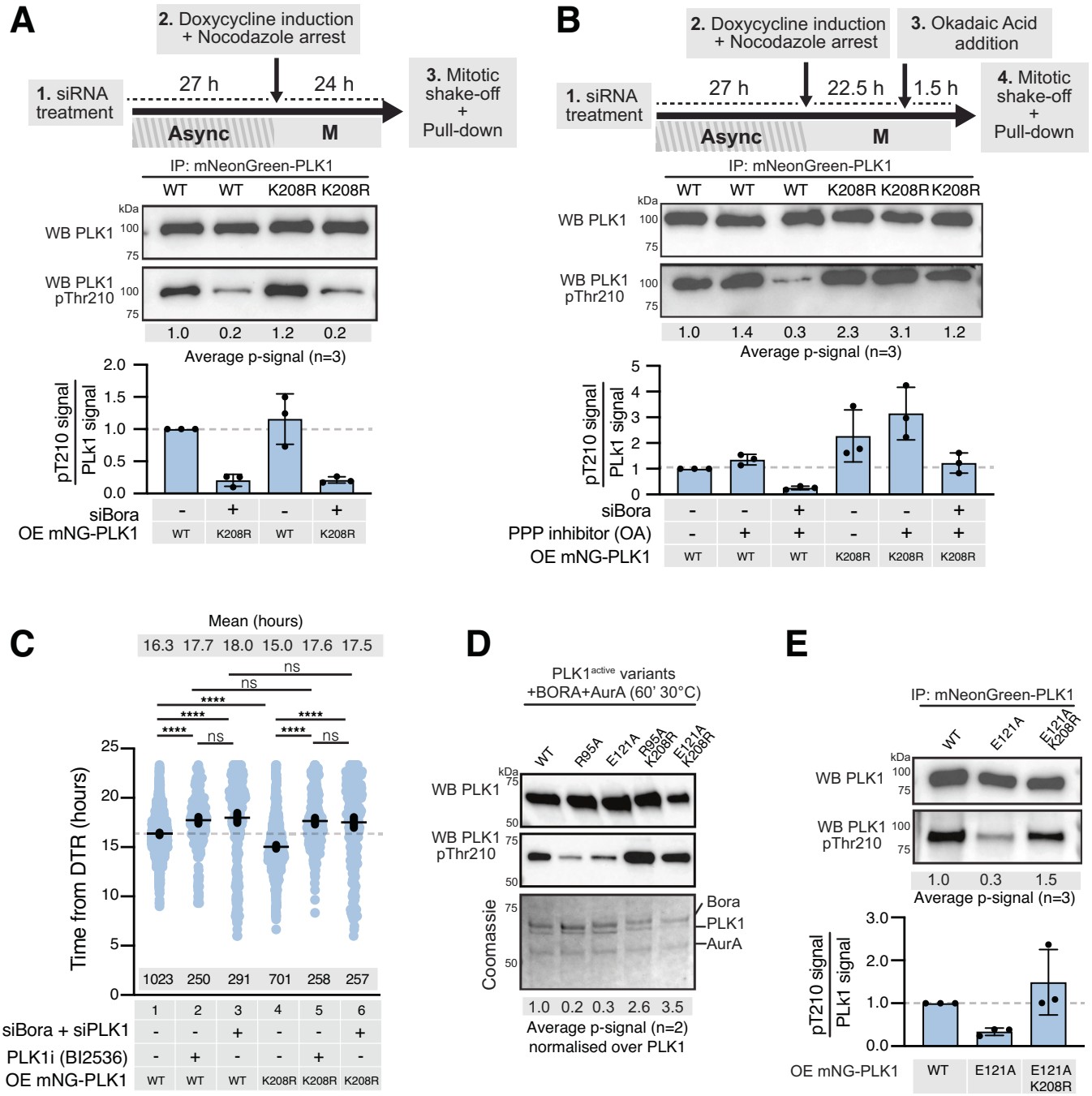

of wild-type PLK1 under two conditions known to modify the reciprocal conformation of the PBD and kinase domains. First, we combined PLK1 with a peptide from *Drosophila melanogaster* MAP205 that binds the PBD to lock PLK1 in an inactive, closed conformation, a condition predicted to restrict access to the PLK1 activation loop (Kachaner et al, 2014; Xu et al, 2013). Second, we combined PLK1 with a BUB1 peptide pre-phosphorylated with CDK1 to generate a high-affinity binding site for the PBD. The phospho-dependent interaction of BUB1 is expected to release the auto-inhibitory interaction between PLK1 PBD and kinase domain

(Kachaner et al, 2014; preprint: Ren et al, 2025; Xu et al, 2013). As a control, we used PLK1 in the absence of PBD ligands, assuming that it might thermally oscillate between open and closed states. Both MAP205 and pBUB1 are able to bind to PLK1 in an analytical SEC binding assay (Fig. EV6A).

These experiments revealed that presence of the MAP205 or BUB1 peptides did not have significant effects on PLK1 Thr210 phosphorylation, with levels being indistinguishable also in comparison to the sample without peptide (Fig. 4D). Furthermore, the presence of PBD-binding peptides did not increase the

◄ **Figure 5. PLK1 K208R mutant remains dependent on Bora for T-loop phosphorylation.**

(A) Pull-down of mNeonGreen-PLK1$^{WT}$ or mNeonGreen-PLK1$^{K208R}$ from mitotic HeLa cell lysates with or without RNAi-mediated Bora depletion, as schematized. The bar plot represents the arithmetic mean of three replicates, and black dots are the normalized values (over PLK1 signal) of signal intensity for each replicate. OE overexpressed; siBora siRNA-mediated depletion of Bora. The bar plot represents the arithmetic mean of three replicates, and black dots are the normalized values (over PLK1 signal) of signal intensity for each replicate. The error bar is the standard deviation. (B) Pull-down performed as in (A), except for the addition of Okadaic acid (100 nM OA, 1 h before cell harvesting), with or without Bora depletion, as schematized. The bar plot represents the arithmetic mean of three replicates, and black dots are the normalized values (over PLK1 signal) of signal intensity for each replicate. OE overexpressed, siBora siRNA-mediated depletion of endogenous Bora, PPPi phosphoprotein phosphatase inhibitor. The bar plot represents the arithmetic mean of three replicates, and black dots are the normalized values (over PLK1 signal) of signal intensity for each replicate. The error bar is the standard deviation. (C) HeLa cell lines expressing mNeonGreen-PLK1$^{WT}$ or mNeonGreen-PLK1$^{K208R}$ were synchronized with a double-thymidine arrest, released in the cell cycle, and followed as they entered mitosis. Dot plots show the distribution of mitotic entry times of individual cells after double-thymidine release (DTR). Each dot represents an individual cell, the horizontal line represents the mean, and the vertical line the 95% CI of the mean. Statistical comparison was performed as indicated in the legend for Fig. 3D,E. OE overexpressed, siBora/siPLK1 siRNA-mediated depletion of endogenous Bora and endogenous PLK1, PLK1i Polo-like kinase 1 inhibitor (BI2536). The experiments were interrupted after 24 h of filming due to the significant effects on cell viability caused by phototoxicity after this time point. The exact *P* values were: 1–2 < 0.0001; 1–3 < 0.0001; 1–4 < 0.0001; 2–3 > 0.9999; 2–5 > 0.9999; 3–6 = 0.6884; 4–5 < 0.0001; 4–6 < 0.0001; 5–6 > 0.9999. (D) In vitro assay with Aurora A:Bora and PLK1$^{WT}$, PLK1$^{R95A}$, PLK1$^{E121A}$, PLK1$^{R95A-K208R}$, and PLK1$^{E121A-K208R}$. After 60 min at 30 °C, activation loop phosphorylation of PLK1 was monitored. (E) Pull-down from mitotic cell lysates to assess the extent of T-loop phosphorylation of mNeonGreen-PLK1$^{WT}$, mNeonGreen-PLK1$^{E121A}$, and mNeonGreen-PLK1$^{E121A-K208R}$. The experimental regime is as in Fig. 3A. The bar plot represents the arithmetic mean of three replicates, and black dots are the normalized values of signal intensity for each replicate. The error bar is the standard deviation. OE overexpressed. Source data are available online for this figure.

accessibility of Thr210 to the non-cognate kinases Aurora A:TPX2 and Aurora B:INCENP (Fig. 4D). Collectively, our results demonstrate that the activation segment of PLK1 is, at least in first approximation, equally accessible when PLK1 is locked in a closed conformation or open by an activator, in line with a structural model of full-length PLK1 (preprint: Ren et al, 2025).

## PLK1 K208R mutant phosphorylation in the absence of Bora

Because the PLK1$^{K208R}$ mutant we have identified is efficiently phosphorylated by various Aurora kinase species in vitro, we asked if its activation had been made independent of Bora also in the cellular environment. For this, we overexpressed mNG-PLK1 WT or K208R, and monitored the levels of T210 phosphorylation in Nocodazole-arrested cells, depleted of Bora. Phosphorylation of PLK1 K208R was comparable to that of PLK1 WT, and depletion of Bora strongly affected phosphorylation of both protein variants (Figs. 5A and EV6B). Reduced Thr210 phosphorylation of PLK1 WT upon Bora depletion was not rescued by okadaic acid, a PP1/PP2A inhibitor, suggesting that in the adopted experimental conditions no alternative kinase phosphorylates Thr210 in the absence of Bora (Figs. 5B and EV6C). In contrast, PLK1 K208R exhibited higher overall Thr210 phosphorylation than PLK1 WT upon addition of okadaic acid, and combined Bora depletion treatment resulted in reduced but still substantial phosphorylation levels (Figs. 5B and EV6C).

HeLa cell lines stably expressing mNG-PLK1$^{WT}$ or mNG-PLK1$^{K208R}$ were synchronized at the G1/S boundary and then released into the cell cycle as in the experiments described above (e.g., Fig. 3D). While the overexpression of mNG-PLK1$^{WT}$ partly compensated PLK1 depletion, it did not compensate Bora depletion, as evidenced by a robust mitotic entry delay. Expression of mNG-PLK1$^{K208R}$ in the absence of Bora led to a mitotic entry delay comparable to that observed with the expression of mNG-PLK1$^{WT}$ (Figs. 5C and EV6D,E). Mitotic entry in control cells expressing mNG-PLK1$^{K208R}$ was significantly faster, although we could not unequivocally ascribe this effect to an enhanced Thr210 phosphorylation (Fig. 5C). Thus, even in cells expressing PLK1$^{K208R}$, there is a continued requirement for Bora. This requirement may reflect additional regulatory controls on PLK1 activation, for

instance, on co-localization of PLK1 with the T210 kinase that only Bora can satisfy.

Introduction of the K208R mutation into the Bora-binding deficient PLK1$^{R95A}$ or PLK1$^{E121A}$ mutants caused a substantial increase of pT210 levels phosphorylation in vitro, confirming the dispensability of the Bora P1 motif for PLK1$^{K208R}$ T210 phosphorylation (Fig. 5D). In line with these results, the PLK1$^{E121A-K208R}$ double mutant variant was substantially phosphorylated also upon expression in HeLa cells, contrary to the PLK1$^{E121A}$ single mutant (Figs. 5E and EV6F). While the identity of the kinase responsible for this phosphorylation remains unknown, this finding demonstrates that the K208R mutation increases the efficiency of PLK1 phosphorylation in cells.

## Resilience of PLK1 pThr210 to protein phosphatases

Bora levels decline upon mitotic entry, but Bora remains essential for PLK1 Thr210 phosphorylation also in mitosis (Bruinsma et al, 2014). Our above observations confirm that there is no mitotic kinase able to phosphorylate PLK1$^{WT}$ on Thr210 after Bora depletion. Conversely, accumulation of pThr210 on PLK1$^{K208R}$ is still dependent on Bora, but this requirement can be partly overcome if phosphatase activity is inhibited. These observations raise interest in the accessibility of the PLK1 activation segment to phosphatase activity, a question that remains poorly studied. Specifically, we were curious to know whether the maintenance of high levels of pThr210 in spite of declining levels of Bora may reflect a poor accessibility of pThr210 by phosphatase activity. Furthermore, we also wanted to investigate the possibility that the presence of the gatekeeper Lys208 may not only make the PLK1 activation segment impervious to "ordinary" Aurora kinases, but also protect it from dephosphorylation by protein phosphatases. Faster dephosphorylation of the arginine-containing motif in PLK1$^{K208R}$ may explain why it does not accumulate in cells unless phosphatase activity is temporarily blocked.

To test this hypothesis, we pre-phosphorylated PLK1 with the Aurora A:Bora complex and then subjected it to dephosphorylation reactions in vitro with three different protein phosphatases (PP) playing critical roles before, during, and after mitosis: PP1 (γ isoform), PP2A:B55 (δ isoform), and PP2A:B56 (γ isoform) (Fig. 6A). As a control, we performed the same dephosphorylation

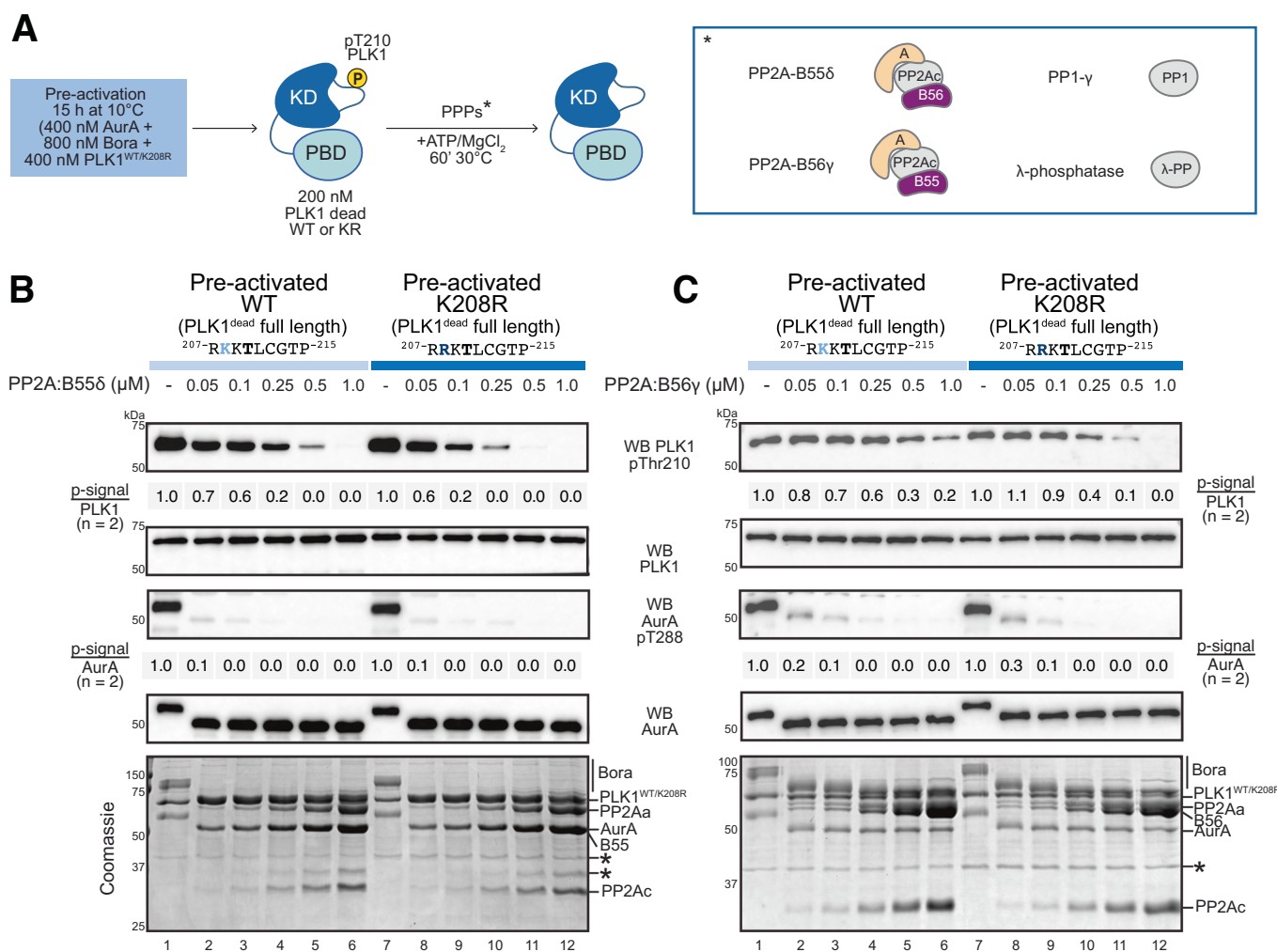

**Figure 6. Thr210 of PLK1 is a poor substrate of mitotic protein phosphatases.**

(A) Experimental scheme adopted to analyze the susceptibility of PLK1 to phosphatases. PLK1 WT or K208R was incubated overnight with Aurora A:Bora, then dephosphorylated with increasing concentrations of PP2A-B55, PP2A-B56, PP1, and lambda-phosphatase. (B) Dephosphorylation assay in the presence of PP2A-B55 from 0.05 to 1 μM. Phosphorylation of PLK1WT or PLK1K208R, and of Aurora A Thr288 was followed by Western blotting with phospho-antibodies. A Western blot of Aurora A and PLK1 and a Coomassie Blue-stained gel were included to verify equal protein loading between conditions. (C) Dephosphorylation assay in the presence of PP2A-B56 from 0.05 to 1 μM. Phosphorylation of PLK1WT or PLK1K208R, and of Aurora A Thr288 was followed by western blotting with the indicated phospho-specific antibodies. A western blot of Aurora A and PLK1, and a Coomassie Blue-stained gel were included to demonstrate equal protein loading between conditions. Source data are available online for this figure.

assay with the broadly specific bacteriophage lambda-phosphatase (Ansai et al, 1996). pThr210 of PLK1WT was remarkably resilient to dephosphorylation by PP2A-B55, and retained a substantial level of phosphorylation even when exposed at an approximately stoichiometric ratio to the action of the phosphatase for 60'. As a comparison, we monitored the dephosphorylation of pThr288 on the Aurora A activation segment, using PP2A-B55 in equimolar amount, revealing that pThr288 of Aurora A underwent rapid and complete dephosphorylation by PP2A:B55δ under all tested conditions (Fig. 6B). Similar results were obtained with PP2A:B56γ, which was only slightly less effective at dephosphorylating pThr288 of Aurora A (Fig. 6C). On the other hand, PP1γ appeared to be the least active of the three phosphatases on the two kinases tested in this assay (Fig. EV7A), while Lambda-phosphatase did not show significant differences in the ability to dephosphorylate Aurora A and PLK1 (Fig. EV7B).

These results provide an initial demonstration that the activation loop of PLK1 is comparatively highly resilient to dephosphorylation by mitotic phosphatases, which may explain why its phosphorylation can be maintained even in spite of Bora's massive degradation upon mitotic entry. In comparison, the activation segment of PLK1K208R, was slightly but reproducibly less resilient to dephosphorylation by PP2A (Figs. 6B,C and EV7), supporting the conclusion that Lys208 protects pThr210 from phosphatase activity more effectively than when this residue is replaced with arginine.

## Discussion

Activation of PLK1 requires Bora as an Aurora A cofactor, but the molecular basis for this selective activation mechanism has remained obscure. Here, we developed an assay that, by using

purified proteins, recapitulates the exquisitely selective requirement of Bora for PLK1 activation loop phosphorylation. The establishment of this assay allowed us to identify the major determinants of Aurora kinases' selectivity towards PLK1. Using PLK1 kinase dead variant (K82R), we excluded that PLK1 autoactivates in the presence of the Aurora A:Bora complex, in line with a previous study (Tavernier et al, 2021).

Demonstrating that the PLK1 T-loop consensus sequence is a suboptimal Aurora substrate, a single conservative substitution from Lysine to Arginine in position -2 from Thr210 was sufficient to promote phosphorylation by nearly all Aurora complexes tested in vitro. Despite a well-documented preference of Aurora kinases for arginine over lysine at position -2, the presence of the latter residue does not universally prevent efficient Aurora phosphorylation of substrates (Alexander et al, 2011; Dephoure et al, 2008; Ferrari et al, 2005; Hegemann et al, 2011; Kettenbach et al, 2011; Koch et al, 2011; Nousiainen et al, 2006). For instance, NDC80 Ser55, used as a control substrate (Fig. 1C), also has a lysine at position $-2$, but it is rapidly phosphorylated in the same time frame by all tested Aurora kinases. Thus, we cannot affirm that Lys208 is the sole element controlling the gatekeeping mechanism we have discovered. However, PLK1 variants bearing other amino acid substitutions at Thr210 neighboring sites (i.e., PLK1$^{E206L}$ and PLK1$^{R207K}$) displayed a very different behavior compared to PLK1$^{K208R}$. The guanidinium group of arginine is permanently protonated under physiological conditions, and its positive charge is delocalized by resonance. This gives it ideal properties for electrostatic and hydrogen-bonding interactions. In contrast, lysine's side chain is a primary amine with a localized charge that can, in principle, transiently deprotonate. We therefore surmise that the more stable and delocalized cationic nature of the arginine side chain promotes stronger electrostatic interactions with negatively charged residues or phosphorylated groups in Aurora kinase, thereby stabilizing the complex. We also considered a potential role of steric hindrance in making the PLK1 activation loop impervious to Aurora kinases in the absence of Bora. A hypothetical accessibility barrier to the PLK1 activation loop may arise from an intramolecular interaction of the PLK1 kinase domain and the PBD. It was proposed that in the resulting "closed" state of PLK1, the region spanning residues 312-360 of the interdomain linker may mask the activation loop, making it inaccessible to Aurora A and inhibiting Thr210 phosphorylation (Xu et al, 2013). We were unable to perform experiments on PLK1$^{1-312}$, the construct used by Xu and colleagues (reference (Xu et al, 2013)), as it appeared poorly stable. Nevertheless, their conclusion seems inconsistent with our findings, as we show that Thr210 is equally accessible when the intramolecular interaction between PLK1's kinase domain and the PBD is actively stabilized (by using an inhibitory MAP205 peptide) or destabilized (by adding a CDK-phosphorylated BUB1 peptide). Furthermore, a short peptide exclusively encompassing the activation loop of PLK1 and exposing a lysine residue at position 208 was a poorer substrate than the same peptide exposing an arginine, indicating that the main obstacle that Bora needs to overcome is intrinsic to the specific sequence of the PLK1's activation loop, rather than to its conformation.

Instead, our work suggests that overcoming the intrinsic barrier to phosphorylation requires the Aurora A:Bora complex to interact directly, albeit transiently, with PLK1. While we were not able to

trap the complex biochemically, due to its instability, we validated a high-confidence AF prediction by mutationally targeting the complementary interacting interfaces predicted by structural modeling. The αC helix is an extremely well-conserved feature in the small lobe of protein kinases (Bayliss et al, 2012). It contributes to the folding of the small lobe and positions residues that stabilize ATP in the active site, promoting catalysis (Palmieri and Rastelli, 2013). Unlike the standard regulation mechanisms involving the spatial orientation of the αC helix by post-translational modifications or cofactor binding (Bayliss et al, 2012; Huse and Kuriyan, 2002), the PLK1 αC helix and neighboring residues dynamically bridge the interaction with a short N-terminal α-helix of Bora (the P1 element). Mutations in these regions of Bora and PLK1 impaired Thr210 phosphorylation. Particularly, the predicted hydrogen bond between PLK1 Glu121 and Bora Trp58 may be essential for efficient catalysis. We conclude that Bora bypasses the unfavorable phosphorylation reaction of the PLK1 activation-loop by forming a physical connection between Aurora A and the PLK1 kinase domain. We speculate, therefore, that the main role of Bora is to prolong the residency of the PLK1 activation loop in the proximity of the Aurora A active site, ultimately overcoming a kinetic barrier to phosphorylation introduced by the peculiar sequence of the PLK1's activation loop, which deviates from the ideal Aurora consensus.

The effects on cell cycle progression from depleting Bora in different model systems range from a modest to a complete blockade to mitotic entry (Macurek et al, 2008; Seki et al, 2008b; Tavernier et al, 2015; Tavernier et al, 2021; Vigneron et al, 2018). In cultured human cells, the main consequence is a delayed mitotic entry, but the effect is strongly exacerbated if cells are forced to recover from a temporary cell cycle arrest caused by DNA damage (Macurek et al, 2008). We mimicked this effect with mutants preventing the Bora:PLK1 interaction. In studies that developed independently in parallel with ours, the mechanistic basis of Bora activation of PLK1 phosphorylation were also addressed, with conclusions that are perfectly in line with ours (Miles et al, 2026; Pillan et al, 2026). Using model systems that emphasize the role of Bora for mitotic entry in comparison to human cells, their study also provides a compelling biological demonstration that the interaction of Bora and PLK1 is essential for PLK1 activation and function in Xenopus egg extracts (Pillan et al, 2026).

Contrary to the mutations at the Bora:PLK1 interface, substitution of Lys208 to arginine showed a more nuanced situation than the one observed in vitro. Thr210 phosphorylation of the PLK1$^{K208R}$ variant remained dependent on Bora. Only upon phosphatase inhibition, we observed Bora-independent PLK1 Thr210 phosphorylation of the mutant motif, but not of its wild-type counterpart. The reason why the activation loop motif of PLK1 remains impervious to Aurora kinases even after introduction of the conservative arginine mutation at the $-2$ position will need further investigation. It may reflect the low cellular concentrations of PLK1 and its activating kinase in vivo, which may justify a decreased but continuous requirement for Bora. A non-mutually exclusive additional explanation is that the lysine-to-arginine conservative substitution at position 208 makes pThr210 a better phosphatase substrate. In line with this idea, we observed a small but highly reproducible acceleration of pThr210 dephosphorylation of the activation loop of PLK1$^{K208R}$ relative to that in PLK1$^{WT}$. This observation could at least partially explain the results obtained on

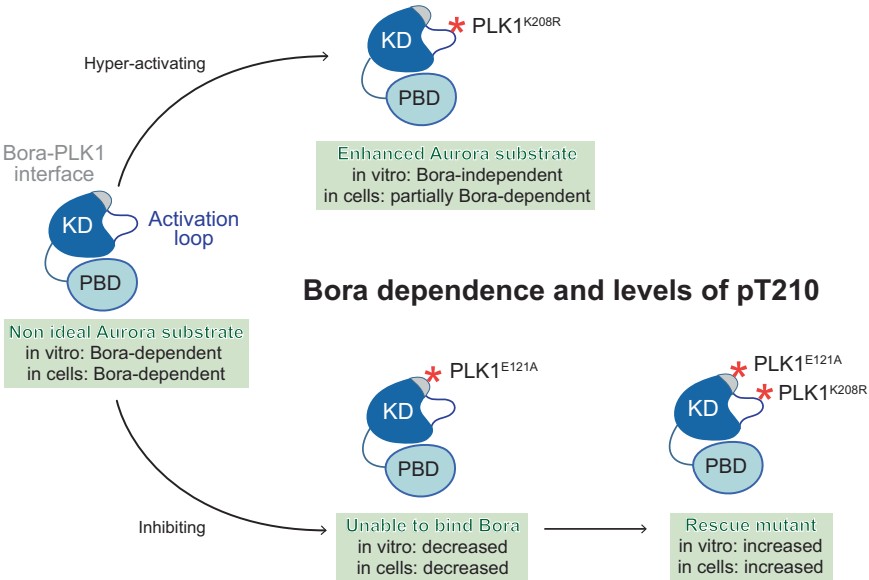

**Figure 7.  Mutations of PLK1 used in the study and their effect on T-loop phosphorylation.**

Scheme summarizing the dependency of activation of PLK1 from Aurora A:Bora upon mutation of either the Bora:PLK1 interface or the activation-loop region, with a distinction on how those two manipulations altered activation-loop phosphorylation in cells or with recombinant proteins.

T-loop phosphorylation in the presence of phosphatase inhibitors, where PLK1 K208R accumulates more phosphorylation both in the presence and in the absence of Bora.

In addition, dephosphorylation of PLK1 proceeds at a much slower rate than dephosphorylation of the Aurora A T-loop on Thr288. PP2A activity is reduced by the presence of a double lysine upstream to the phospho-site, such as the one found in the PLK1 T-loop (Hoermann et al, 2020). The Aurora A T-loop, on the other hand, contains no upstream lysines and one fewer basic residue at position P−1, likely explaining its efficient dephosphorylation. On the other hand, PP1, expected to be very active toward basic phosphomotifs (Hoermann et al, 2020), was the least active in our experimental setup in both PLK1 and Aurora A dephosphorylation reaction. The comparatively low activity of our recombinant PP1 may be due to its expression in bacteria (Peti et al, 2012). In vivo, PP1 has been proposed to act on pThr210 in complex with the regulator MYPT1 (Yamashiro et al, 2008). λ-phosphatase behaved as a suitable control in this experiment, displaying broad substrate specificity and being able to dephosphorylate Aurora A and PLK1 with similar efficiency.

Whether these dephosphorylation reactions in vitro explain our results in vivo will require further studies. Nonetheless, the observation that Thr210 of PLK1 is largely impervious to PP2A and PP1 phosphatases is tantalizing, as it may explain the puzzling observation that the decrease of Bora levels in mitosis does not have major consequences for the PLK1 activation state, provided that low residual levels of Bora are present (Bruinsma et al, 2014). While it was initially speculated that the persistence of pThr210 under these conditions or after Aurora A inhibition may reflect temporal inactivation of the appropriate phosphatase during mitosis (Bruinsma et al, 2014) or protection by additional cofactor proteins such as Apolo1 (reference (Xu et al, 2021)), our results suggest that also sequence or structural elements around pThr210 site hamper its dephosphorylation by active phosphatases.

There are some corollary observations in our data that deserve a brief comment. First, our experiments with the Aurora A:Bora complex indicate that it can phosphorylate other substrates in addition to PLK1, as shown previously on a generic substrate (Hutterer et al, 2006). Whether this property of the Aurora A:Bora complex is physiologically significant will require further analysis. Second, our observations indicate that CEP192 acts as a partial inhibitor of Aurora A towards the substrates tested (Fig. 1F–I). This result is partly in line with recent work showing CEP192 does not activate (Park et al, 2023) or even inhibits Aurora A (Holder et al, 2024). It has been proposed that oligomerization of Aurora A on CEP192 in the presence of an oxidative environment promotes activation loop auto-phosphorylation and activation of Aurora A (Byrne et al, 2020; Lim et al, 2020; Wang et al, 2017). The specified condition cannot be straightforwardly tested using in vitro phosphorylation assays, and therefore we cannot exclude that CEP192-bound Aurora A at the centrosome is able to activate PLK1. Nevertheless, we can say that at least intrinsically, the complex phosphorylates PLK1 inefficiently on the activation loop.

Our results are graphically summarized in Fig. 7. We identify PLK1 as a suboptimal Aurora kinase substrate whose phosphorylation requires Bora. Such requirement can be bypassed by replacing lysine 208 of the activation loop with arginine. On the other hand, mutating residues at the Bora:PLK1 interface that we have identified abrogates Thr210 phosphorylation altogether, an effect that is rescued by introducing the K208R mutation. Thus, in summary, the two crucial elements of this mechanism are (a) the suboptimal target sequence of the motif containing Thr210, which creates a major barrier to phosphorylation, and (b) the overcoming of this limitation through a direct interaction of Bora with PLK1's kinase domain. Although additional studies will be needed to fully comprehend the network generated by the binding of Bora to crucial mitotic kinases such as CDK1, Aurora A, and PLK1, our

data offer the long-sought-for molecular explanation for why Bora is required as an Aurora A cofactor to phosphorylate Thr210 of PLK1.

# Methods

### Reagents and tools table

| Reagent/resource | Reference or source | Identifier or catalog number |
|---|---|---|
| **Experimental models** | | |
| HeLa cells | IEO Milan | N/A |
| HeLa Flp-In T-REx | Gift from Stephen Taylor Lab (Tighe et al, 2008) | N/A |
| HeLa Flp-In T-REx CENP-A-SNAP | Pan et al, 2017 | N/A |
| HeLa Flp-In T-REx CENP-A SNAP mNG-PLK1 WT | This study | N/A |
| HeLa Flp-In T-REx CENP-A SNAP mNG-PLK1 K208R | This study | N/A |
| HeLa Flp-In T-REx CENP-A SNAP mNG-PLK1 R95A | This study | N/A |
| HeLa Flp-In T-REx CENP-A SNAP mNG-PLK1 E121A | This study | N/A |
| HeLa Flp-In T-REx CENP-A SNAP mNG-PLK1 K208R E121A | This study | N/A |
| HeLa Flp-In T-REx CENP-A SNAP mNG-Bora WT | This study | N/A |
| HeLa Flp-In T-REx CENP-A SNAP mNG-Bora F56A | This study | N/A |
| HeLa Flp-In T-REx CENP-A SNAP mNG-Bora W58A | This study | N/A |
| HeLa Flp-In T-Rex CENP-A SNAP mCherry-PLK1 WT | This study | N/A |
| E.coli:BL21CodonPlus (DE3)-RIL strain | Agilent Technologies | Cat#230240 |
| E.coli:OmniMax | InvitrogenAgilent Technologies | |
| E.coli:EMBacY | GENEVA BIOTECH (from Imre Berger Group) | |
| Spodoptera frugiperda 9 (Sf9) | ThermoFisher Scientific, Invitrogen | |
| Trichoplusia ni 38 (Tnao38) | Gift from Gary W. Blissard | |
| Expi293F cells | Thermo Fisher Scientific | |
| **Recombinant DNA** | | |
| pCDNA5/FRT/TO | Invitrogen | Cat#V601020 |
| pCDNA5/FRT/TO-EGFP-IRES | Krenn et al, 2012 | N/A |
| pCDNA4/TO | Gift from Andrew Carter Lab | N/A |
| pOG44 | Gift from Stephen. Taylor; Krenn et al, 2012 | N/A |
| pET28 | GenScript | |
| pET Duet-1 | Novagen | Cat#71146 |
| pCDNA5 mNeonGreen – PLK1 WT | This study | N/A |
| pCDNA5 mNeonGreen – PLK1 K208R | This study | N/A |
| pCDNA5 mNeonGreen – PLK1 R95A | This study | N/A |
| pCDNA5 mNeonGreen – PLK1 E121A | This study | N/A |

| Reagent/resource | Reference or source | Identifier or catalog number |
|---|---|---|
| pCDNA5 mNeonGreen – PLK1 E121A K208R | This study | N/A |
| pCDNA5 mNeonGreen – Bora WT | This study | N/A |
| pCDNA5 mNeonGreen – Bora F56A | This study | N/A |
| pCDNA5 mNeonGreen – Bora W58A | This study | N/A |
| pCDNA5 mCherry-PLK1 WT | This study | N/A |
| pcDNA4/TO- eGFP-(HRV3C)-B55 | This study | N/A |
| pcDNA4/TO-StreptagII-(TEV)-PP2Ac | This study | N/A |
| pET28-6xHis-(Thrombin)-(T7)-Aurora A | This study | N/A |
| pET28-6xHis-(Thrombin)-(T7)-Aurora A T288V | This study | N/A |
| pETDuet-1 MBP-(TEV)-TPX2-8xHis | This study | N/A |
| pETDuet-1 MBP-(TEV)-CEP192$^{400-600}$-8xHis | This study | N/A |
| pETDuet-1 MBP-(TEV)-Bora$^{1-224}$-8xHis | This study | N/A |
| pETDuet-1 MBP-(TEV)-Bora$^{1-224}$-8xHis F56A | This study | N/A |
| pETDuet-1 MBP-(TEV)-Bora$^{1-224}$-8xHis W58A | This study | N/A |
| pETDuet-1 MBP-(TEV)-Bora$^{1-224}$-8xHis S59A | This study | N/A |
| pETDuet-1 MBP-(TEV)-Bora$^{1-224}$-8xHis I66A | This study | N/A |
| pETDuet-1 MBP-(TEV)-MAP205$^{264-322}$-8xHis | This study | N/A |
| pETDuet-1 MBP-(TEV)-BUB1$^{569-616}$-8xHis | This study | N/A |
| pETDuet-1 MBP-(TEV)-(SpyT3rd)-INCENP $^{351-918}$ : Halo-(HRV3C)-Aurora B-8xHis | This study | N/A |
| pETDuet-1 6xHis-MBP-(TEV)-PPP2R1A$^{9-589}$ | This study | N/A |
| pETDuet-1 6xHis-(HRV3C)-MBP-(TEV)-PLK1 | This study | N/A |
| pETDuet-1 6xHis-(HRV3C)-MBP-(TEV)-PLK1 K208R | This study | N/A |
| pETDuet-1 6xHis-(HRV3C)-MBP-(TEV)-PLK1 R48A | This study | N/A |
| pETDuet-1 6xHis-(HRV3C)-MBP-(TEV)-PLK1 R95A | This study | N/A |
| pETDuet-1 6xHis-(HRV3C)-MBP-(TEV)-PLK1 R106A | This study | N/A |
| pETDuet-1 6xHis-(HRV3C)-MBP-(TEV)-PLK1 E121A | This study | N/A |
| pETDuet-1 6xHis-(HRV3C)-MBP-(TEV)-PLK1 K82R | This study | N/A |
| pETDuet-1 6xHis-(HRV3C)-MBP-(TEV)-PLK1 K82R K208R | This study | N/A |
| pETDuet-1 6xHis-(HRV3C)-MBP-(TEV)-PLK1 K82R K208R E121A | This study | N/A |
| pETDuet-1 6xHis-(HRV3C)-MBP-(TEV)-PLK1 K82R E121A | This study | N/A |
| pETDuet-1 6xHis-(HRV3C)-MBP-(TEV)-PLK1 K82R K208R R95A | This study | N/A |
| pETDuet-1 6xHis-(HRV3C)-MBP-(TEV)-PLK1 K82R R95A | This study | N/A |
| pETDuet-1 6xHis-(HRV3C)-MBP-(TEV)-PLK1 1-326 K82R | This study | N/A |

| Reagent/resource | Reference or source | Identifier or catalog number |
|---|---|---|
| pETDuet-1 6xHis-(HRV3C)-MBP-(TEV)-PLK1 1-326 K82R K208R | This study | N/A |
| pETDuet-1 6xHis-(HRV3C)-MBP-(TEV)-PLK1 1-326 K82R E121A | This study | N/A |
| pETDuet-1 6xHis-(HRV3C)-MBP-(TEV)-PLK1 1-326 K82R R95A | This study | N/A |
| pETDuet-1 6xHis-(HRV3C)-MBP-(TEV)-PLK1 1-326 K82R K208R R95A | This study | N/A |
| pETDuet-1 6xHis-(HRV3C)-MBP-(TEV)-PLK1 1-326 K82R K208R E121A | This study | N/A |
| pETDuet-1 6xHis-(HRV3C)-MBP-(TEV)-PLK1 1-326 K82R E206L | This study | N/A |
| pETDuet-1 6xHis-(HRV3C)-MBP-(TEV)-PLK1 1-326 K82R R207K | This study | N/A |
| pETDuet-1 6xHis-(HRV3C)-MBP-(TEV)-PLK1 203-217 | This study | N/A |
| pETDuet-1 6xHis-(HRV3C)-MBP-(TEV)-PLK1 203-217 K208R | This study | N/A |
| pETDuet-1 6xHis-(HRV3C)-MBP-(TEV)-SA2[1217-1232] | This study | N/A |
| pGEX6P-1 GST-(HRV3C)-B56γ3 | This study | N/A |
| pGEX4T-1 GST-(TEV)-PPP2R1A | This study | N/A |
| pDEST8-8xHis-PPP2CA | This study | N/A |
| pLIB-GST-(HRV3C)-CDK1 | Huis In 't Veld et al, 2022 | N/A |
| pLIB-His-(TEV)-Cyclin B1 | Huis In 't Veld et al, 2022 | N/A |
| pLIB-His-(TEV)-Cks1 | Huis In 't Veld et al, 2022 | N/A |
| MBP-scCAK1 | Huis In 't Veld et al, 2022 | N/A |
| GST-(HRV3C)-NUF2[1-169]:SPC24[122-197], NDC80[1-286]:SPC25[118-224] | Ciferri et al, 2008 | N/A |
| **Antibodies** | | |
| Aurora A (mouse monoclonal, diluted 1/1000) | BD Biosciences | #610938 |
| Aurora A pT288 /Aurora B pT232/ Aurora C pT236 (rabbit monoclonal, diluted 1/1000) | Cell Signalling | #2914 |
| Bora (rabbit monoclonal, diluted 1/1000) | Cell Signalling | #12109S |
| CENP-A pS7 (rabbit polyclonal, diluted 1/1000) | Abcam | #ab62242 |
| GAPDH (rabbit polyclonal, diluted 1/5000) | Sigma | # G9545 |
| H3 pS10 (mouse monoclonal, diluted 1/1000) | Abcam | #ab14955 |
| Ndc80 pS55 (rabbit polyclonal, diluted 1/1000) | Invitrogen | #PA5-85846 |
| PLK1 (mouse monoclonal, diluted 1/1000) | Abcam | #ab17057 |
| PLK1 pT210 (mouse monoclonal, diluted 1/1000) | Abcam | #ab39068 |
| PLK1 pT210 (mouse monoclonal, diluted 1/1000) | BioLegend | #628901 |
| SA2 pS1224 (rabbit, diluted 1/1000) | Gift from Jan Michael Peters Lab | NA |
| Vinculin (mouse monoclonal, diluted 1/10000) | SIGMA | #V9131 |
| anti-mouse IgG Alexa 647 secondary (diluted 1/2000) | Invitrogen | A31571 |

| Reagent/resource | Reference or source | Identifier or catalog number |
|---|---|---|
| anti-IgG mouse coupled to HRP (secondary antibody – source sheep) (diluted 1/5000) | Amersham | NXA931-1ML |
| anti-IgG rabbit coupled to HRP (secondary antibody – source donkey) (diluted 1/5000 | Amersham | NA934V |
| **Oligonucleotides and other sequence-based reagents** | | |
| siBora 5'-TAACTAGTCCTTCGCCTATTT-3' | Chan et al, 2008 | N/A |
| siPLK1 5'-AUACCUUGUUAGUGGGCAATT-3' | Invitrogen | |
| **Chemicals, enzymes, and other reagents** | | |
| Adenosine 5'-triphosphate (ATP) | SigmaCarl Roth | |
| Blasticidin | Invitrogen | Cat#A1113902 |
| Bovine serum albumin (BSA) | VWR | Cat#422351S |
| Coomassie Brilliant Blue R-250 dye | Thermo Fisher | Cat#20278 |
| BI2536 | Focus BiomoleculesSelleck Chemicals Llc | Cat#S110910-421 |
| Bradford reagent | ThermoFisher Scientific | |
| DMEM | PAN Biotech | Cat#P0403609 |
| DMSO | Serva | |
| DNAse I | Roche | |
| Doxycycline | Sigma | Cat#D9891 |
| Ethanol | Serva | |
| Expi293 Expression Medium | ThermoFisher Scientific | |
| Fetal Bovine Serum (FBS) | Clonetech | Cat#631107 |
| Gibson assembly enzymes | New England Biolabs | |
| Glutathione Sepharose 4 Fast Flow beads | ServaGE Healthcare/Cytiva | Cat#17-5132-01 |
| Glycerol | Sigma | Cat#G5516 |
| GSTrap FF column 5 mL | GE Healthcare/Cytiva | Cat#17513102 |
| HEPES | Sigma | Cat#H3375 |
| HiTrap TALON crude | GE Healthcare/Cytiva | Cat#45-002-386 |
| HisTrap FF column | GE Healthcare/Cytiva | |
| Hygromycin B | Invitrogen | Cat#10687010 |
| Imidazole | Sigma | Cat#I5513 |
| IPTG | Sigma | Cat#I6758 |
| L-glutamine | PAN Biotech | Cat#P04-80100 |
| L-Glutathione reduced | Sigma | Cat#G6529 |
| Lipofectamine 2000 | Invitrogen | Cat#11668019 |
| Nocodazole | Sigma | Cat#M1404 |
| MBPTrap HP 5 mL | Cytiva | Cat#28918780 |
| MgCl2 | Sigma | Cat#M8266 |
| mNeongreen Trap Magnetic Agarose beads | Chromotek | AB_2827594 |
| Mowiol | Calbiochem | Cat#475904 |
| Ni-NTA beads | Roche | |
| Nonidet-P40 Substitute | Fluka Chemie AG | Cat#9016-45-9 |
| Okadaic acid | Merck | Cat#04906845001 |
| Optimem | Invitrogen | |
| PAE | Sigma | |
| Penicillin-Streptomycin | PAN Biotech | Cat# P06-07100 |

| Reagent/resource | Reference or source | Identifier or catalog number |
|---|---|---|
| PhosSTOP phosphatase inhibitor | Roche | Cat#4906845001 |
| PMSF | Serva | |
| Poly-L-Lysine | Sigma | P4832-50ml |
| ProQ Diamond Phosphoprotein Gel Stain | ThermoFisher Scientific | Cat#P33300 |
| Protease inhibitor cocktail | Serva | Cat#39107 |
| Resource Q anion exchange chromatography column | GE Healthcare/Cytiva | Cat#17-1179-01 |
| RNAiMAX | Invitrogen | Cat#13778100 |
| SF900III SFM(1x) | ThermoFisher Scientific | |
| SiRDNA | Spirochrome | |
| Sodium Chloride (NaCl) | Sigma | Cat#S9888 |
| Superdex 5/150 200 pg sizing column | GE Healthcare/Cytiva | |
| Superdex 10/300 200 pg sizing column | GE Healthcare/Cytiva | |
| Superdex 10/300 75 pg sizing column | GE Healthcare/Cytiva | |
| Superdex 16/600 200 pg sizing column | GE Healthcare/Cytiva | |
| TCEP | Sigma | Cat#75259 |
| Thymidine | Sigma | Cat#T9250 |
| Triton X-100 | Sigma | Cat#T8787 |
| Trypsin/EDTA | PAN Biotech | Cat#P10-027100 |
| Tris-(hydroxymethyl)-aminomethane (Tris) | Biosynth AG | |
| Tris-(2-carboxyethyl)-phosphine (TCEP) | Sigma | Cat#75259 |
| X-tremeGENE | Roche | Cat#4476093001 |
| Ultrafiltration concentrator Amicon | Merck Millipore | |
| Western Blot ECL detection kit | Cytiva | |
| TEV protease | Produced in-house | N/A |
| HRV3C protease | Produced in-house | N/A |
| λ-Phosphatase | Produced in-house | N/A |
| 6xHis-PP1 | Gift from Carlos Conde Lab | N/A |
| 6xHis-Aurora A full-length | This study | N/A |
| 6xHis-Aurora A full-length T288V | This study | N/A |
| MBP-TPX2-8xHis full length | This study | N/A |
| MBP-CEP192$^{400-600}$-8xHis | This study | N/A |
| MBP-Bora$^{1-224}$-8xHis | This study | N/A |
| MBP-Bora$^{1-224}$-8xHis F56A | This study | N/A |
| MBP-Bora$^{1-224}$-8xHis W58A | This study | N/A |
| MBP-Bora$^{1-224}$-8xHis S59A | This study | N/A |
| MBP-Bora$^{1-224}$-8xHis I66A | This study | N/A |
| 6xHis-MBP-PLK1 | This study | N/A |
| 6xHis-MBP-PLK1 K82R | This study | N/A |
| 6xHis-MBP-PLK1 K208R | This study | N/A |
| 6xHis-MBP-PLK1 K82R K208R | This study | N/A |
| 6xHis-MBP-PLK1 R48A | This study | N/A |
| 6xHis-MBP-PLK1 R95A | This study | N/A |
| 6xHis-MBP-PLK1 K82R R95A K208R | This study | N/A |
| 6xHis-MBP-PLK1 R106A | This study | N/A |

| Reagent/resource | Reference or source | Identifier or catalog number |
|---|---|---|
| 6xHis-MBP-PLK1 E121A | This study | N/A |
| 6xHis-MBP-PLK1 K82R E121A K208R | This study | N/A |
| 6xHis-MBP-PLK1 K82R R95A | This study | N/A |
| 6xHis-MBP-PLK1 K82R E121A | This study | N/A |
| 6xHis-MBP-PLK1 K82R E207L | This study | N/A |
| 6xHis-MBP-PLK1 K82R R207K | This study | N/A |
| 6xHis-MBP-PLK1$^{1-326}$ K82R | This study | N/A |
| 6xHis-MBP-PLK1$^{1-326}$ K82R K208R | This study | N/A |
| 6xHis-MBP-PLK1$^{203-217}$ | This study | N/A |
| 6xHis-MBP-PLK1$^{203-217}$ K208R | This study | N/A |
| 6xHis-(HRV3C)-MBP-TEV-SA2$^{1217-1232}$ | This study | N/A |
| (SpyT3rd)-INCENP 351-918/Halo-(HRV3C)-Aurora B-8xHis | This study | N/A |
| PPP2R1A/PP2Ac/B55 complex | This study | N/A |
| MBP-(TEV)-dmMAP205$^{264-322}$-8xHis | This study | N/A |
| MBP-(TEV)-BUB1$^{569-616}$-8xHis | This study | N/A |
| PPP2R1A/8xHis-PPP2CA/B56γ3 complex | This study | N/A |
| CENP-A$^{NCP}$ | Weir et al, 2016 | N/A |
| H3$^{NCP}$ | Weir et al, 2016 | N/A |
| CDK1:Cyclin-B:CKS1 | Huis In't Veld et al, 2022 | N/A |
| Ndc80 bonsai | Ciferri et al, 2008 | N/A |
| **Software** | | |
| GraphPad Prism 10.0.3.(217) | GraphPad Software Inc | https://www.graphpad.com |
| ImageJ Version 2.0.0-rc-69/1.52p | NIH | https://imagej.nih.gov/ij/ |
| softWoRx Version | GE Healthcare/Cytiva | |
| ImageLab | Bio-Rad | |
| Pymol | Schrödinger | https://www.pymol.org |
| Adobe Illustrator | Adobe Inc. | |
| Microsoft office | Microsoft Corporation | |
| Unicorn | GE Healthcare/Cytiva | |
| **Other** | | |
| ÄKTA system | GE Healthcare/Cytiva | |
| ÄKTA micro system | GE Healthcare/Cytiva | |
| Deltavision Elite | GE Healthcare/Cytiva | |
| ClarioStar Plus plate reader | BMG Labtech | |
| Bio-Rad ChemiDoc MP Imaging System | Bio-Rad | |
| Mini Prep Cell apparatus | Bio-Rad | |
| Trans-Blot system | Bio-Rad | |

## Methods and protocols

### Materials and methods

Mutagenesis and sub-cloning into expression plasmids. All DNA constructs were generated by PCR amplification of the DNA fragment encoding the protein (insert) and the expression vector of choice or PCR amplification of the insert only and vector cleavage with restriction enzymes. Both strategies were followed by a ligation

step via Gibson Assembly (enzymes were purchased from New England Biolabs, and the master mix was produced in-house). Mutations were introduced via site-directed mutagenesis with a single forward primer. The protein encoding region of all the final constructs was verified by Sanger sequencing (Microsynth Seqlab). Oligonucleotides used for site-directed mutagenesis and cloning were purchased from Sigma. A list of plasmids used in this study can be found in the Reagents and Tools Table.

Expression of recombinant proteins. Full-length human His-Aurora A$^{WT}$, His-Aurora A$^{T288V}$, Halo-Aurora B$^{FL}$ co-expressed with MBP-INCENP$^{351-918}$, GST-B56γ3, GST-PPP2R1A, as well as all human MBP-Bora-His fragments and mutants, His-MBP-PLK1$^{WT}$ and its truncations or mutants, His-MBP-SA2$^{1217-1232}$, MBP-TPX2$^{FL}$-His, MBP-CEP192$^{400-600}$-His, MBP-Map205$^{264-322}$-His, MBP-BUB1$^{569-616}$-His were expressed in Escherichia coli BL21 CodonPlus (DE3)-RIL (Agilent Technologies). Transformed bacteria were incubated in Luria Broth (Bora, B56γ3, and TPX2) or Terrific Broth (Aurora A, Aurora B:INCENP, PLK1, PPP2R1A, MAP205, BUB1, and CEP192) media at 37 °C under agitation, and the cultures were shifted at 18 °C at $OD_{600}$ = 0.6-0.8. Protein expression was induced by the addition of IPTG (final concentration 100 µM) and the cultures were incubated 18 h at 18 °C under agitation. Cyclin B-CDK1-CKS1 was obtained from insect cells TnaO38 expressing GST-CDK1 with a 1:25 dilution of viral stock in the expression culture (for a standard protocol of virus generation, see reference (Trowitzsch et al, 2010)). His-CKS1 and MBP-scCAK1 (also with a 1:25 dilution) were cultured for three days at 27 °C (Huis In 't Veld et al, 2022). His-PPP2CA was purified from insect cells (Tnao38), kept in culture after transfection of the virus at a 1:20 ratio, for 4 days at 19 °C (Ikehara et al, 2016). Ndc80$^{bonsai}$ expression in E. coli BL21(DE3) was induced with 400 µM IPTG at $OD_{600}$ = 0.45–0.6 for 12–16 hr at 20–25 °C (Ciferri et al, 2008). PP2Ac:B55δ were co-expressed in Expi293F cells (Thermo Fisher Scientific), grown in suspension in Expi293 Expression Medium (Thermo Fisher Scientific), using the BigBac system to generate the virus for transfection (Weissmann et al, 2016). Briefly, the DNA fragments encoding full-length B55 delta (B55d) and PP2A catalytic (PP2Ac) subunits were cloned into pcDNA4/TO vector (containing CMV promoter for the expression in mammalian cells) downstream to and in frame with a region encoding an N-terminal enhanced Green Fluorescent Protein (eGFP) followed by the HRV 3C/Prescission cleavage site, or an N-terminal StreptagII followed by TEV protease cleavage site, respectively. The coding sequences were then cloned from pcDNA4/TO into pBIG1a vector, using cassette primers, to generate the bacmid. For the expression in 1 L of Expi293F cells, they were infected with V2 virus culture precipitated overnight. To precipitate V2 virus, 4× Polyethylene glycol (PEG) solution was used—32% PEG6000 (Merck)—, 400 mM of NaCl, 40 mM of HEPES pH 7.4). The virus was mixed with 4xPEG solution in 3:1 ratio and incubated at 4 °C in the dark overnight. The day before the infection Expi293F cells were split to 2–2.5 × 10$^6$/mL, so that on the day of infection the density was 3.5–4.5 × 10$^6$/mL. The following day, the mixture of V2 virus and PEG was centrifuged at 4000 RPM, at 4 °C, for 30 min. The supernatant was removed, and the pellet was resuspended in 5 mL of Expi293 Expression Medium (Thermo Fisher Scientific) prewarmed to 37 °C. Once the cell density reached the desired range (3.5–4.5 × 10$^6$/mL), the resuspended virus was added at a 1:200 ratio. In all, 6–8 h post infection, sodium butyrate (Sigma-Aldrich) was added to Expi293F culture to 5 mM

final concentration. The expression was performed at 37 °C, 8% $CO_2$ under agitation. In all, 48–52 h post infection cells were pelleted by centrifugation, snap-frozen in liquid nitrogen, and stored at −80 °C until further use.

Purification of recombinant proteins. All steps of the purification procedures below were performed on ice or at 4 °C.

Aurora A WT and T288V was purified as a 6xHis-thrombin site-T7 tag fusion. Bacteria were pelleted and resuspended in lysis buffer (50 mM HEPES pH 7.5, 300 mM NaCl, 5% vol/vol glycerol, 1 mM TCEP, 2 mM PMSF), lysed by ultrasonication on ice, and cleared by centrifugation at 80,000× g at 4 °C for 30-45 min. The resulting soluble lysate was subjected to Nickel or Cobalt-based chromatography by loading it on 5-mL of packed NiNTA or Talon column (Cytiva), pre-equilibrated in affinity buffer (50 mM HEPES pH 7.5, 300 mM NaCl, 5% vol/vol glycerol, 1 mM TCEP). After extensive washes, protein was eluted with 300 mM Imidazole in the affinity buffer. The eluted sample was concentrated using an ultrafiltration centrifugal protein concentrator (MerckMillipore). The protein was loaded on a HiLoad 16/600 Superdex 200 pg sizing column (Cytiva), equilibrated in SEC buffer (50 mM HEPES pH 7.5, 300 mM NaCl, 5% vol/vol glycerol, 1 mM TCEP). The protein-containing fractions were pooled and concentrated as described above. The purified protein was aliquoted, flash-frozen in liquid nitrogen, and stored at −80 °C.

Human Bora$^{1-224}$, human TPX2$^{FL}$, human CEP192$^{400-600}$, MAP205$^{264-322}$ from Drosophila melanogaster, and human BUB1$^{569-616}$, were expressed and purified as tagged proteins, bearing an N-terminal MBP tag and a C-terminal 8xHis fusion. The SA2 peptide comprising the PLK1 phosphorylation site Ser1224 was purified as an N-terminally tagged His-MBP-protein. Bacteria from 1-L culture (Bora, MAP205, BUB1 and SA2) or 2-liter cultures (TPX2 and CEP192) were harvested by centrifugation and resuspended in lysis buffer A (20 mM Tris pH 8, 500 mM NaCl, 1 mM TCEP, 0.1% Triton X-100) for Bora, TPX2 and CEP192 or B for MAP205, BUB1 and SA2 (50 mM HEPES pH 7.5, 250 mM NaCl, 5% vol/vol glycerol, 1 mM TCEP), supplemented with HP plus protease inhibitor mix (Serva) and DNase I (Roche). The same protocol described before has been applied for protein purification and storage, except that buffer A devoid of Triton (Bora, TPX2, and CEP192), or buffer B (MAP205, BUB1, and SA2) were used throughout the entire procedure.

Human PLK1 wt and its variants were expressed as TEV cleavable 6xHis-MBP fusions, which leaves ten non-native residues (GPGA-SASALA) at the N-terminus upon TEV digestion. Bacteria from 2-L culture were pelleted and resuspended in lysis buffer (50 mM Hepes pH 7.5, 300 mM NaCl, 5% vol/vol glycerol, 1 mM TCEP, 2 mM PMSF). Every subsequent step was performed in lysis buffer devoid of PMSF. The purification protocol described above has been modified to allow the TEV-mediated cleavage of His-MBP after elution. His-TEV protease (produced in-house) has been added and incubated overnight with the eluted protein. The protein was finally resolved on a HiLoad 16/600 Superdex 200 pg sizing column (Cytiva) and additionally separated from His-TEV by mounting a 5 mL NiNTA pre-packed column downstream of the SEC column.

B56γ3 was expressed as an H3RVC-cleavable GST-fusion, which leaves five non-native residues (GPLGS) at the N-terminus upon Prescission digestion. Bacteria from 1 Liter culture were harvested by centrifugation and resuspended in lysis buffer (50 mM Hepes pH 7.5,

300 mM NaCl, 5% vol/vol glycerol, 1 mM TCEP) complemented with HP plus protease inhibitor mix (Serva) and DNase I (Roche). The cell resuspension was lysed by ultrasonication on ice and cleared by centrifugation at 80,000× g at 4 °C for 30–45 min. The resulting clarified lysate was subjected to GSH-based affinity chromatography by loading it on 5 mL of pre-equilibrated Glutathione agarose beads (Serva) slurry. After extensive washes, GST-Prescission protease (produced in-house) was added to the lysis buffer, and the beads were kept shaking gently to allow the removal of the GST-tag, overnight. The sample containing cleaved B56 protein was collected and concentrated by ultrafiltration as described for other protein purifications. The protein was resolved on a pre-equilibrated HiLoad 16/600 Superdex 200 pg sizing column (Cytiva). The protein-containing fractions were pooled and concentrated as described above. The purified protein was aliquoted, flash-frozen in liquid nitrogen, and stored at -80 °C. PPP2R1A was cloned as a TEV-cleavable GST-fusion, which leaves one non-native residue (G) at the N-terminus after TEV digestion. The same protocol as for GST-B56 was applied, except that cleavage was performed with His-TEV protease, which was separated by SEC from the purified protein.

6xHis-PPP2CA-expressing cells from a 2-L culture were resuspended in lysis buffer (50 mM Hepes pH 7.5, 150 mM NaCl, 2 mM TCEP, 5% vol/vol glycerol) supplemented with HP plus protease inhibitor mix (Serva) and DNase I (Roche). After ultrasonication and centrifugation at 85,000× g for 45 min, the supernatant was filtered through a 0.8 µm filter. The sample was loaded on 2 × 5-mL Talon column and washed extensively with lysis buffer, to then be eluted in 2-mL fractions with elution buffer (50 mM Hepes pH 7.5, 150 mM NaCl, 2 mM TCEP, 5% vol/vol glycerol, 250 mM imidazole). Fractions containing the protein were selected by using Bradford reagent (Thermo Fisher Scientific). The selected fractions were pooled and diluted 1:4 in ion exchange buffer (50 mM HEPES pH 7.5, 50 mM NaCl, 5% vol/vol glycerol, 1 mM TCEP), to be loaded on a 6-mL ResourceQ column (Cytiva). After a washing step with ion exchange buffer, protein was eluted with a gradient up to 400 mM NaCl in ion exchange buffer A. Protein content was inspected via Coomassie Blue-stained SDS–PAGE gel, and fractions were pooled and concentrated by ultrafiltration. The resulting sample was loaded and resolved on a Superdex 10/300 75 pg sizing column (Cytiva), pre-equilibrated in SEC buffer (same composition as lysis buffer). Fractions from SEC were pooled, concentrated as described above, and aliquoted. Aliquots were snap-frozen and stored at −80 °C. PP2A-B56 complex formation was carried out prior to the experiment by mixing a 1:1:1 ratio of PPP2R1A, PPP2CA, and B56.

PP2A-B55 complex was purified as previously described, with few adjustments reported below (Padi et al, 2024). The PP2Aa subunit (residues 9–589) was expressed as a His-MBP-tagged fusion. Cells from 500 mL of culture were resuspended in lysis buffer (50 mM Tris pH 8.0, 500 mM NaCl, 5 mM imidazole, 0.5 mM TCEP, 0.1% Triton X-100) supplemented with protease inhibitors (Serva), and DNase (Roche) and lysed by ultrasonication. The lysate was clarified by centrifugation at 80,000× g for 45 min at 4 °C, and the supernatant was loaded onto 2×5 mL HisTrap FF columns (Cytiva) pre-equilibrated with affinity buffer (50 mM Tris pH 8.0, 500 mM NaCl, 5 mM imidazole, 0.5 mM TCEP). Following extensive washing, the protein was eluted in a single step with 500 mM imidazole in the same buffer. Eluted fractions were pooled and dialyzed overnight at 10 °C with in-house purified His-TEV protease in SnakeSkin tubing (Thermo Fisher Scientific) to cleave the His-MBP tag. The dialyzed

mixture was incubated with equilibrated Ni-NTA agarose beads (Roche) to remove the tag, and the flow-through containing cleaved PP2Aa was collected. The sample was concentrated by ultrafiltration, diluted to 100 mM NaCl with low-salt buffer (20 mM Tris pH 8.0, 0 mM NaCl, 0.5 mM TCEP), and brought to 100 mL total volume with buffer A (20 mM Tris pH 8.0, 100 mM NaCl, 0.5 mM TCEP). The solution was filtered (0.2 µm) and loaded onto a 6 mL HiTrap Q column (Cytiva). Bound protein was eluted over a linear 100–1000 mM NaCl gradient. Relevant fractions were pooled, concentrated by ultrafiltration, and subjected to SEC (HiLoad 16/600 Superdex 200 pg, Cytiva) in SEC buffer (20 mM Tris pH 8.0, 150 mM NaCl, 0.5 mM TCEP). Fractions containing PP2Aa were pooled, concentrated, aliquoted, flash-frozen in liquid nitrogen, and stored at 80 °C. The His-PPP2CA:B55δ holoenzyme was purified from 1 L of Expi293F cell pellets co-expressing StrepII-PP2Ac and eGFP-B55δ. All steps were performed on ice or at 4 °C. Cells were resuspended in lysis buffer (20 mM Tris pH 8.0, 500 mM NaCl, 0.5 mM TCEP, 0.1% Triton X-100) supplemented with protease inhibitors (Serva), and DNase (Roche) and lysed by sonication. To promote holoenzyme assembly, 6 mg of purified PP2Aa was added to the lysate before clarification by centrifugation at 80,000× g for 45 min at 4 °C. The supernatant was filtered (0.8 µm) and then incubated with GSH agarose resin (Serva) pre-bound to GST-GFP nanobodies (see nanobody purification section below). After washing, beads were resuspended directly on the column in cleavage buffer (20 mM Tris pH 8.0, 250 mM NaCl, 1 mM MnCl₂, 0.5 mM TCEP), and in-house purified GST-PreScission protease was added for overnight cleavage at 10 °C. The flow-through containing the eluted complex was collected, concentrated by ultrafiltration, and diluted to 100 mM NaCl by addition of low-salt buffer (20 mM Tris pH 8.0, 0 mM NaCl, 1 mM MnCl₂, 0.5 mM TCEP). The total volume was adjusted to 100 mL with buffer A (20 mM Tris pH 8.0, 100 mM NaCl, 1 mM MnCl₂, 0.5 mM TCEP), filtered through a 0.2 µm membrane (Cytiva), and loaded onto a 6 mL HiTrap Q column (Cytiva). The column was washed with buffer A and eluted over a linear gradient from 100 mM to 1000 mM NaCl in buffer A. Fractions containing the PP2A:B55δ holoenzyme were pooled, concentrated in storage buffer (20 mM Tris pH 8.0, 150 mM NaCl, 1 mM MnCl₂, 0.5 mM TCEP) by ultrafiltration, aliquoted, snap-frozen in liquid nitrogen, and stored at −80 °C.

For purification of GST-GFP nanobody, 4 liters of bacterial cell pellets expressing GST-GFP nanobody were resuspended in lysis buffer (50 mM Hepes pH 7.5, 300 mM NaCl, 5% glycerol, 1 mM TCEP) with 1 mM PMSF (Serva). The suspension was lysed by ultrasonication and centrifuged (80000x g, 45 min). The supernatant was loaded onto 5 mL GST-trap FF column (Cytiva), equilibrated in lysis buffer. The column was washed, and the bound protein was eluted using elution buffer (50 mM Hepes pH 7.5, 300 mM NaCl, 5% glycerol, 1 mM TCEP, 20 mM Glutathione (GSH) (Roth)). The eluted protein was concentrated by ultrafiltration and further purified, using SEC, HiLoad 16/600 Superdex 200 pg (Cytiva), in SEC buffer (50 mM HEPES pH 7.5, 300 mM NaCl, 5% glycerol, 1 mM TCEP). The fractions containing the protein were pooled, concentrated by ultrafiltration, snap-frozen in liquid nitrogen, and stored at −80 °C. The day of PP2A purification, the GST-eGFP nanobodies were mixed with GSH agarose resin (Serva) equilibrated with the PP2A:B55d resuspension buffer (20 mM Tris pH 8.0, 500 mM NaCl, 0.5 mM TCEP) for 3 h at 10 °C to immobilize the nanobodies to the agarose beads, using a magnetic stirrer at low

speed. The beads were collected into 10 mL drop columns, and washed with the resuspension buffer, to be then directly used for mixing with the supernatant of Expi293F cells expressing PP2Ac and B55d subunits.

Halo-Aurora B-8xHis/$^{MBP}$INCENP$^{351-C}$ (co-expressed via transformation of BL21CodonPlus(DE3)-RIL with a pETDuet-1 plasmid) from 4 liters of expression culture was resuspended in Lysis buffer (50 mM Hepes pH 7.5, 500 mM NaCl, 5% vol/vol glycerol, 1 mM TCEP, 5 mM MgCl$_2$, 1 mM PMSF). After lysis via sonication, Dnase (Roche) was added and incubated 30 min at 4 °C. Lysate was clarified by centrifugation at 75,000×$g$ for 30 min. Cleared lysate was supplemented of Polyethylenimine (PEI, from Sigma) at 0.25% vol/vol, and the sample was spun down at 75,000×$g$ for 30 min. The supernatant was collected and loaded on Ni-NTA slurry beads placed in a drop-column at 4 °C. After washing, a 1 mM ATP wash in lysis buffer was added to remove chaperones. The elution was performed with elution buffer (50 mM HEPES pH 7.5, 300 mM NaCl, 5% vol/vol glycerol, 1 mM TCEP, and 300 mM imidazole). The eluate was incubated with His-TEV overnight. The protein after cleavage was concentrated, spun down and finally resolved on a HiLoad 16/600 Superdex 200 pg sizing column (Cytiva), pre-equilibrated in SEC buffer (50 mM Hepes pH 7.5, 300 mM NaCl, 5% vol/vol glycerol, 1 mM TCEP). Fractions containing the soluble complex of Halo-Aurora B-8xHis and INCENP 351-C were pooled, spun down aliquoted, and snap-frozen, to be stored at −80 °C for long-term storage.

Cyclin B-CDK1-Cks1 complex was purified similarly to what was previously described (Huis In 't Veld et al, 2022). In brief, frozen pellets containing overexpressed active GST-CDK1 and His-Cks1 were thawed and resuspended in lysis buffer (50 mM HEPES pH 7.4, 250 mM NaCl, 2 mM TCEP, 5% vol/vol glycerol) supplemented with HP plus protease inhibitor mix (Serva) and DNase I (Roche). Cell lysates were prepared by sonication and cleared by centrifugation at 80,000×$g$ at 4 °C for 30–45 min. The soluble lysate was passed through a 0.8 μm filter and loaded onto a column with 20 ml Glutathione Sepharose 4 FF resin (Cytiva). After washing, CDK1 was cleaved with Prescission protease and His-TEV protease (both produced in-house) for 16 h. The eluate was concentrated through centrifugation and separated on a Superdex 200 16/600 column equilibrated in SEC buffer (same composition as lysis buffer). To remove GST, uncleaved GST-CDK1, GST-3C-PreScission, and uncleaved Cks1 or His-TEV, a 5 ml GSH column (Cytiva) and a 5-mL NiNTA (Cytiva) were mounted after the size-exclusion column.

His-TEV-Cyclin-B containing lysates were prepared as described above with 15 mM imidazole in lysis and wash buffers (same composition as for CDK1 and Cks1). After loading onto a 5 ml or 10 ml Talon (Clontech) or Ni-NTA (GE Healthcare) column, and washing, Cyclin-B was eluted in buffer with 250 mM imidazole, and concentrated as described in the purification protocols above. To remove the polyhistidine tag, Cyclin-B was exposed to TEV protease for 16 h. Cyclin-B was further purified on a Superdex 200 16/600 column equilibrated in buffer A. To remove His-TEV protease and uncleaved His-Cyclin-B, a 5 ml Talon column (GE Healthcare) was mounted after the size-exclusion column.

Purified CDK1:Cks1 and Cyclin-B were mixed in a 1:1 ratio for 1–2 h on ice, and complex formation was assessed by SEC on a Superdex 200 16/600 column in buffer A. Fractions containing CCC were concentrated through centrifugation, flash-frozen in liquid

nitrogen, and stored at −80 °C.

Ndc80 bonsai purification was performed as described (Ciferri et al, 2008). Bacterial pellets were resuspended in lysis buffer (50 mM Tris-HCl, pH 7.6, 300 mM NaCl, 1 mM DTT, 1 mM EDTA), with the addition of Dnase (Roche) and protease-inhibitor mix HP Plus (Serva). Sonicated lysates were cleared by centrifugation at 80,000×$g$ for 45–60 min. The cleared lysate was bound to Glutathion-Agarose beads (3 ml resin for 5 L expression culture, Serva) equilibrated in washing buffer (lysis buffer without protease inhibitors). The beads were washed extensively, and protein was cleaved from the beads by overnight cleavage with 3 C PreScission protease (generated in-house). The eluate was concentrated using ultrafiltration and applied to a Superdex 200 10/300 column (Cytiva) equilibrated in 50 mM Hepes, pH 8.0, 250 mM NaCl, 2 mM TCEP, 5% v/v glycerol. Relevant fractions were pooled, concentrated, flash-frozen in liquid nitrogen, and stored at −80 °C.

CENP-A and H3-containing mononucleosomes assembled on 145-bp DNA were reconstituted precisely as described (Walstein et al, 2021). For the assembly of dinucleosomes, CENP-A and H3-containing NCPs were first reconstituted using two different CEN1-like DNA fragments. One micromolar of NCPs reconstituted on CEN1-like DNA fragment 1 and CEN1-like DNA fragment 2 were ligated to each other for 16 h at 4 °C using 8xHis-tagged T4 DNA ligase. The ligated NCPs were incubated with cOmplete nickel beads (Roche) for 7 h at 4 °C to remove the T4 DNA ligase. The ligated dinucleosomes were concentrated, and glycerol was added at a final concentration of 2% (v/v). The mixture was loaded on a cylindrical gel containing 5% reduced bis-acrylamide. Native PAGE was carried out on a Mini Prep Cell apparatus (Bio-Rad) at 1 W of constant power. Dinucleosomes were eluted at a constant flow rate of 0.1 ml/min overnight at 4 °C in nucleosome buffer (50 mM NaCl, 10 mM triethanolamine, and 1 mM EDTA) and collected in 250 μl of fractions on 96-well plates. The OD$_{260}$ (optical density at 260 nm) and OD$_{280}$ of the individual fractions were measured using a CLARIOstar Plus plate reader (BMG Labtech). Fractions containing dinucleosomes were pooled, concentrated, and stored at 4 °C.

Phosphorylation assays. The PLK1 phosphorylation assay consisted of the addition of 200 nM of PLK1 K82R (dead) or other variants as indicated in the figure legends, to 200 nM Aurora A, in the presence of 400 nM of Bora, TPX2, CEP192, or to 200 nM Aurora B:INCENP. The kinase buffer (50 mM HEPES 7.5, 150 mM NaCl, 5 mM MgCl$_2$, 1 mM TCEP, 2.5 mM ATP) was used to reach a final reaction volume of 60 μL, incubated for 1 h at 30 °C (or as otherwise indicated in figure legends and schemes). The reaction was stopped by adding 5× Laemmli sample buffer. Experiment in Fig. EV4F was performed at a higher concentration of PLK1 as substrate (2.5 μM) and a titration of Aurora kinase A and B up to equimolar concentration (as indicated in figure labels), and incubation was carried out for 30 min at 30 °C.

The assay with CDK1 was performed in two steps. Step 1 consisted of the phosphorylation of 4 μM Bora 1–224 with 100 nM Cyclin B1-CDK1-CKS1 in 30 μL of kinase buffer incubated for 1 h at 30 °C. The step 2 consisted of the dilution of Bora to 400 nM in the reaction mixture. The final concentration of the components and the experimental setup were the same as described above.

The phosphorylation of His-MBP-SA2$^{1217-1231}$ was performed by the addition of 10 nM PLK1 kinase WT or mutants, to 200 nM substrate, in 60 μL of kinase buffer (50 mM HEPES 7.5, 150 mM

NaCl, 5 mM MgCl$_2$, 1 mM TCEP, 2.5 mM ATP) incubated for the timepoints indicated in the figures, at 30 °C. The reaction was stopped by the addition of 5X Laemmli sample buffer.

Purified phospho-BUB1 (MBP-Bub1$^{569-616}$-His) was generated by thawing 10 mg of pure protein (diluted at a 500 µM concentration) and incubating it with 2.5 µM CDK1/CyclinB1 in kinase buffer for 15 h at 10 °C. Phosphorylated BUB1 was loaded on a pre-equilibrated Superdex 10/300 200 pg sizing column (Cytiva), concentrated, aliquoted, and snap-frozen for subsequent use in binding assays.

Purified phospho-Bora (MBP-Bora$^{1-224}$-His) was generated by thawing approximately 400 µg of pure protein (100 µM) and incubating it with 1 µM CDK1/CyclinB1 in kinase buffer (50 mM HEPES 7.5, 150 mM NaCl, 5 mM MgCl$_2$, 1 mM TCEP, 2.5 mM ATP) for 15 h at 10 °C. Phosphorylated Bora was loaded on a pre-equilibrated Superdex 5/150 200 pg sizing column (Cytiva), concentrated, aliquoted and snap-frozen for subsequent use in the experiment of Fig. EV1E,F.

Dephosphorylation assays. PLK1$^{wt}$ or PLK1$^{K208R}$ was pre-phosphorylated for 15 h at 10 °C using the kinase buffer described above and the following protein concentration: 400 nM PLK1, 400 nM Aurora A, 800 nM Bora$^{1-224}$. Phospho-PLK1 was diluted 1:2 in a reaction mixture containing 50, 100, 250, 500, 1000 nM phosphatase (PP2A-B56γ, PP2A-B55δ, PP1γ, λ-PPP) in phosphatase buffer (50 mM HEPES 7.5, 150 mM NaCl, 1 mM TCEP; 1 mM MnCl2 was added only in the case of PP1 and λ-PPP because it is needed to promote dephosphorylation). A final reaction volume of 60 µL was incubated for 1 h at 30 °C. The reaction was stopped by the addition of 5X Laemmli sample buffer.

Purified dephosphorylated His-Aurora A was generated by thawing approximately 150 µg of pure protein (60 µM) and incubating it with 1 µM of λ-PP in phosphatase buffer (50 mM HEPES 7.5, 150 mM NaCl, 1 mM TCEP; 1 mM MnCl2) for 15 h at 10 °C. Dephosphorylated Aurora A was loaded on a pre-equilibrated Superdex 5/150 200 pg sizing column (Cytiva), concentrated, aliquoted, and snap-frozen for subsequent use in the experiment of Fig. EV1E,F.

Analytical SEC binding assays. Analytical SEC was performed on a Superdex 5/150 Increase 200 pg sizing column (Cytiva), mounted on an ÄKTA™ micro system (Cytiva). In total, 50 µL samples at 5 µM single protein concentration have been incubated for 1 h on ice, then centrifuged at 16,900× *g* for 20 min. After sample injection, the protein absorbance was monitored at 280 nm, and the proteins were eluted under isocratic condition, at a flow of 0.15 mL/min, in 100 µl fractions at 4 °C. Fractions were analyzed by SDS–PAGE and Coomassie Blue staining. Fractions from 1 mL elution volume to 2.3 mL have been loaded on SDS–PAGE gels, subsequently stained with Coomassie brilliant blue (produced in-house).

Pull-down/immunoprecipitation assays. Hela FlpIn T-REx cells expressing mNG-PLK1 WT/R95A/E121A or mCherry-PLK1 WT were seeded, and after 24 h, if stated in figure legends, RNAi depletion of Bora was performed by using 20 (Fig. 5A,B) or 50 nM (Fig. 3B) oligo and RNAiMAX reagent (Thermo Fisher Scientific). If stated in figure legends, after 19 h, a transient transfection of mNG-Bora WT/W58A/F56A was performed by adding 1 ng/µL of pCDNA5 vector containing the Bora gene and Lipofectamine 2000

(Thermo Fisher Scientific). After 8 h from transfection (27 h from depletion), cells media was supplemented with Nocodazole 3.3 µM (Fig. 5A,B) or 0.33 µM (Fig. 3B). After 24 h, cells were harvested by centrifugation, washed once with 1 mL with PBS, and snap-frozen to be stored at -80 °C until pull-down experiment.

Cells were thawed the day of the experiment, resuspended in IP buffer (50 mM Hepes 7.5, 150 mM NaCl, 5% glycerol, 1 mM TCEP, 0.1% NP-40, 1 phosSTOP tablet/10 mL IP buffer, DNAase from Roche, Protease Inhibitors from Serva) and 1.5–3 mg of total protein (clarified cell lysate) was mixed with pre-equilibrated 20 µL slurry anti-mNeonGreen magnetic beads or Red Fluorescent Protein magnetic beads (for mCherry-tagged PLK1) and incubated 3 h at 4 °C. Input sample was collected right before beads were washed twice with 500 µL of IP buffer and elution was performed by the addition 5× Laemmli sample buffer. Samples were boiled for 5 min at 95 °C.

Structural modeling. Alpha-Fold prediction of Aurora A:Bora shown in Figs. 1B,C and EV1A was performed using AlphaFold version 3, while the model in Figs. 2A and EV2A was predicted using AlphaFold version 2.3.1. All figures for structural models were created using PyMOL v2.5.2 (Schrödinger, LLC). Uniprot codes of the proteins used in the predictions: Aurora A (O14965), Aurora B (Q96GD4), CEP192 (Q8TEP8), TPX2 (Q9ULW0), INCENP (Q9NQS7), Bora (Q6PGQ7). Each protein subunit was shown in all figures using the following colors: PLK1 Kinase Domain = dark blue; Polo-box domain = light blue; Aurora A and B = pale green; Bora = coral red.

Western blotting. The samples from in vitro phosphorylation assays and cell lysates were loaded on SDS–PAGE gels (produced in-house) and transferred onto a nitrocellulose membrane (Bio-Rad) via the Trans-Blot system (Bio-Rad) with pre-defined transfer protocols. The blotted membrane was blocked with either 5% milk or 5% BSA, and probed with the antibodies listed in the Reagents and Tools Table, diluted in TBS-T (Tris-buffered saline with 0.1% Tween-20) as indicated. After 18-hour incubation at 4 °C, the membrane was washed with 10 mL TBS-T three times for ten minutes and mixed with secondary antibodies (diluted at 1:10,000 ratio in TBS-T, or 1:5000 for Bora antibody) for 1 h at 20 °C. As secondary antibodies, anti-mouse or anti-rabbit (NXA931 and NA934; Amersham, 1:5000) conjugated to horseradish peroxidase were used (in the experiment in Fig. EV5, an anti-mouse IgG Alexa 647-coupled secondary antibody from Invitrogen was used, in a 1:5000 dilution). After incubation with ECL western blotting reagent (Cytiva), images were acquired with the ChemiDoc MP System (Bio-Rad) and analyzed using Image Lab 6.0.1 software.

ProQ diamond staining. Samples from in vitro phosphorylation assays were separated by SDS–PAGE (gels produced in-house). Following electrophoresis, gels were fixed in fixation solution (50% methanol, 10% glacial acetic acid in ddH$_2$O) for two rounds of incubation: 30 min and 16 h at room temperature under gentle rotation. The gels were then washed three times with water for ~10–20 min each. Subsequently, Pro-Q Diamond phosphoprotein staining solution (ThermoFisher Scientific) was applied for 1.5 h at room temperature under rotation, followed by three washes in destaining solution (50 mM sodium acetate, pH 4.0, 20% acetonitrile in ddH$_2$O) for 30 min each at room temperature. After two brief

rinses in water, fluorescence images were acquired using a ChemiDoc MP Imaging System (Bio-Rad) and processed with Image Lab software. The same gel was then restained with Coomassie Brilliant Blue.

Cell culture. HeLa Flp-In T-REx cell lines were maintained in Dulbecco's Modified Eagle's Medium (DMEM) supplemented with 10% tetracycline-free FBS, 50 µg/ml Penicillin/Streptomycin, and 2 mM L-glutamine (all reagents/media were from PAN Biotech).

Flp-In T-REx HeLa expression cell lines for PLK1 and Bora were generated and maintained as previously described (Krenn et al, 2012). DNA fragments encoding mNeonGreen-PLK1 or mNeonGreen-Bora were sub-cloned into a pCDNA5/FRT/TO-EGFP-IRES plasmid and co-transfected with pOG44 (Invitrogen), encoding the Flp recombinase, into cells using X-tremeGENE (Roche) according to the manufacturer's instructions. Subsequently, cells were selected for 2 weeks in DMEM supplemented with hygromycin B (250 µg/ml; Thermo Fisher Scientific) and blasticidin (4 µg/ml; Thermo Fisher Scientific). Single-cell colonies were isolated, expanded, and transgene expression was induced by the addition of 300 ng/ml doxycycline (Sigma-Aldrich) and checked by immunofluorescence microscopy and immunoblotting. Transgene expression during experiments was induced by the addition of 50 ng/ml doxycycline (Sigma-Aldrich) for 24 h.

To achieve single or double depletion of PLK1 and Bora, cells were transfected with RNAiMax 1:25 ratio in Optimem (Invitrogen), Bora siRNA(5′-TAACTAGTCCTTCGCCTATTT-3) (reference (Chan et al, 2008)) and PLK1 siRNA (5′-AUACCUUGUUAGUGGG-CAATT-3′) at 20 nM and 50 nM respectively for 48-72 h (see experimental scheme of the individual experiments in the figures) following the manufacturer's protocol. To perform transient transfection of mNeongreen-Bora WT, W58A, and F56A, HeLa FRT cells were supplemented with 1 ng/µL pCDNA5 mNeongreen-Bora plasmid and Lipofectamine 2000 1:50 ratio in Optimem. After a 5-min incubation of the Lipofectamine 2000 in Optimem, DNA and the reagent were mixed 1:1 and incubated for 20 min at 20 °C. A list of the cell lines used and the siRNA oligos can be found in the Reagents and Tools Table.

Live-cell imaging. For time-lapse experiments, 60,000 cells were seeded into each well of a µ-Slide 8-well microscopy chamber (Ibidi). The proteins of interest were knocked down during seeding by reverse siRNA transfection. 24 h after seeding, cells were exposed to 2 mM Thymidine (Sigma-Aldrich) and synchronized for 16 h. The following morning, cells were released from the first Thymidine block into media containing 50 ng/ml doxycycline (Sigma-Aldrich). After 8-10 h, cells were exposed to 2 mM Thymidine and 50 ng/ml doxycycline for 20 h. The following morning, cells were released in L15 supplemented with 5% FBS, including 300 nM of BI2536 (Focus Biomolecules), if required by the experiment. Five hours after release, cells were imaged at 37 °C every 20 min for 18 h (for a total tracking of 23 h) on a DeltaVision Elite (GE Healthcare) deconvolution microscope, equipped with an IX71 inverted microscope (Olympus, Japan), a PLAPON x60/1.42NA oil objective (Olympus) and a pco.edge sCMOS camera (PCO-ECH Inc., USA). For each field of view, 3 z-sections spaced at 6 µm each were taken. Fluorescence detected in the 488 nm channel was used to assess transgene expression (mNeonGreen PLK1 or Bora), while the 647 nm channel was employed to visualize cells' DNA via SiRDNA staining

(Spirochrome). A brightfield image was acquired in parallel. Images were converted into maximal intensity projections in SoftWoRx (Cytiva) for analysis. For each experimental repeat, at least 50 cells were tracked. Quantification was performed manually using the software Fiji (Schindelin et al, 2012). Cell fluorescence intensity (as the one measured and plotted in Fig. EV6E) was measured by circling precisely each cell of interest and measuring the integrated density. The background was subtracted by selecting areas of the image where cells were not present and obtaining an average background intensity. Measurements were exported in Excel (Microsoft) and plotted with GraphPad Prism 10 (GraphPad Software). Figures were assembled using Adobe Illustrator 2025.

Data quantifications. Dot plots allow the data distribution to be visualized along with the 95% confidence intervals (thick vertical bars) calculated around the mean (thin horizontal lines in the relevant plots). The measurements were first cleaned of outliers using Prism 8 (GraphPad), using ROUT analysis with a cut-off of Q at 2%. Plots were produced using GraphPad Prism 10 (GraphPad Software). Statistical comparisons were performed in Prism 10. All cell biology experiments were repeated at least 3 times to attain statistical significance. We used the Kruskal–Wallis test coupled with Dunn's multiple comparisons for experiments comparing 3 or more conditions. To convert $P$ values into an asterisk-based significance system, we used the default GraphPad Prism convention: not significant (ns) = $P > 0.05$; $*P \leq 0.05$ but $>0.01$; $**P \leq 0.01$ but $>0.001$; $***P \leq 0.001$ but $>0.0001$; and $****P \leq 0.0001$.

Bar plots of signal intensity were generated using ImageJ Fiji (Schindelin et al, 2012) quantification of signal of the raw image (integrated density), acquired with a Bio-Rad ChemiDoc MP Imaging System (.scn files), after using the background subtraction function of Fiji. In more detail, a rectangular selection (region of interest, ROI) with constant area comprising the whole signal of each band was used to measure the integrated density value of each band. Integrated density value of a background ROI (within the membrane but outside the observable signal) was also measured for background subtraction. Normalization of the integrated density value against total substrate or against total kinase was used in case of pull-down from cell lysates or in cases in which there was significant variation of substrate or kinase amount between conditions, and the amount of signal was always calculated as relative signal to the reference condition (set to 1). The final value represents the arithmetic mean between two, three or four replicates, as described in the figure legends. The dots in the bar plots indicate the normalized, relative integrated densities of single biological replicates. The error bars (available for pull-down from biological samples) represent the standard deviation of the triplicate, as also described in the figure legends.

Protein conservation and sequence alignment were analyzed and visualized using Jalview (Waterhouse et al, 2009). All protein sequences for Figs. 2B and EV2D,E used in the alignment were obtained from UniProt (UniProt, 2023). Sequence Logo were generated by creating a fasta file of the T-loop region around the phospho-threonine of Aurora A (T288), Aurora B (T232), and PLK1 (T210) from all the species annotated on TreeFam (Li et al, 2006). For PLK1, all the entries annotated as other isoforms of PLK (isoform 2, 3) were removed. The resulting file was loaded on the WEBLOGO server from the University of Berkeley (Crooks et al, 2004; Schneider and Stephens, 1990).

## Data availability

The live cell imaging data from this publication have been deposited to the BioImage Archive database (https://www.ebi.ac.uk/bioimage-archive) and assigned the identifier S-BIAD2516.

The source data of this paper are collected in the following database record: biostudies:S-SCDT-10_1038-S44318-025-00681-0.

## Peer review information

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

## Acknowledgements

We thank Patricia Stege, Doro Vogt, and Tiziano Crocilla for their support in cloning, insect cell, bacterial expression, and protein purification. We thank Marius Hedtfeld for sharing the Halo-Aurora B:INCENP reagent; Jan Michael Peters and Gabriele Litos for sharing the antibody against pSA2 1224; Carlos Conde for sharing active His-PP1; Andrew Carter for sharing protocols and reagents related to protein expression in Expi293F cells; the Biotechnology facility of our institute for generating competent cells, providing lambda-phosphatase, and reagents for protein expression and buffer preparation; Stefano Maffini and Nico Schmidt for support with light microscopy; Franca Rossi for helpful comments on the manuscript. We are very grateful to Anaïs Pillan and Lionel Pintard for sharing results in advance of publication. This work was supported by the Max Planck Society (AMu), the European Research Council (synergy grant 951430 BIOMECANET to AMu), the German Research Foundation (DGF Collaborative Research Centre grant 1430 "Molecular Mechanisms of Cell State Transitions" to AMu), the CANTAR network under the Netzwerke-NRW program (AMu), and an EMBO Long-Term Fellowship ALTF 439-2019 (DC).

## Author contributions

**Arianna Esposito-Verza**: Conceptualization; Supervision; Validation; Investigation; Visualization; Writing—original draft; Project administration; Writing—review and editing. **Duccio Conti**: Investigation; Writing—review and editing. **Paulo D Rodrigues Pedroso**: Investigation. **Lina Oberste-Lehn**: Investigation. **Carolin Koerner**: Investigation. **Sabine Wohlgemuth**: Investigation. **Artem Mansurkhodzhaev**: Investigation; Writing—review and editing. **Ingrid R Vetter**: Investigation. **Marion E Pesenti**: Resources; Writing—review and editing. **Andrea Musacchio**: Conceptualization; Supervision; Funding acquisition; Validation; Visualization; Writing—original draft; Project administration; Writing—review and editing.

Source data underlying figure panels in this paper may have individual authorship assigned. Where available, figure panel/source data authorship is listed in the following database record: biostudies:S-SCDT-10_1038-S44318-025-00681-0.

## Funding

## Disclosure and competing interests statement

The authors declare no competing interests.

# Expanded View Figures

**Figure EV1.   Aurora A binds to its cofactors in vitro.**

(A) pLDDT scores of the AlphaFold 3 simulation of the Aurora A:Bora complex (Fig. 1B) mapped onto its coordinates. (B) Predicted alignment error (PAE) for the model prediction in (D). (C) Schematic representation of the Aurora A protein cofactors used in the experiment. Bora, TPX2 and CEP192 are N-terminally fused to MBP, as described in more detail in "Methods". Aurora A binding sites of the cofactors are indicated with a green box. (D) Analytical size-exclusion chromatography of recombinant human Aurora A (5 μM) mixed with equimolar amounts of recombinant human Bora, TPX2 and CEP192 proteins used throughout the study. (E) SDS–PAGE and corresponding Western blots with the indicated antibodies of phosphorylated Aurora A (p-AurA), dephosphorylated Aurora A (dep-AurA), or Aurora A T288V (AurA T288V) in the indicated conditions, including addition of unphosphorylated Bora, kinase dead PLK1 as substrate, and ATP where indicated. Phosphorylation was performed for 60 min at 30 °C. (F) The experiment in (E) was also repeated with CDK1-phosphorylated Bora (pBora$^{CDK1}$). Source data are available online for this figure.

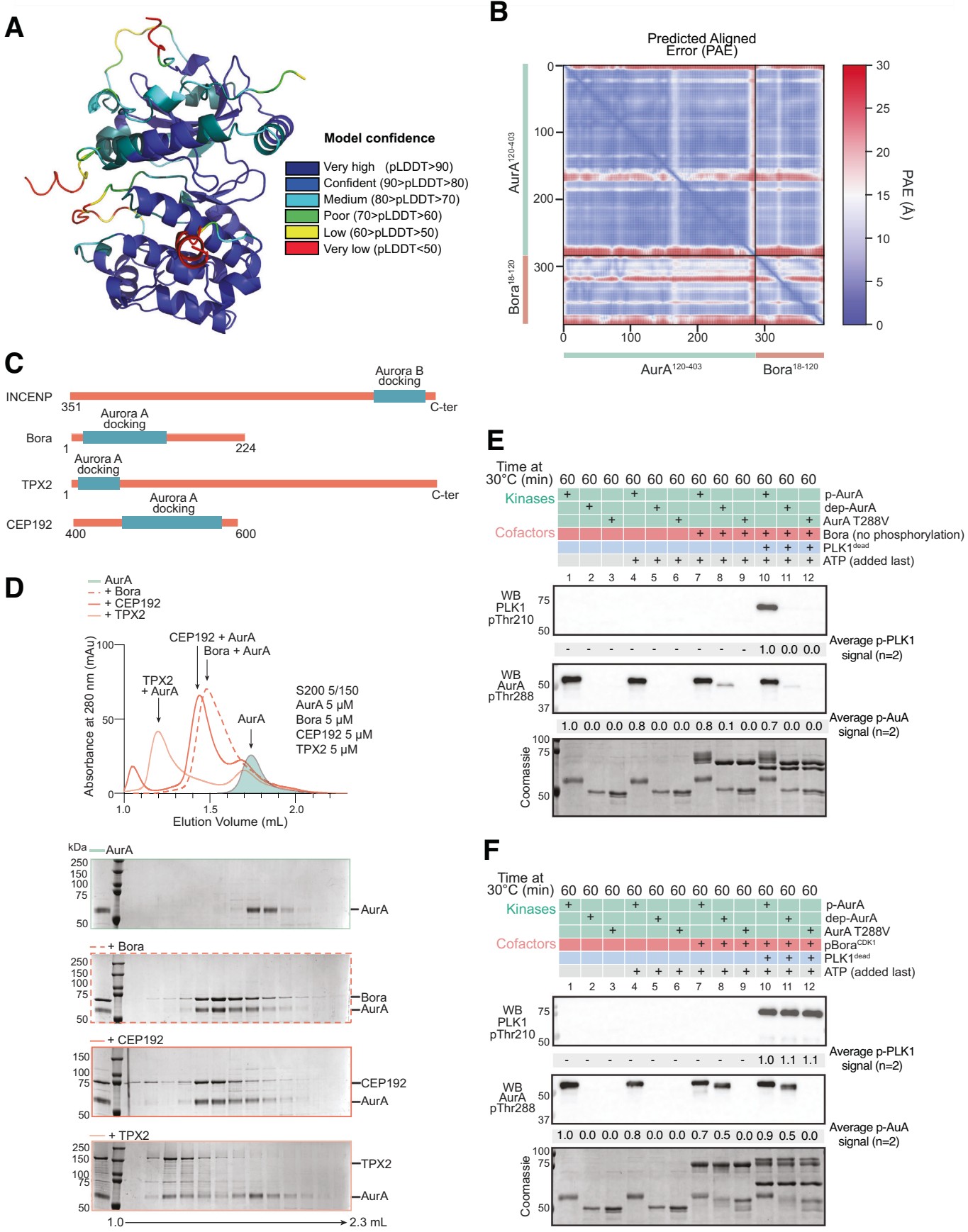

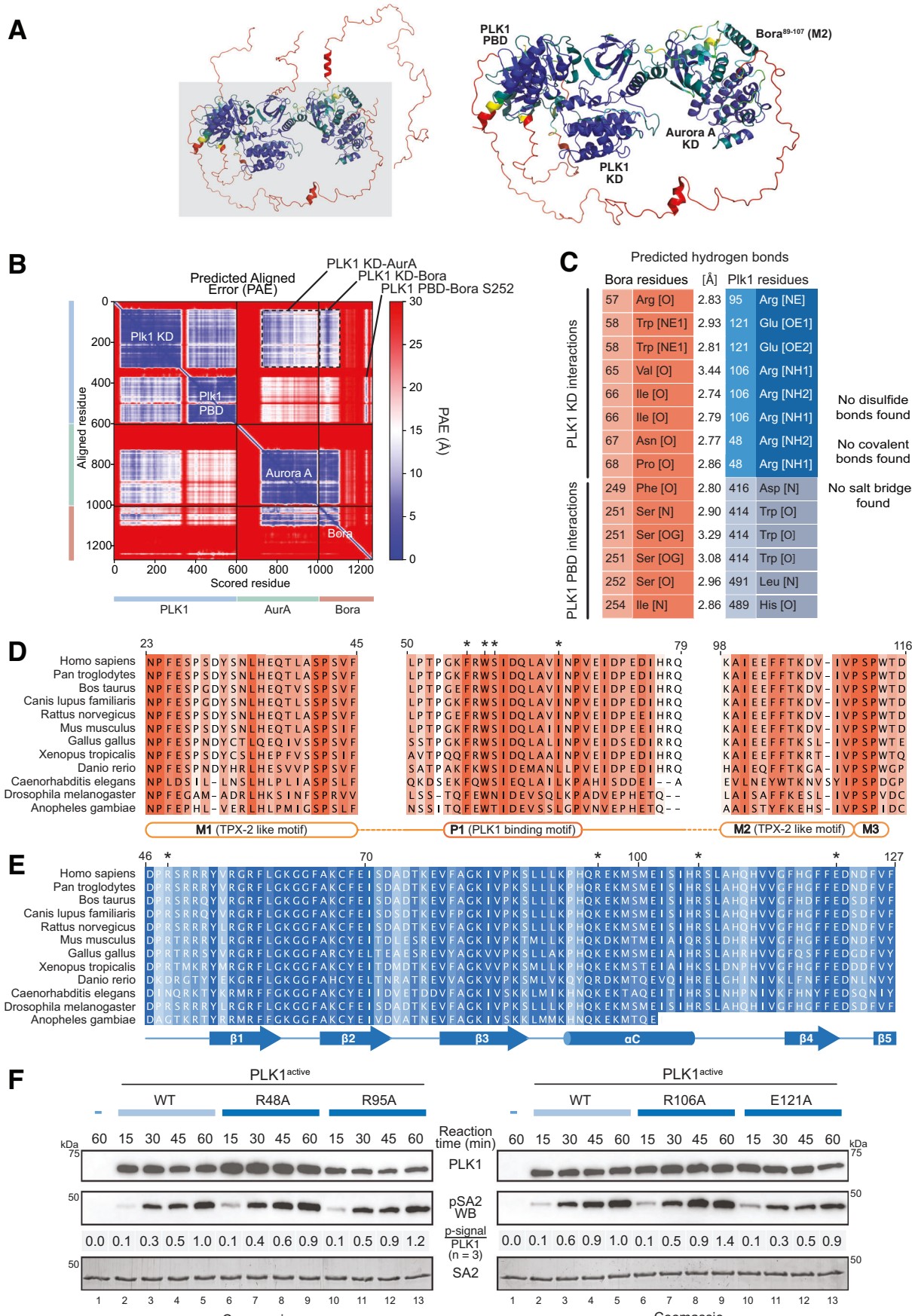

◄ **Figure EV2.  PLK1 WT and P1 interface mutants: activation and activity on substrate.**

(A) pLDDT scores of the simulation of the Aurora A:Bora[18-280]:PLK1 AlphaFold2 model shown in Fig. 2A. (B) Predicted Aligned Error (PAE) plot for the model in (A). (C) List of interactions detected in the AlphaFold2 model, by using the PDBePisa server (Krissinel and Henrick, 2007). Atom nomenclature follows standard conventions: atom names beginning with C denote carbon atoms, N nitrogen, and O oxygen. For amino acid atoms, the character following the element symbol is the remoteness indicator, transliterated as follows: α = A, β = B, γ = G, δ = D, ε = E, ζ = Z, η = H. If present, the subsequent character denotes a branch indicator. (D) Sequence alignment across species of the regions of Bora involved in Aurora A binding (indicated as M1, M2 and M3) and PLK1 binding (P1). Residues selected for mutagenesis and biochemical validation are marked with an asterisk. The extent of sequence conservation is rendered in different colors, ranging from white (no conservation) to darkest orange (full conservation). (E) Sequence alignment across species of the regions of PLK1 involved in the binding with the P1 motif of Bora. Residues selected for mutagenesis and biochemical validation are marked with an asterisk. The extent of sequence conservation is rendered in different colors, ranging from white (no conservation) to darkest blue (full conservation). (F) PLK1[WT] and mutants time-course activity assay on SA2 1217-1231 ($n = 3$), containing the PLK1 target site Ser1224. Source data are available online for this figure.

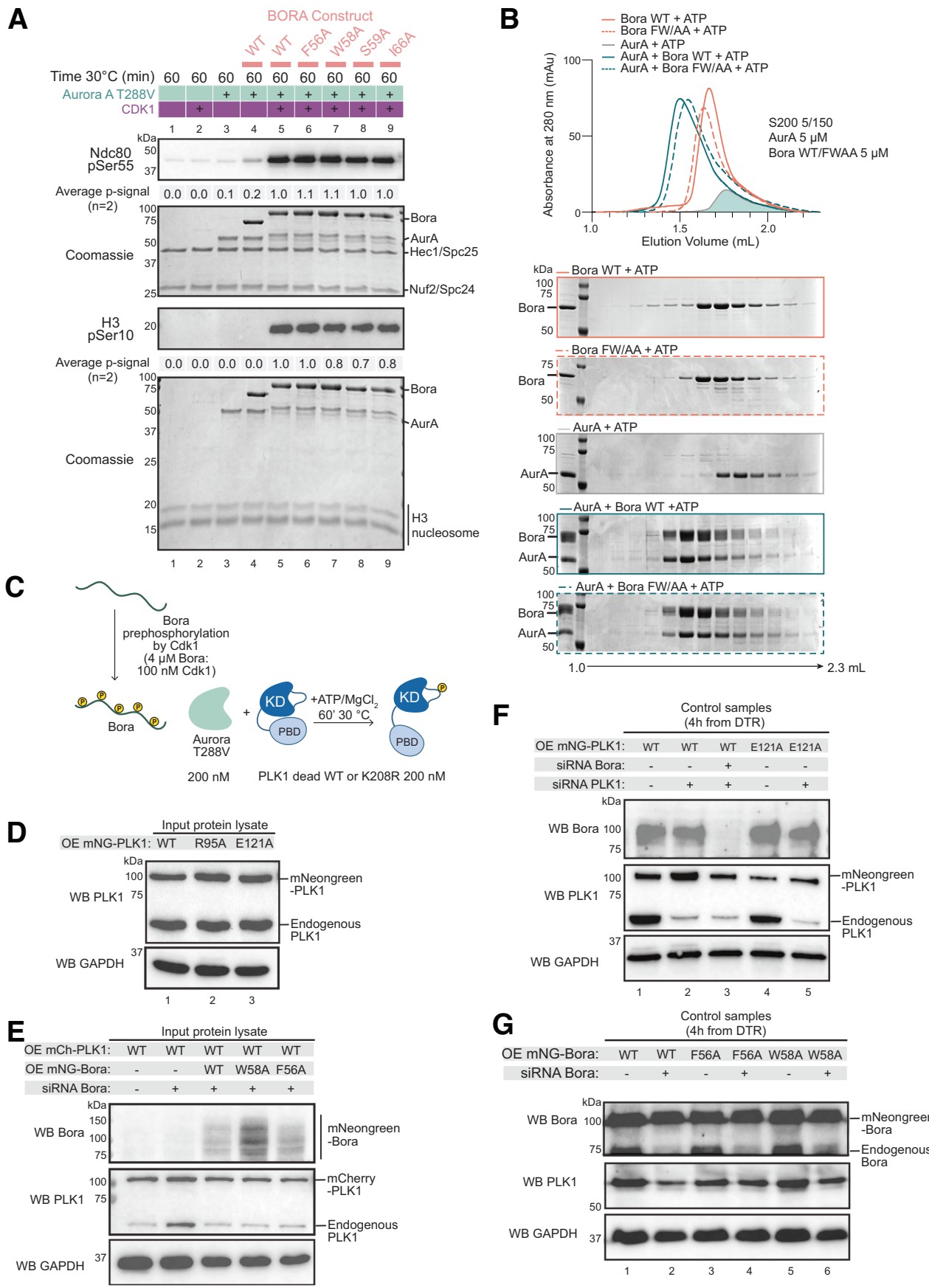

**Figure EV3.  Binding of Aurora A to Bora P1 mutants.**

(**A**) Phosphorylation of Ndc80 bonsai and H3 nucleosome histone tail by Aurora kinase A T288V was monitored in presence or absence of Bora and the P1 motif mutants (pre-phosphorylated with CDK1). (**B**) Size-exclusion chromatography-based binding assay showing that Bora[WT] or Bora[-F56A-W58A] bind Aurora A identically at equimolar concentrations (set to 5 μM). (**C**) Schematic representation of the reaction protocol used in Fig. 2F and  EV3A. (**D**) Total protein lysate input of the IP experiment shown in Fig. 3A. (**E**) Total protein lysate input of the IP experiment shown in Fig. 3B. (**F**) Control samples of the live-cell imaging experiment shown in Fig. 3D. (**G**) Control samples of the live-cell imaging experiment shown in Fig. 3E. Source data are available online for this figure.

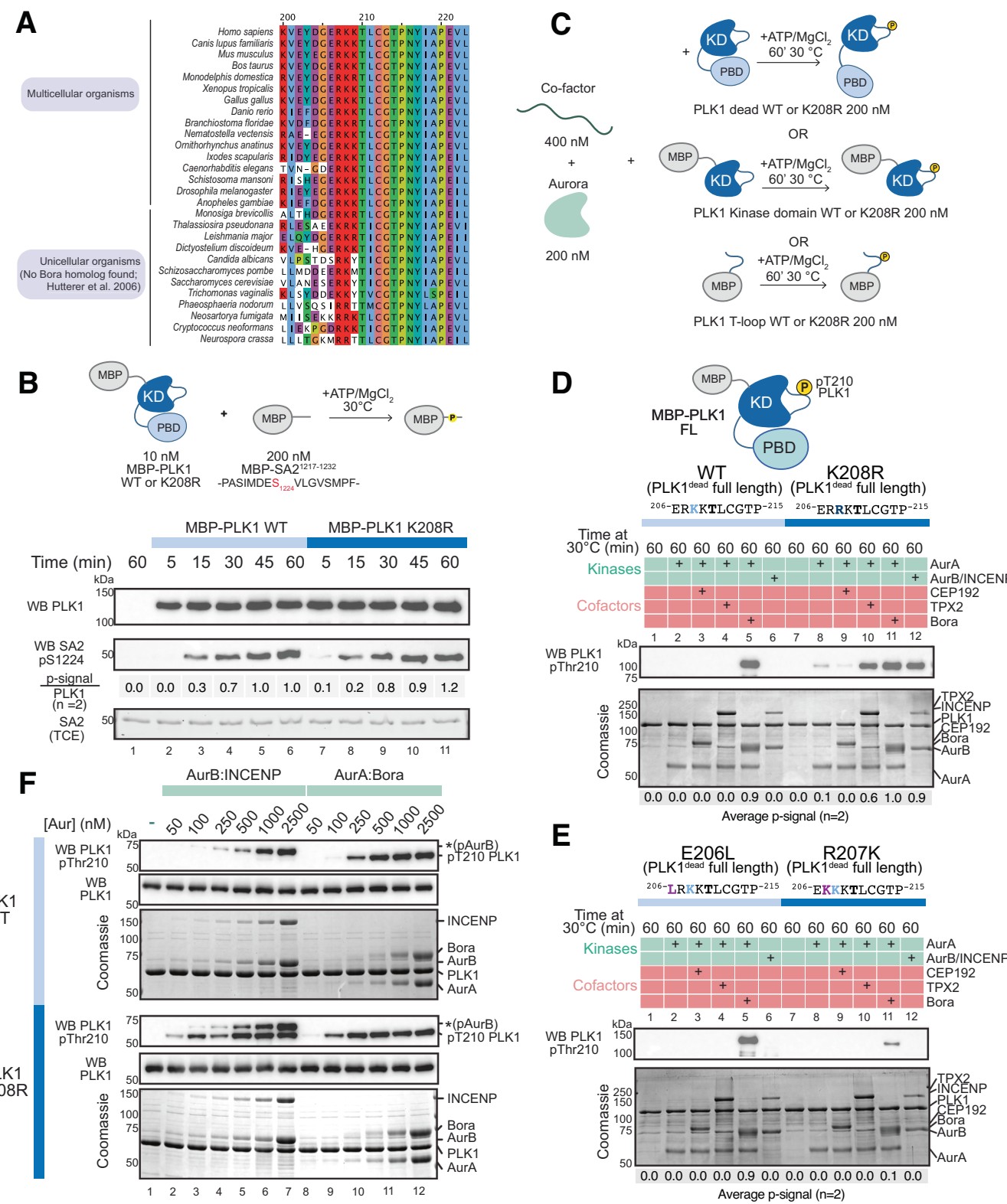

**Figure EV4.  PLK1 WT and K208R activation and activity on substrate.**

(A) Sequence alignment across species of the T-loop region around PLK1 Thr210. Alignment was downloaded from the Proviz web server (Jehl et al, 2016). (B) MBP-PLK1$^{WT}$ or MBP-PLK1$^{K208R}$ were assayed in parallel for activity towards an SA2 peptide containing Ser1224. (C) Schematic representation of the reaction protocol used in Fig. 4A–C. (D) Kinase dead PLK1$^{WT}$ and PLK1$^{K208R}$ were assayed in parallel for activation with the same Aurora complexes. A western blot is shown to assess relative phosphorylation of the substrates, quantified with Aurora A:Bora activity on PLK1$^{K208R}$ as reference (equal to 1.0). The arithmetic mean of two independent experiments is shown. A Coomassie Brilliant Blue-stained gel of the same samples was combined to verify equal protein loading between conditions. (E) Kinase dead PLK1$^{E206L}$ and PLK1$^{R207K}$ were assayed in parallel as in (D). (F) Phosphorylation of PLK1$^{WT}$ or PLK1$^{K208R}$ by increasing amounts of Aurora A:Bora or Aurora B:INCENP. The phosphorylation after 30 min at 30 °C was visualized by use of phospho-specific antibodies by western blotting, while the protein content of each sample is shown on Coomassie Blue-stained SDS–PAGE gels of the same samples. The experiment was repeated twice. Source data are available online for this figure.

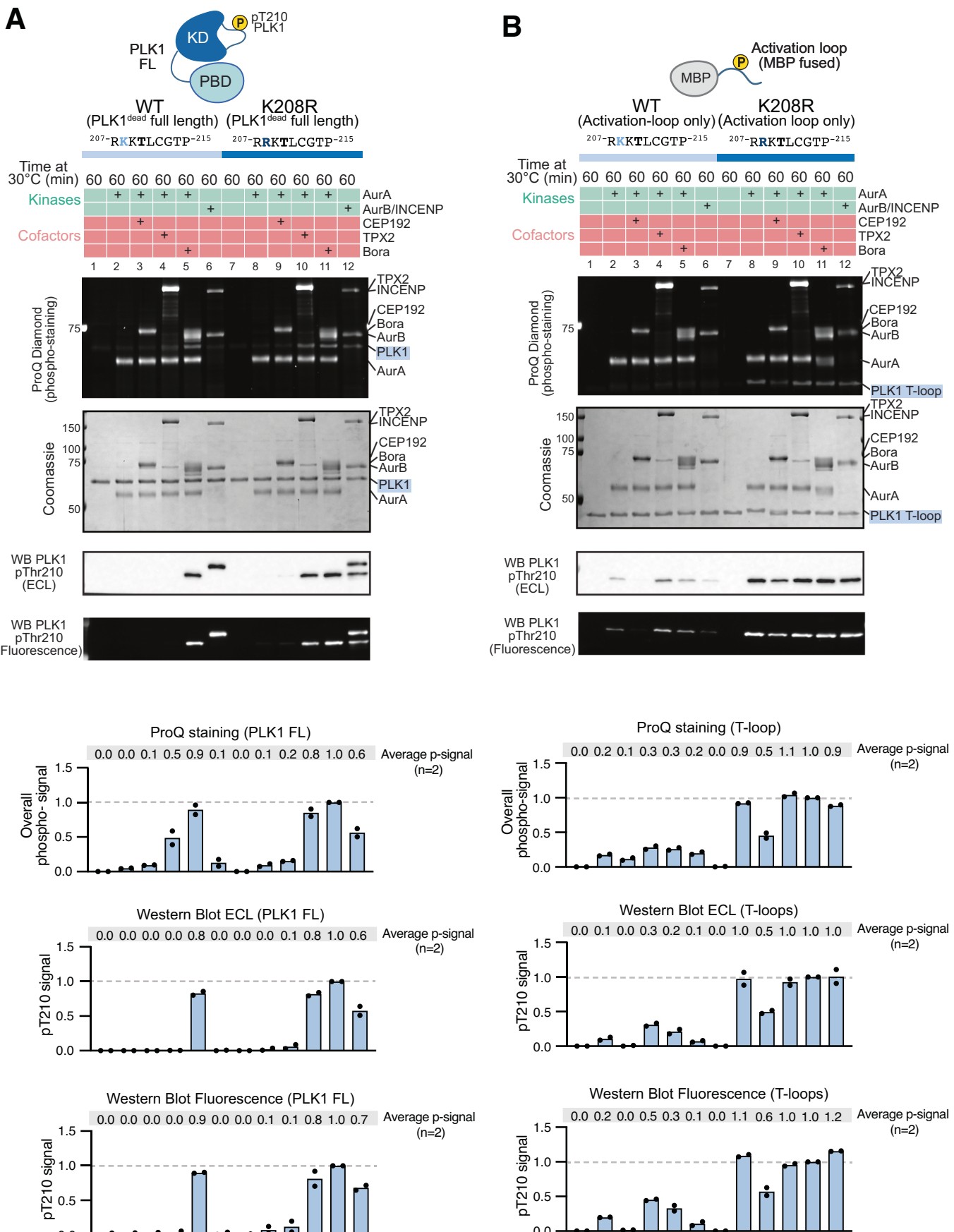

◀    **Figure EV5.   Orthogonal validation of detection of PLK1[K208] mutant phosphorylation.**

(**A**) PLK1[WT] and PLK1[K208R] were assayed in parallel for activation with the same indicated Aurora complexes. A Coomassie Brilliant Blue-stained gel of the samples is shown to verify equal protein loading between conditions. ProQ[TM] Diamond detects phosphorylation. Western blots against pT210 of PLK1 were developed by enhanced chemiluminescence (ECL) or with fluorescent antibodies and the results quantified from two technical repeats. (**B**) As in (**A**), but with the isolated activation segments of PLK1 in their wild-type or K208R versions. Source data are available online for this figure.

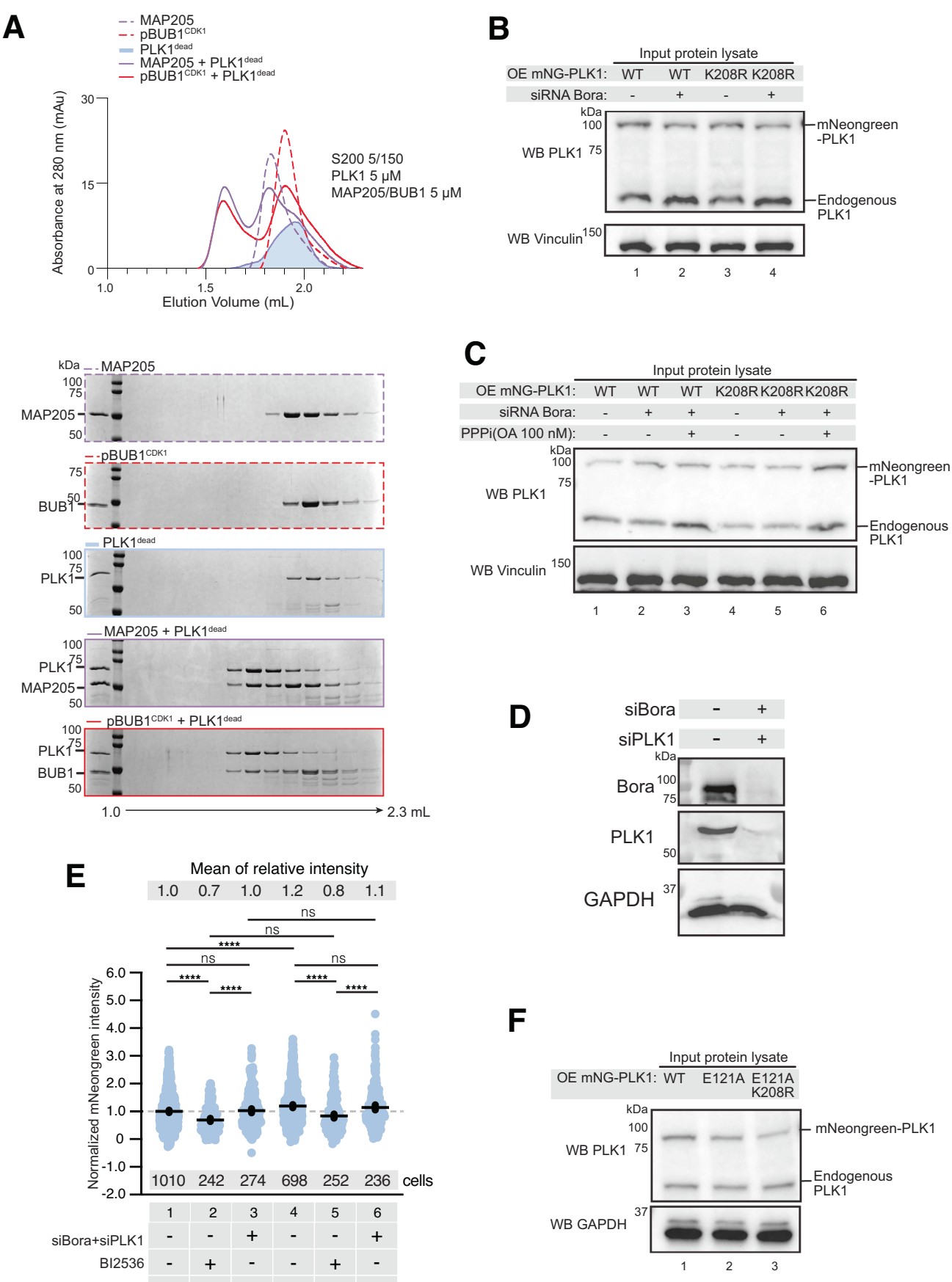

◀  **Figure EV6.   MAP205 and pBUB1 binding to PLK1.**

(**A**) Analytical size-exclusion chromatography of recombinant human dead PLK1 (5 μM) mixed with equimolar amounts of recombinant MAP205 and BUB1 peptides, also used in the activation assay in Fig. 4D. (**B**) Total protein lysate input of the IP experiment shown in Fig. 5A. (**C**) Total protein lysate input of the IP experiment shown in Fig. 5B. (**D**) Depletion control for Bora and PLK1 related to the live-cell imaging experiment in Fig. 5C. (**E**) Mean mNeonGreen-PLK1 expression levels in the samples in Fig. 5C were quantified by recording fluorescence signal intensity during the live-cell imaging experiment. Statistical comparison was performed as indicated in the legend for Fig. 3D,E from three independent experiments. The exact $P$ values were: 1–2 < 0.0001; 1–3 < 0.9999; 1–4 < 0.0001; 2–3 < 0.0001; 2–5 = 0.3385; 3–6 > 0.9999; 4–5 < 0.0001; 4–6 > 0.9999; 5–6 < 0.0001. (**F**) Total protein lysate input of the IP experiment shown in Fig. 5E. Source data are available online for this figure.

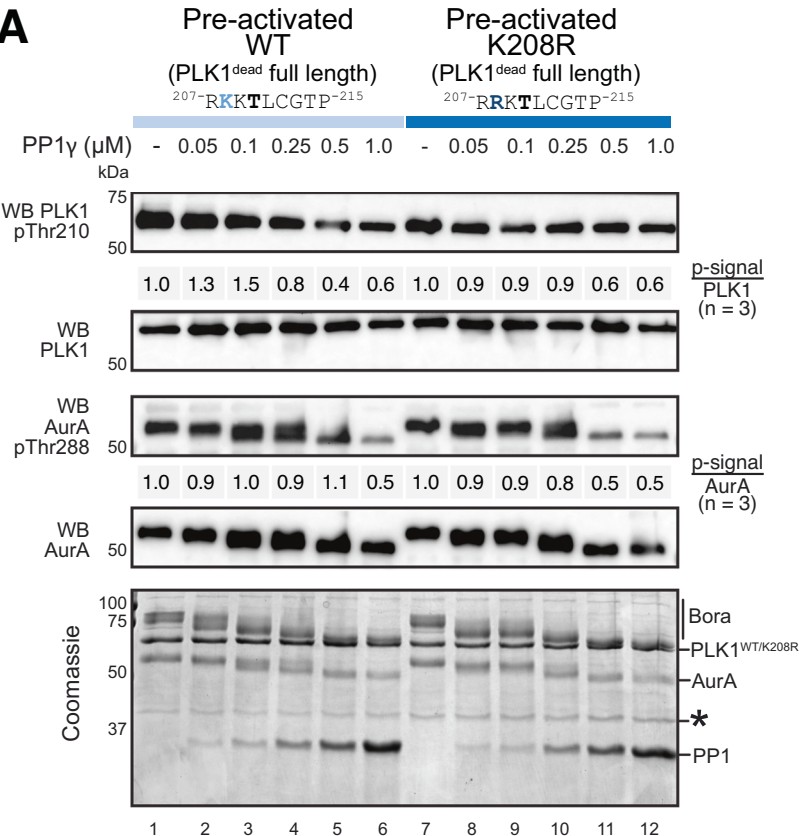

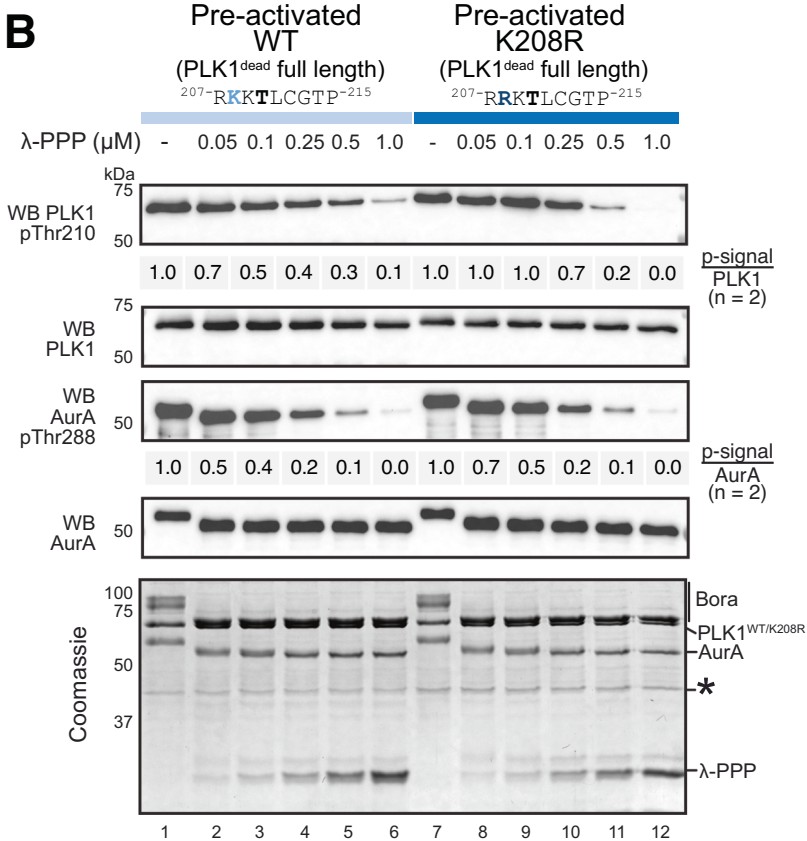

◀ **Figure EV7. PLK1 WT and KR dephosphorylation by phosphatases.**

(A) PP1 -mediated dephosphorylation of PLK1$^{WT}$ or PLK1$^{K208R}$ mutant in a time-course reaction in vitro. Aurora A activation-loop dephosphorylation was also monitored as control. Aurora A and PLK1 blots, together with a Coomassie Blue-stained gel were included to verify equal protein loading. (B) λ-PPP-mediated dephosphorylation of PLK1$^{WT}$ or PLK1$^{K208R}$ mutant in a time-course reaction in vitro. Aurora A activation-loop dephosphorylation was also monitored as control. Aurora A and PLK1 blots, together with a Coomassie Blue-stained gel were included to verify equal protein loading. Source data are available online for this figure.

