## [Peer Review File · The EMBO Journal]

Molecular requirements for PLK1 activation by T-loop phosphorylation

Arianna Esposito-Verza, Duccio Conti, Paulo Rodriguez Pedroso, Lina Oberste-Lehn, Carolin Koerner, Sabine Wohlgemuth, Artem Mansurkhodzhaev, Ingrid Vetter, Marion Pesenti, and Andrea Musacchio

Corresponding authors: Andrea Musacchio (andrea.musacchio@mpi-dortmund.mpg.de) , Arianna Esposito-Verza (arianna.espositoverza@mpi-dortmund.mpg.de)

Review Timeline:

Transferred from Review Commons:	2nd Nov 25
Editorial Decision:	20th Nov 25
Revision Received:	18th Dec 25
Accepted:	19th Dec 25

Editor: Hartmut Vodermaier

Transaction Report:

This manuscript was transferred to The EMBO Journal following peer review at Review Commons.

Review #1**1. Evidence, reproducibility and clarity:****Evidence, reproducibility and clarity (Required)**

The study from Verza and colleagues carefully evaluates the long-sought mechanism by which Aurora A controls PLK1 activity, and helps explain why BORA binds to both Aurora A and PLK1, and especially why T210 is not a remotely efficient Aurora A substrate in the absence of other factors. PLK1 does not autoactivate in the presence of Aurora A/Bora, . Instead, the authors find that in the presence of BORA, substrate inefficiency is overcome, allowing Aurora A (but not Aurora B, or any of likely the vast number of other basophilic kinases) to activate PLK1 through T210 phosphorylation. Mechanistically, and guided by the remarkable powers of AF modelling using AF Multimer and analysis with purified components, Thr210 phosphorylation by Aurora A is shown to be prevented by mutation of Lys208 to Arg (a subtle, but conserved PLK1 change in the upstream consensus), suggesting a relevant interface for BORA that assembles a holoenzyme capable of guiding Aurora A into a conformation relevant for PLK1 activation via T210 phosphorylation. BORA pre-phosphorylation by CDK overcomes mutational effects, suggesting that this is the relevant 'activating' species of BORA. Findings are then tested in cells, with data (notably BoraF56A or BoraW58A mutants that cannot employ the 'Y8/Y10 TPX2' docking motif first characterised for Xenopus TPX2 in PMID: 14701852). Clear and well controlled data suggest that no other mitotic kinase can phosphorylate PLK1WT on Thr210 after Bora depletion. Together, this confirms the hypothesis that pBora and PLK1 interact to 'licence' Aurora A to phosphorylate the PLK1 activation loop, a very nice explanation of years of work in various systems.

****Referees cross-commenting****

We all agree this should be published, essentially as is.

2. Significance:**Significance (Required)**

Building on quite recent work by Lorca and colleagues (and a second, complementary study submitted to Review Commons, which is discussed in this study, coming to similar conclusions), this work makes ample use of available reagent sets, whilst generating a number of new ones, and exploits modelling to reveal the likely mechanism of PLK1 activation by Aurora A/BORA. It is a significant study, with particular strengths in revealing why it is Aurora A/BORA **specifically** that controls PLK1, rather than other potential

activities. It will be of interest to all students of the cell cycle, not just kinase aficionados.

Highlights. The K208R mutation is used to great effect, as is AF modelling and convincing (and challenging) sets of cellular experiments; the supplementary work is extremely strong in supporting the main findings. The section on phosphatases is an excellent addition, providing the impetus for future work; the demonstration that PLK1 T210 is comparatively resilient to dephosphorylation by mitotic phosphatases in particular is a nice finding, as is the suggestion that Lys to Arg mutation may modulate the phosphatase interactome.

****Minor points:****

1) T295V Aurora A is a preferred Aurora A mutant here, since it can be used to prove that Aurora A T-loop phosphorylation is not involved, but what happens with WT Aurora A in a side-by-side pBORA/PLK1 experiment (an inactive version of which lacking pT288 can be generated by dephosphorylation?)

2) p.3 with TPX2 led to propose is missing 'them'.

3. How much time do you estimate the authors will need to complete the suggested revisions:

Estimated time to Complete Revisions (Required)

(Decision Recommendation)

Less than 1 month

Yes

Review #2

1. Evidence, reproducibility and clarity:

Evidence, reproducibility and clarity (Required)

In the manuscript by Verza et al., the authors investigated the molecular requirements for activation of Polo-like kinase 1 by phosphorylation of its T-loop. The authors used AlphaFold predictions to model the interaction between PLK1 and Bora and subsequently tested these predictions using in vitro biochemical assays. The main finding is that a transient interaction of Bora with several residues in the catalytic domain of PLK1 is necessary for PLK1 modification at Threonine 210 and subsequently its activation. Interestingly, Lysine 208 in the activation loop of PLK1 turned out to be the major determinant for the requirement of Aurora-A:Bora complex for PLK1 activation and its mutation to arginine allowed for efficient phosphorylation of PLK1 by other Aurora kinases and their cofactors. Nevertheless, this was observed only in vitro, since expression of the K208R mutant failed to rescue PLK1 modification in mitotic cells lacking Bora (Fig. 5A).

****Major comments****

1. The proposed model in which Lys208 contributes to recognition and activation of PLK1 by Aur-A:Bora complex is interesting. Throughout the manuscript, the authors use antibody to detect the phosphorylated T210 residue in the T loop. A potential technical problem is a different reactivity of this antibody with various epitopes determined not only by the phosphorylation itself but largely also by the neighbouring amino acids. The authors should exclude the possibility that K208R mutation in combination with the phosphorylated Thr210 generates a better epitope recognized by the antibody compared to the wild type sequence. To this end, they could use synthetic (phospho)peptides. Alternatively, they could run radioactive assays to prove that the differences in the signal between WT and K208R mutant represent different level of phosphorylation rather than different reactivity of the antibody.
2. Statistical evaluation is missing in experiments comparing the dynamics of mitotic entry (Fig. 3D, 3E, 5C) making the conclusions difficult.

****Minor comments****

1. The following sentence in the abstract is confusing and should be reformulated: We show that Aurora A:Bora phosphorylates Thr210 of PLK1 in vitro, while Aurora A, other Aurora A:activator complexes, and Aurora B:INCENP fail to target T210 even at high kinase/substrate ratios.
2. At several places of the manuscript, authors state that the predictions by Alpha-Fold were of high-confidence but without quantification it remains unclear what that means.

2. Significance:

Significance (Required)

Although the main proteins involved in PLK1 activation at mitotic entry have been already known, the precise molecular mechanism was missing. Original data presented in this manuscript fill the gap in the understanding of how PLK1 is activated by the joined function of the upstream kinase Aurora-A and its cofactor Bora. Therefore, this study will be of interest for the broad cell cycle community.

3. How much time do you estimate the authors will need to complete the suggested revisions:

Estimated time to Complete Revisions (Required)

(Decision Recommendation)

Between 1 and 3 months

Yes

Review #3

1. Evidence, reproducibility and clarity:

Evidence, reproducibility and clarity (Required)

The manuscript explores the mechanism of Plk1 activation by Aurora A-Bora. First, it shows that only Aurora A-Bora can phosphorylate Plk1 on Thr210 (the site that in turn activates Plk1). Other Aurora A-activator complexes (e.g. AurkA-CEP192, AurkA-Tpx2) and Aurora B do not efficiently phosphorylate Plk1. This phosphorylation requires not only activation of Aurora A by Bora, but also a transient, direct interaction between Bora and Plk1 that was identified through structural modeling. The authors experimentally test several sites that are predicted to be involved in this interaction. To this end, they introduce point mutants into cultured mammalian cells and monitor Plk1 activity as well mitotic entry delay (a key Plk1 phenotype). They show that Plk1 has specifically adapted to only be activated by

Aurora A-Bora, due to the presence of Lys208 near the Thr210 site. If the site is mutated, the K208R mutant Plk1 can be efficiently phosphorylated by other Aurora A-activator complexes as well as Aurora B. They further show that this site is necessary to keep Plk1 phosphorylated throughout mitosis as the K208R mutant, although it is easier to be phosphorylated, it is also easier to be dephosphorylated after the degradation of Bora.

Overall, the manuscript presents a large amount of carefully planned and executed experiments. That data shown on the figures are very high quality and fully support the conclusions drawn. Western blots are quantified and quantifications show data from multiple repeats. However, for some experiments an average of only two experiments are shown. More generally, Western blots have been developed using ECL, which is known to provide a linear response only in a rather limited range. It would have increased data quality if the authors used fluorescently labeled antibodies.

****Major comments:****

- If Plk1 K208R mutant can be phosphorylated in vitro by all other kinases besides Bora-Aurora A, why does Bora depletion affect mitotic entry timing in cells with Plk1 K208R mutant? Could the authors expand on the explanation that "...the continued requirement for Bora may reflect additional regulatory controls on PLK1 activation, for instance through co-localization"?

- The authors state: "the PLK1E121A-K208R double mutant variant was substantially phosphorylated also upon expression in HeLa cells, contrary to the PLK1E121A single mutant (Figure 5E and Figure S5D). While the identity of the kinase responsible for this phosphorylation remains unknown, this finding demonstrates that the K208R mutation increases the efficiency of PLK1 phosphorylation in cells." It would be very exciting and important to identify this kinase to completely define the molecular requirements for Plk1 activation.

- It is intriguing that exchange of lysine to arginine, both basic amino acids, has such a dramatic effect. Could the authors elaborate on how this relatively minor change can have such a dramatic effect?

****Minor comments:****

- Statistical testing should be included in all the comparisons of mitotic entry timing. Why is the median and not the mean shown when showing mitotic entry timing?

- I would prefer if it was always made very clear even just by looking at the figures whether

the data shown is an AlphaFold structure model or an experimentally determined structure.

2. Significance:

Significance (Required)

This is a very important study. Firstly, it reveals unique and unexpected structural features of how Plk1 is activated specifically by Aurora A-Bora as revealed by AlphaFold modeling. Secondly, these findings are then validated experimentally for their in vivo cellular effects that together provide an explanation for how Plk1 is activated at mitotic entry, revealing a key facet of how mitotic kinase activities are regulated in space and time.

The manuscript complements the study by Pintard et al. that has been submitted at the same time. While Pintard et al. provide more details on the Aurora A-Bora complex, Verza and coworkers focus more on effects on Plk1 activity, which they also extensively validate in vivo, in the cellular context. Together, these two studies make a very substantial advance in understanding the structural details of Plk1 activation by Aurora A-Bora revealing many unique features. At the same time they provide one of the first models for how mitotic kinase activities are controlled in space and time.

3. How much time do you estimate the authors will need to complete the suggested revisions:

Estimated time to Complete Revisions (Required)

(Decision Recommendation)

Between 1 and 3 months

Yes

Reviewer #1 (Evidence, reproducibility and clarity (Required)):

The study from Verza and colleagues carefully evaluates the long-sought mechanism by which Aurora A controls PLK1 activity, and helps explain why BORA binds to both Aurora A and PLK1, and especially why T210 is not a remotely efficient Aurora A substrate in the absence of other factors. PLK1 does not autoactivate in the presence of Aurora A/Bora, . Instead, the authors find that in the presence of BORA, substrate inefficiency is overcome, allowing Aurora A (but not Aurora B, or any of likely the vast number of other basophilic kinases) to activate PLK1 through T210 phosphorylation. Mechanistically, and guided by the remarkable powers of AF modelling using AF Multimer and analysis with purified components, Thr210 phosphorylation by Aurora A is shown to be prevented by mutation of Lys208 to Arg (a subtle, but conserved PLK1 change in the upstream consensus), suggesting a relevant interface for BORA that assembles a holoenzyme capable of guiding Aurora A into a conformation relevant for PLK1 activation via T210 phosphorylation. BORA pre-phosphorylation by CDK overcomes mutational effects, suggesting that this is the relevant 'activating' species of BORA. Findings are then tested in cells, with data (notably BoraF56A or BoraW58A mutants that cannot employ the 'Y8/Y10 TPX2' docking motif first characterised for Xenopus TPX2 in PMID: 14701852). Clear and well controlled data suggest that no other mitotic kinase can phosphorylate PLK1WT on Thr210 after Bora depletion. Together, this confirms the hypothesis that pBora and PLK1 interact to 'licence' Aurora A to phosphorylate the PLK1 activation loop, a very nice explanation of years of work in various systems.

We are grateful to the reviewer for their appreciation of our work

Referees cross-commenting

We all agree this should be published, essentially as is.

We have modified the text and performed a few additional experiments to answer requests by all three reviewers.

Reviewer #1 (Significance (Required)):

Building on quite recent work by Lorca and colleagues (and a second, complementary study submitted to Review Commons, which is discussed in this study, coming to similar conclusions), this work makes ample use of available reagent sets, whilst generating a number of new ones, and exploits modelling to reveal the likely mechanism of PLK1 activation by Aurora A/BORA. It is a significant study, with particular strengths in revealing why it is Aurora A/BORA *specifically* that controls PLK1, rather than other potential activities. It will be of interest to all students of the cell cycle, not just kinase aficionados.

Highlights. The K208R mutation is used to great effect, as is AF modelling and convincing (and challenging) sets of cellular experiments; the supplementary work is extremely strong in supporting the main findings. The section on phosphatases is an excellent addition, providing the impetus for future work; the demonstration that PLK1 T210 is comparatively resilient to dephosphorylation by

mitotic phosphatases in particular is a nice finding, as is the suggestion that Lys to Arg mutation may modulate the phosphatase interactome.

We are very grateful for the support of our manuscript.

Minor points:

1) T295V Aurora A is a preferred Aurora A mutant here, since it can be used to prove that Aurora A T-loop phosphorylation is not involved, but what happens with WT Aurora A in a side-by-side pBORA/PLK1 experiment (an inactive version of which lacking pT288 can be generated by dephosphorylation?)

We suspect the reviewer meant T288V of Aurora A, rather than T295V, as T288 is mentioned in the penultimate line. We are happy to present a new experiment that addresses this important question (Figure S1E). We have generated the dephosphorylated Aurora A sample suggested by the reviewer. The main conclusion is that CDK1-phosphorylated Bora (pBora) allows dephosphorylated Aurora A to auto-phosphorylate on the activation loop (lanes 8 and 11). In the absence of pBora, dephosphorylated Aurora A is essentially unable to auto-phosphorylate (lane 5). In presence of pBora, all three Aurora A variants (phosphorylated, dephosphorylated, and T288V were able to phosphorylate the PLK1 activation loop). This is an interesting and important addition to this manuscript, and we are grateful to the reviewer for raising this point.

2) p.3 with TPX2 led to propose is missing 'them'.

This was meant as written, i.e. the observation “led to propose” the activation model discussed in reference 32.

Reviewer #2 (Evidence, reproducibility and clarity (Required)):

In the manuscript by Verza et al., the authors investigated the molecular requirements for activation of Polo-like kinase 1 by phosphorylation of its T-loop. The authors used AlphaFold predictions to model the interaction between PLK1 and Bora and subsequently tested these predictions using in vitro biochemical assays. The main finding is that a transient interaction of Bora with several residues in the catalytic domain of PLK1 is necessary for PLK1 modification at Threonine 210 and subsequently its activation. Interestingly, Lysine 208 in the activation loop of PLK1 turned out to be the major determinant for the requirement of Aurora-A:Bora complex for PLK1 activation and its mutation to arginine allowed for efficient phosphorylation of PLK1 by other Aurora kinases and their cofactors. Nevertheless, this was observed only in vitro, since expression of the K208R mutant failed to rescue PLK1 modification in mitotic cells lacking Bora (Fig. 5A).

(Significance (Required)):

Although the main proteins involved in PLK1 activation at mitotic entry have been already known, the precise molecular mechanism was missing. Original data presented in this manuscript fill the gap in the understanding of how PLK1 is activated by the joined function of the upstream kinase Aurora-A and its cofactor Bora. Therefore, this study will be of interest for the broad cell cycle community.

We are very grateful to the reviewer for his/her support of our manuscript.

Major comments

1. The proposed model in which Lys208 contributes to recognition and activation of PLK1 by Aur-A:Bora complex is interesting. Throughout the manuscript, the authors use antibody to detect the phosphorylated T210 residue in the T loop. A potential technical problem is a different reactivity of this antibody with various epitopes determined not only by the phosphorylation itself but largely also by the neighbouring amino acids. The authors should exclude the possibility that K208R mutation in combination with the phosphorylated Thr210 generates a better epitope recognized by the antibody compared to the wild type sequence. To this end, they could use synthetic (phospho)peptides. Alternatively, they could run radioactive assays to prove that the differences in the signal between WT and K208R mutant represent different level of phosphorylation rather than different reactivity of the antibody.

We have addressed this important concern by providing an independent, orthogonal assessment of T210 phosphorylation using ProQ staining as well as fluorescence detection, in addition to ECL. As ProQ staining is directed to all phosphorylation of a target protein, a pre-condition for the success of this experiment was that T210 is the only residue hit by Aurora A under the condition of our assay. We demonstrate this to be the case. We have added this important new control as Figure S5A-B.

2. Statistical evaluation is missing in experiments comparing the dynamics of mitotic entry (Fig. 3D, 3E, 5C) making the conclusions difficult.

This point also echoes a concern raised by reviewer 3, We have included statistical evaluation for relevant experimental pairs in revised plots.

Minor comments

1. The following sentence in the abstract is confusing and should be reformulated: We show that Aurora A:Bora phosphorylates Thr210 of PLK1 in vitro, while Aurora A, other Aurora A:activator complexes, and Aurora B:INCENP fail to target T210 even at high kinase/substrate ratios.

Thank you. We have reformulated as follows “We show that Aurora A:Bora phosphorylates Thr210 of PLK1 in vitro. On the contrary, T210 was not phosphorylated by isolated Aurora A, additional Aurora A:activator complexes, or Aurora B:INCENP, even when used at high kinase/substrate ratios.”

2. At several places of the manuscript, authors state that the predictions by Alpha-Fold were of high-confidence but without quantification it remains unclear what that means.

That is a good point and we have now added pLDDT and PAE plots for the two main predictions presented in the manuscript. They appear as new panels S1A-B and S2A.

Reviewer #3 (Evidence, reproducibility and clarity (Required)):

The manuscript explores the mechanism of Plk1 activation by Aurora A-Bora. First, it shows that only Aurora A-Bora can phosphorylate Plk1 on Thr210 (the site that in turn activates Plk1). Other Aurora A-activator complexes (e.g. AurkA-CEP192, AurkA-Tpx2) and Aurora B do not efficiently phosphorylate Plk1. This phosphorylation requires not only activation of Aurora A by Bora, but also a transient, direct interaction between Bora and Plk1 that was identified through structural modeling. The authors experimentally test several sites that are predicted to be involved in this interaction. To this end, they introduce point mutants into cultured mammalian cells and monitor Plk1 activity as well mitotic entry delay (a key Plk1 phenotype). They show that Plk1 has specifically adapted to only be activated by Aurora A-Bora, due to the presence of Lys208 near the Thr210 site. If the site is mutated, the K208R mutant Plk1 can be efficiently phosphorylated by other Aurora A-activator complexes as well as Aurora B. They further show that this site is necessary to keep Plk1 phosphorylated throughout mitosis as the K208R mutant, although it is easier to be phosphorylated, it is also easier to be dephosphorylated after the degradation of Bora.

Overall, the manuscript presents a large amount of carefully planned and executed experiments. That data shown on the figures are very high quality and fully support the conclusions drawn. Western blots are quantified and quantifications show data from multiple repeats.

(Significance (Required)):

This is a very important study. Firstly, it reveals unique and unexpected structural features of how Plk1 is activated specifically by Aurora A-Bora as revealed by AlphaFold modeling. Secondly, these findings are then validated experimentally for their in vivo cellular effects that together provide an explanation for how Plk1 is activated at mitotic entry, revealing a key facet of how mitotic kinase activities are regulated in space and time.

The manuscript complements the study by Pintard et al. that has been submitted at the same time. While Pintard et al. provide more details on the Aurora A-Bora complex, Verza and coworkers focus more on effects on Plk1 activity, which they also extensively validate in vivo, in the cellular context. Together, these two studies make a very substantial advance in understanding the structural details of Plk1 activation by Aurora A-Bora revealing many unique features. At the same time they provide one of the first models for how mitotic kinase activities are controlled in space and time.

We are very grateful to the reviewer for his/her support of our manuscript.

For some experiments an average of only two experiments are shown.

Our yearlong experience with simple biochemical assays with purified proteins, in particular size-exclusion chromatography and pulldown assays, has invariably confirmed that they deliver extremely reproducible results if experiments are executed under strictly similar conditions, which is the reason why we have limited the number of repeats to two in these limited cases.

More generally, Western blots have been developed using ECL, which is known to provide a linear response only in a rather limited range. It would have increased data quality if the authors used fluorescently labeled antibodies.

We are aware of this potential limitation of ECL and agree with the reviewer that we should eventually upgrade our detection method to one with a greater dynamic range of detection. To try alleviate this concern on the present dataset, and in combination to our response to a specific concern by reviewer 2, we repeated the experiment in Figure 4A and quantified the results by WB using ECL and fluorescent antibodies and also visualizing phosphorylated proteins with ProQ staining. We show that when protein amounts typically used in our experiments are used, the quantified outcome of these experiments is essentially identical. The results are presented in a new Figure S5A-B.

Major comments:

- If Plk1 K208R mutant can be phosphorylated in vitro by all other kinases besides Bora-Aurora A, why does Bora depletion affect mitotic entry timing in cells with Plk1 K208R mutant? Could the authors expand on the explanation that "...the continued requirement for Bora may reflect additional regulatory controls on PLK1 activation, for instance through co-localization"?

We have now rephrased and expanded our explanation and write: "Thus, even in cells expressing PLK1^{K208R}, there is a continued requirement for Bora. This requirement may reflect additional regulatory controls on PLK1 activation, for instance on co-localization of PLK1 and the T210 kinase that only Bora can satisfy."

- The authors state: "the PLK1E121A-K208R double mutant variant was substantially phosphorylated also upon expression in HeLa cells, contrary to the PLK1E121A single mutant (Figure 5E and Figure S5D). While the identity of the kinase responsible for this phosphorylation remains unknown, this finding demonstrates that the K208R mutation increases the efficiency of PLK1 phosphorylation in

cells." It would be very exciting and important to identify this kinase to completely define the molecular requirements for Plk1 activation.

We agree with the reviewer that this is an important point that the work does not address. We note that providing an accurate answer to this question won't be straightforward, as AGC kinases share related target sites. Thus, we cannot exclude the involvement of kinases other than Aurora-family members in this phosphorylation. We also suspect that Aurora A:Bora remains a plausible candidate for this phosphorylation, maybe because of the localization constraints mentioned in our answer to the previous major comment. To address this question, we tried to ablate Aurora A or Aurora B activity with selective inhibitors. However, under our experimental conditions, we did not find an unequivocal "sweet spot" that would have allowed us to ascribe the effects of inhibition to one of these kinases. For this reason, we prefer to defer the answer to this important question to future analyses.

- It is intriguing that exchange of lysine to arginine, both basic amino acids, has such a dramatic effect. Could the authors elaborate on how this relatively minor change can have such a dramatic effect?

We have now included a sentence that elaborates on the possible causes of this effect: "The guanidinium group of arginine is permanently protonated under physiological conditions, and its positive charge is delocalized by resonance. This gives it ideal properties for electrostatic and hydrogen-bonding interactions. In contrast, lysine's side chain is a primary amine with a localized charge that can, in principle, transiently deprotonate. We therefore surmise that the more stable and delocalized cationic nature of the arginine side chain promotes stronger electrostatic interactions with negatively charged residues or phosphorylated groups in Aurora kinase, thereby stabilizing the complex."

Minor comments:

- Statistical testing should be included in all the comparisons of mitotic entry timing. Why is the median and not the mean shown when showing mitotic entry timing?

Thank you for raising this point, which echoes a similar point by reviewer 2. We have added statistical significance for comparison of relevant experimental pairs. We initially chose to display the median to minimize the effects of outliers. In this revised version of the manuscript, we opted to display the mean and performed statistical analysis as discussed in the main text.

- I would prefer if it was always made very clear even just by looking at the figures whether the data shown is an AlphaFold structure model or an experimentally determined structure.

We agree with the reviewer and now added "AF model" to panel 2A (this label was already present for the other models displayed in the manuscript).

Dr. Andrea Musacchio
Max Planck Institute of Molecular Physiology
Mechanistic Cell Biology
Otto Hahn Strasse 11
Dortmund 44227
Germany

20th Nov 2025

Re: EMBOJ-2025-122931-T
Molecular requirements for PLK1 activation by T-loop phosphorylation

Dear Andrea,

Thank you for submitting your revised Review Commons manuscript for consideration by The EMBO Journal. Given the interest of the subject and the very positive transferred referee reports, I decided to treat the work like a regular EMBO Journal revision, and returned it directly to the original referee 3, who was fully satisfied with your responses and revision. We shall therefore be happy to accept the study for publication, following adjustment to our specific journal format and incorporation of a few other editorial modifications as follows:

GENERAL:

- Please download and complete our author checklist (link provided below).
- Please provide suggestions for a short 'blurb' text prefacing and summing up the conceptual aspect of the study in two sentences (max. 250 characters), followed by 3-5 one-sentence 'bullet points' with brief factual statements of key results of the paper; they will form the basis of an editor-written 'Synopsis' accompanying the online version of the article. Please also upload a synopsis image, which can be used as a "visual title" for the synopsis section of your paper. The image (maybe based on Figure 7?) should be in PNG or JPG format, and please make sure that it remains in the modest dimensions of (exactly) 550 pixels wide and 300-600 pixels high.
- You shall also receive a separate message from our Source Data curation team, with instructions on how to prepare and upload relevant image and numerical raw data.

TEXT:

- Please upload the manuscript text as an editable DOCX file, without figures included.
- Please adjust the order as well as the headers of the different manuscript sections: Title page with complete author information, Abstract, Keywords, Introduction, Results, Discussion, Methods, Data Availability, Acknowledgements, Disclosure and Competing Interests Statement, References, Main Figure Legends, Tables, Expanded Figure Legends.
- Please reduce the number of keywords on the abstract page to five (ideally choosing broad general terms).
- Please note that Materials and Methods need to be described in the main text using our 'Structured Methods' format (for detail, see <https://www.embopress.org/page/journal/14693178/authorguide#structuredmethods>). The in-text "Methods" section should contain method and protocol descriptions (ideally using a step-by-step protocol format to facilitate adoption of the methodologies across labs), while all key reagents, experimental models, software and relevant equipment - including their sources and relevant identifiers - should be listed in a separately uploaded Reagents and Tools Table, a template for which can be downloaded from the above section of our Author Guidelines.
- As we are switching from a free-text author contribution statement towards a more formal statement based on Contributor Role Taxonomy (CRediT) terms, please remove the present Author Contribution section and instead specify each author's contribution(s) directly in the Author Information page of our submission system during upload of the final manuscript. See <https://casrai.org/credit/> for more information.
- Please rename the Competing Interest section into "Disclosure and Competing Interests Statement", in accordance with our updated Guide to Authors (<https://www.embopress.org/competing-interests>)
- Please carefully go through the reference list and make sure that each reference is complete with citation year, journal name, volume/page/locator numbers - some of these informations are currently missing in several entries. Also, please adjust the

format for citation of preprints as specified in our author guidelines:

The citation in the text should be: "(preprint: NAME1 et al, YEAR)"

The citation in the reference list: "NAME1, NAME2, ... (YEAR) ARTICLE TITLE. bioRxiv/medRxiv/ResearchSquare(...) doi: XXX"

For citation of the cosubmitted work, I would suggest to best add a placeholder "in press" citation which could be updated at proof stage.

- Please include a dedicated "Data Availability" section at the end of the Material and Methods (suggested wording: "The [structural coordinates | microarray | mass spectrometry] data from this publication have been deposited to the [name of the database] database [URL] and assigned the identifier [accession | permalink | hashtag]."); should there no data deposition to public repositories linked to the study, this should still be stated as "This study includes no data deposited in external repositories."

- Please make sure to include all relevant funding information not only in the manuscript text, but also in our submission system.

DATA:

- Please upload all main Figures and Expanded View figures as individual, image-only files with sufficient resolution/quality for production.

- Please refer to our author guide (www.embopress.org/page/journal/14602075/authorguide#expandedview) regarding "supplementary information". I would suggest to turn the current supplemental figures into Expanded View figures - nomenclature/call-outs "Figure EV1/2/3..." - and their legends should be included at the very end of the manuscript text.

- For "Tables S1-3", please either turn them into Tables 1-3, or collate them in a dedicated Appendix PDF as "Appendix Tables S1-3", or -maybe best- consider whether their contents could become incorporated into the required Reagents & Tools table (see above).

- Finally, during routine pre-acceptance checks, our data editors have raised the following queries regarding figures, data, and legends; I would appreciate if you briefly answered to them in the cover letter of your final submission, and made the requested text modifications with changes/additions highlighted via the "Track changes" option, to facilitate our final checking:

1. Please note that the exact p values are not provided in the legends of figures 3D, E; 5C
2. Please note that the error bars are not defined in the legends of figures 5A, B, E

Should you need additional guidance/feedback regarding this final adjustments, please do not hesitate to contact us directly. Thank you again for the opportunity to consider this work for The EMBO Journal, and I look forward to receiving your final version!

With kind regards,

Hartmut

9) To facilitate reproducibility and cross-laboratory adoption of methodologies, please structure the Materials & Methods section as outlined in our guide to authors, including a completed Reagents and Tools Table that can be downloaded from our author guidelines as well (<https://www.embopress.org/page/journal/14602075/authorguide#structuredmethods>).

10) Digital image enhancement is acceptable practice, as long as it accurately represents the original data and conforms to community standards. If a figure has been subjected to significant electronic manipulation, this must be clearly noted in the figure legend and/or the 'Materials and Methods' section. The editors reserve the right to request original versions of figures and the original images that were used to assemble the figure. Finally, we generally encourage uploading of numerical as well as gel/blot image source data; for details see: embopress.org/page/journal/14602075/authorguide#sourcedata

In the interest of ensuring the conceptual advance provided by the work, we recommend submitting a revision within 3 months (18th Feb 2026). Please discuss the revision progress ahead of this time with the editor if you require more time to complete the revisions. Use the link below to submit your revision:

Link Not Available

Referee #1:

We previously reviewed this manuscript for Review Commons. As detailed in that review, we find this work important and of high quality. The revised version now submitted to EMBO Journal addresses most of our criticisms. Therefore, I am happy to recommend the manuscript for publication.

Rev_Com_number: RC-2025-03134

New_manu_number: EMBOJ-2025-122931-T

Corr_author: Musacchio

Title: Molecular requirements for PLK1 activation by T-loop phosphorylation

The authors addressed the remaining editorial issues.

Dear Andrea and Arianna,

Thank you for submitting your final re-revised manuscript files to our editorial office. I am pleased to inform you that your manuscript has been accepted for publication in the EMBO Journal!

You may qualify for financial assistance for your publication charges - either via a Springer Nature fully open access agreement or an EMBO initiative. Check your eligibility: <https://link.springer.com/journal/44318/how-to-publish-with-us>

With kind regards,

Hartmut

Please note that it is The EMBO Journal policy for the transcript of the editorial process (containing referee reports and your response letters) to be published as an online supplement to each paper. If you should prefer removal of any referee-only figures included in the point-by-point response(s), e.g. because they may still be used for future publication or because they have been reproduced from published work by others, please do let us know immediately via response email.

More information is available here: <https://link.springer.com/partners/embo-press/editorial-policies#Peer%20review>

Rev_Com_number: RC-2025-03134

New_manu_number: EMBOJ-2025-122931R

Corr_author: Musacchio

Title: Molecular requirements for PLK1 activation by T-loop phosphorylation